# EVOLVING ALIGNMENT *via* ASYMMETRIC SELF-PLAY

*Scalable Preference Fine-Tuning Beyond Static Human Prompts*

## ABSTRACT

Current RLHF approaches for aligning large language models (LLMs) typically assume a fixed prompt distribution, which is sub-optimal and limits the generalization capabilities for language models. To address this issue, we introduce a general framework that casts alignment as an asymmetric game between two players: (i) **a creator** that generates increasingly informative prompt distributions using the reward model, and (ii) **a solver** that learns to produce more preferred responses on prompts produced by the creator. This framework of Evolving Alignment *via* Asymmetric Self-Play (**eva**), results in a simple and scalable approach that can utilize any existing RLHF algorithm. **eva** outperforms state-of-the-art methods on widely-used benchmarks, without the need of any additional human crafted prompts. Specifically, **eva** improves the win rate of GEMMA2-9B-IT on Arena-Hard from 51.6% to 60.1% with DPO, from 55.7% to 58.9% with SPPO, from 52.3% to 60.7% with SimPO, and from 54.8% to 60.3% with ORPO, surpassing its 27B version and matching `claude-3-opus`. This improvement is persistent even when new human crafted prompts are introduced. Finally, we show **eva** is effective and robust under various ablation settings.

*What I cannot create, I do not understand.*

*– Richard P. Feynman*

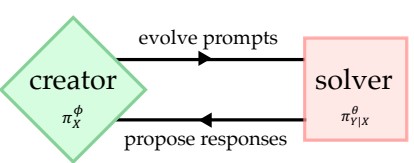
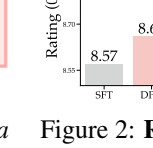
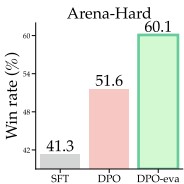
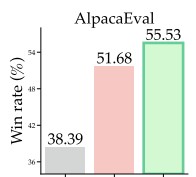

Figure 1: **eva**: Open-Ended RLHF *via* Asymmetric Self-Play. The creator is the prompt generation policy $\pi_X$ and the solver is the response policy $\pi_{Y|X}$.

Figure 2: **Results**: Our method **eva** achieves concrete performance gain especially on *hard* alignment benchmarks, without relying on any additional human prompts. Here, we report results for DPO-**eva**; see more in §4.1.

## 1 INTRODUCTION

Long-lived artificial intelligence must deal with an ever-evolving, open-ended world, yet currently face constraints in both the *scale* and *quality* of available data, and the *growth rate* at which new, useful information is created. High quality human data, crucial for scaling large language model (LLM) based intelligence, is projected to run out in the next few years (Villalobos et al., 2024); the quality of such data is also expected to stagnate: as LLMs become more capable, they need to solve increasingly complex or new challenges, requiring training data beyond abilities of humans to create. This necessitates a new fundamental mechanism for self-improving, where models can continuously self-generate and self-solve harder problems. We thereby investigate the research question below:

*Can language models* self-create *new, learnable tasks to work on,*
*to* self-improve *to* generalize *better for human preferences alignment?*

Figure 3: **Pipeline**: We generalize classical RLHF with open-ended RLHF, optimized with a creator-solver game for self-improving language models. Our proposed **eva** strategically evolves prompt distributions with a creator policy, which synthesizes prompts with an easy-to-implement *estimate, sample then evolve* procedure; specifically, it estimates the informativeness for each prompt by how contrastive the self-generated responses are to the prompt, from the reward signals it receives. The creator evolves new prompts from highly informative prompts, which the solver uses for continual training. Both the solver and creator policy can share the same network or operate independently. See more on our minimax-regret objective that drives the above design in § 3.

Many preference optimization algorithms (Christiano et al., 2017; Rafailov et al., 2023; Zhao et al., 2023; Wu et al., 2024; Liu et al., 2023a; Guo et al., 2024) have been proposed to improve the alignment of language models, however, they all default to fixed prompt training distributions. Such fixed training paradigm inevitably leads to: (i) *generalization issues* (models may underperform or hack on instructions that are insufficiently represented within the fixed set) and (ii) *efficiency issues* (data annotation and model training are costly, however not all prompt provide the same utility; it is wasteful to invest in sub-optimal fixed set, while identifying informative prompts through human efforts is expensive and time-consuming) (Team et al., 2023; 2024b; Dubey et al., 2024).

The objective thereby should not only be to optimize over a specific, static distribution of prompts, yet to develop an agent that can autonomously evolve the training data distribution for self-improvement, to align well across unseen, novel environments or tasks (instantiated by prompts).

Thus, we develop **eva** (Evolving Alignment *via* Asymmetric Self-Play), as illustrated in Figure 1. Central to our approach is a game with the minimax-regret objective, achieved through alternating optimization between creating prompts and solving them. The interplay encourages evolving curricula (Parker-Holder et al., 2022), potentially benefits both generalization and efficiency (see also § 3.4). Orthogonal to many recent self-play studies in LLM alignment (Munos et al., 2023; Choi et al., 2024; Wu et al., 2024), **eva** is *asymmetric* (Sukhbaatar et al., 2017), with two policies of different goals:

- **Creator**: evolves the prompt distribution for alignment.
- **Solver**: produces responses and optimizes alignment based on the evolving prompts.

Our main contributions are summarized as:

- **A new principle**: We propose a generalized **Open-Ended RLHF** objective for aligning language models, which seeks to jointly optimize the prompt distribution and the response policy, thus incentivizes models to self-improve to generalize well on new, unseen tasks beyond the initial training prompt distribution for alignment, as in Definition 1.
- **A new algorithm**: To optimize the objective, we design a practical algorithm *via* asymmetric self-play, which is implemented through alternating optimization in a **creator-solver game**, and can be easily plugged into any existing alignment pipeline, as in Algorithm 1.
- **State-of-the-art performance**: We empirically validate our method on public alignment benchmarks and present general strong performance improvement when plugged in with different preference optimization algorithms (*i.e.*, DPO, SPPO, SimPO, ORPO). We also conduct extensive ablation studies that provide additional insights on the choice of informativeness metric, reward model, and training schedules, as in § 4.

**eva is easy to implement**. We hope it can serve as a scalable method for the research community to build open-ended, robust, and self-improving language agents, that align with human values.

## 2 PRELIMINARIES

We hereby review major concepts, which we later in § 3 use *regret* and the proxy by *advantage* to identify informative prompts, leading to learning curricular implicitly maximizing *contrastive ratio* .

**Alignment by RLHF.** Classical RLHF (Ouyang et al., 2022) optimizes on a fixed distribution $\mathcal{D}$:

$$\max_{\pi_{\boldsymbol{\theta}}} \mathbb{E}_{\mathbf{x}\sim\mathcal{D},\mathbf{y}\sim\pi_{\boldsymbol{\theta}}(\cdot|\mathbf{x})}\left[ r(\mathbf{x},\mathbf{y}) \right] - \mathbb{E}_{\mathbf{x}\sim\mathcal{D}}\left[ \beta \cdot \mathbb{D}_{\mathrm{KL}}\left[ \pi_{\boldsymbol{\theta}}(\mathbf{y}\mid\mathbf{x}) \,\|\, \pi_{\mathrm{SFT}}(\mathbf{y}\mid\mathbf{x}) \right] \right], \qquad (1)$$

where $\mathbf{x}$ and $\mathbf{y}$ denote the prompts and responses, and $r(\cdot,\cdot)$ is the reward function.

**Reward.** Let the *optimal policy* of Eq. 1 be $\pi^{\star}(\cdot)$ and $Z(\cdot)$ be the partition function, we have:

$$r(\mathbf{x},\mathbf{y}) = \beta \cdot \log\frac{\pi^{\star}(\mathbf{y}\mid\mathbf{x})}{\pi_{\mathrm{SFT}}(\mathbf{y}\mid\mathbf{x})} + \beta \cdot \log Z(\mathbf{x}). \qquad (2)$$

**Regret.** Given the optimal policy $\pi^{\star}$, the *regret* of a policy $\pi_{\boldsymbol{\theta}}$ at $\mathbf{x}$ is:

$$\mathrm{Regret}(\mathbf{x},\pi_{\boldsymbol{\theta}}) = \mathbb{E}_{\mathbf{y}\sim\pi^{\star}(\mathbf{y}|\mathbf{x})}\left[ r(\mathbf{x},\mathbf{y}) \right] - \mathbb{E}_{\mathbf{y}\sim\pi_{\boldsymbol{\theta}}(\mathbf{y}|\mathbf{x})}\left[ r(\mathbf{x},\mathbf{y}) \right]. \qquad (3)$$

**Advantage.** The *advantage* function quantifies how much better a response $\mathbf{y}$ is w.r.t. a baseline:

$$A(\mathbf{x},\mathbf{y}) = r(\mathbf{x},\mathbf{y}) - \mathbb{E}_{\mathbf{y}'\sim\pi(\mathbf{y}'|\mathbf{x})}\left[ r(\mathbf{x},\mathbf{y}') \right]. \qquad (4)$$

Variants of advantage (*e.g.*, the worst-case advantage $A^{\star}_{\min}$) are related to regret, as shown in Table 2.

**Direct preference optimization.** The DPO (Rafailov et al., 2023) objective for RLHF is:

$$\mathcal{L}^{\mathrm{DPO}}_{\beta}(\pi_{\boldsymbol{\theta}}) = \sum_{(\mathbf{y}_+,\mathbf{y}_-,\mathbf{x})\in\mathcal{D}} -\log\left[ \sigma\left( \beta \cdot \Delta^{\mathbf{x}}_{\boldsymbol{\theta};\,\mathrm{ref}} \right) \right], \qquad (5)$$

where we use $+, -$ to denote chosen and rejected responses, and denote the **contrastive ratio** as:

$$\Delta^{\mathbf{x}}_{\boldsymbol{\theta};\,\mathrm{ref}} := \log\frac{\pi_{\boldsymbol{\theta}}(\mathbf{y}_+\mid\mathbf{x})}{\pi_{\mathrm{ref}}(\mathbf{y}_+\mid\mathbf{x})} - \log\frac{\pi_{\boldsymbol{\theta}}(\mathbf{y}_-\mid\mathbf{x})}{\pi_{\mathrm{ref}}(\mathbf{y}_-\mid\mathbf{x})}. \qquad (6)$$

By reward reparameterization with Eq. 2, advantage also relates to contrastive ratio, as in § 3.4.

## 3 METHOD

**Algorithm overview.** On a high level, **eva** extends classical RLHF to open-ended RLHF *via* a creator that adapts prompt distributions with an easy-to-implement *estimate, sample then evolve* procedure, mimicing the *minimax-regret* policy of asymmetric self-play games, as detailed in §3.3.

---

**Algorithm 1 eva: Evolving Alignment via Asymmetric Self-Play**

**Input:** initial policy $\pi_{\boldsymbol{\theta}_0}$, initial prompt set $\mathcal{X}_0$

1: **for** iteration $t = 1, 2, \ldots$ **do**

    ▽   /* **creator step** */

2:    *estimate informativeness*:     $\mathcal{X}_{t-1} \leftarrow \{(\mathbf{x}_i, \mathbf{info}(\mathbf{x}_i)) \mid \mathbf{x}_i \in \mathcal{X}_{t-1}\}$

       *sample subset*:             $\mathcal{X}^{\mathtt{info}}_{t-1} \leftarrow \mathbf{sample}(\mathcal{X}_{t-1})$

       *self-evolve prompts*:      $\mathcal{X}_t \leftarrow \mathbf{evolve}(\mathcal{X}^{\mathtt{info}}_{t-1})$

    ▽   /* **solver step** */

3:    *self-generate responses*:    $\forall\, \boldsymbol{x}_i \in \mathcal{X}_t$, generate $\{\boldsymbol{y}^{(j)}_i\} \sim \pi_{\boldsymbol{\theta}_{t-1}}(\cdot \mid \boldsymbol{x}_i)$

       *annotate rewards*:         $\mathcal{X}'_t \leftarrow \mathcal{X}_t \cup \{(\boldsymbol{y}^{(j)}_i, r^{(j)}_i)\}$

       *preference optimization*:   $\boldsymbol{\theta}_t \leftarrow \boldsymbol{\theta}_{t-1} - \eta\nabla_{\boldsymbol{\theta}}\mathcal{L}_{\mathcal{X}'_t}(\boldsymbol{\theta})$

4: **end for**

5: **return** final solver policy $\pi_{\boldsymbol{\theta}_T}$

---

**Section overview.** We aim to develop language models that can *self-improve to generalize well on new, unseen tasks* beyond the initial training prompt distribution for alignment. In §3.1, we present the limitations of classical RLHF and generalize it to the new objective of open-ended RLHF. In §3.2, we introduce the creator-solver game to tractably optimize the objective. In §3.3, we detail the practical algorithm, compatible with any preference optimization method as the solver, with our designed creator in the loop. Lastly in §3.4, we present various interpretations for **eva**.

### 3.1 THE PRINCIPLE: OPEN-ENDED RLHF FOR JOINT SELF-IMPROVEMENT

**Intuition.** Classical RLHF (*cf.*, Eq. 1) optimizes over a *static* prompt distribution, meaning that the agent is only aligned to a fixed prompt set $\mathcal{D}$, making it brittle when it is evaluated on new problems from the ever-changing real world. Our Open-Ended RLHF breaks away from this static framework, with the goal to develop an agent that *generalizes* well across *unseen, novel* environments (where the tasks entailed in prompts may not have been explicitly encountered during training). To achieve this, we must design a new objective that agents can generate its problems for self-improvement.

**Formalization.** We introduce an *open-ended* reference distribution $p_{\text{ref}}(\mathbf{x})$, which is conceptually approached by a prompt policy $\pi_\phi(\mathbf{x})$, co-evolving with the response policy for alignment:

---

**Definition 1 (Open-Ended RLHF)** *We define* evolving alignment *as the open-ended joint optimization on the prompt and response policy for alignment w.r.t the joint reference policy:*

$$\max_{\phi,\theta} \mathbb{E}_{\mathbf{x}\sim\pi_\phi(\cdot),\, \mathbf{y}\sim\pi_\theta(\cdot|\mathbf{x})} \left[ r(\mathbf{x},\mathbf{y}) \right] - \beta \cdot \mathbb{D}_{KL}\left[ \pi_{\phi,\theta}(\mathbf{x},\mathbf{y}) \,\|\, \pi_{ref}(\mathbf{x},\mathbf{y}) \right], \qquad (7)$$

*where $\pi_{\phi,\theta}(\mathbf{x},\mathbf{y}) := \pi_\phi(\mathbf{x}) \cdot \pi_\theta(\mathbf{y} \mid \mathbf{x})$ and $\pi_{ref}(\mathbf{x},\mathbf{y}) := p_{ref}(\mathbf{x}) \cdot \pi_{SFT}(\mathbf{y} \mid \mathbf{x})^a$.*

---
[a]*This generalizes classical RLHF (Eq. 1). One may extend the above and rewrite coefficients to be:*

$$\max_{\phi,\theta} \mathbb{E}_{\mathbf{x}\sim\pi_\phi(\cdot)}\left[ \mathbb{E}_{\mathbf{y}\sim\pi_\theta(\cdot|\mathbf{x})}\left[ r(\mathbf{x},\mathbf{y}) \right] - \beta_1 \mathbb{D}_{\text{KL}}\left[\pi_\theta(\mathbf{y}|\mathbf{x}) \,\|\, \pi_{\text{SFT}}(\mathbf{y}|\mathbf{x})\right] \right] - \beta_2\, \mathbb{D}_{\text{KL}}\left[\pi_\phi(\mathbf{x}) \,\|\, p_{\text{ref}}(\mathbf{x})\right]. \quad (8)$$

---

Here, $p_{\text{ref}}(\mathbf{x})$ represents an *underspecified*, potentially intractable probability distribution over possible tasks (instantiated *via* prompts) in the wild, as a realizable **open-ended reference** that covers the full diversity and complexity of tasks agents may encounter, *not* the initial static prompt set $\mathcal{D}$. The joint regularization towards $\pi_{\text{ref}}(\mathbf{x},\mathbf{y})$ captures the objective for agents to generalize on alignment in $p_{\text{ref}}(\mathbf{x})$ with broader open-ended prompts, while being close to the SFT policy $\pi_{\text{SFT}}(\mathbf{y}|\mathbf{x})$.

### 3.2 THE MECHANISM: ASYMMETRIC SELF-PLAY *via* THE CREATOR V.S. SOLVER GAME

**Intuition.** It can be hard to directly solve Eq. 7, due to (i) the **intractability** of the underspecified reference (Dennis et al., 2020), (ii) the **instability** of joint differentiation (Goodfellow et al., 2014). We present a heuristic of iterative alternating optimization by casting it as an **asymmetric creator-solver game**, that **implicitly achieves** the conceptual probability matching of $\mathbb{D}_{\text{KL}}[\pi_\phi(\mathbf{x}) \,\|\, p_{\text{ref}}(\mathbf{x})]$ by iteratively creating a *sequence* of prompt distributions to expand over the task space.

- **Creator** : the prompt player $\pi_\phi(\mathbf{x})$ that strategically generate new prompt distributions.
- **Solver** : the response player $\pi_\theta(\mathbf{y} \mid \mathbf{x})$ that learns to optimize for preference alignment.

This game serves as one potential choice to implicitly achieve the Open-Ended RLHF principle.

- By design, the creator can guide the solver with an *informative prompt curriculum*, for it to develop more general capabilities to handle complexities in the wild (Jiang, 2023).
- Intuitively, this resembles RL optimization *via* expectation-maximization (Dayan and Hinton, 1997; Singh et al., 2023), where $\phi$ for the prompt distribution is fixed at each step.

**Formalization.** We consider the *minimax regret* strategy (Savage, 1951), where the solver minimizes and the creator maximizes regret[1], *i.e.*, the reward difference of the current and KL-optimal policy:

$$\text{Regret}(\pi_\phi, \pi_\theta) = \mathbb{E}_{\mathbf{x}\sim\pi_\phi(\cdot)}\left[ \mathbb{E}_{\mathbf{y}\sim\pi_\theta(\mathbf{y}|\mathbf{x})}\left[ r(\mathbf{x},\mathbf{y}) \right] - \mathbb{E}_{\mathbf{y}\sim\pi_{\text{KL}}^\star(\mathbf{y}|\mathbf{x})}\left[ r(\mathbf{x},\mathbf{y}) \right] \right]. \qquad (9)$$

---
[1]Please see § G and Eq. 14 for details on the KL-optimal policy $\pi_{\text{KL}}^\star(\mathbf{y} \mid \mathbf{x})$.

At the equilibrium (Nash et al., 1950), prior works (Dennis et al., 2020) have shown:

> **Remark 1 (Minimax Regret)** *If the above solver-creator game reaches an equilibrium, the solver follows a minimax regret policy, i.e., the worst-case regret is bounded:*
>
> $$\pi^{\star}_{\mathcal{Y}|\mathcal{X}} \in \arg\min_{\pi_{\mathcal{Y}|\mathcal{X}}} \max_{\pi_{\mathcal{X}}} \mathbb{E}_{\mathbf{x} \sim \pi_{\mathcal{X}}} \Big[ \text{Regret}(\mathbf{x}, \pi_{\mathcal{Y}|\mathcal{X}}) \Big]. \tag{10}$$

To illustrate, open-ended RLHF allows for the creation of evolving prompt distributions that challenge the agent progressively for better generalization; the introduced minimax regret objective further adds *robustness* on top of such evolving curricula by *incentivizing agents to perform well in all cases*.

However, while it is often straightforward for the **solver** to minimize the regret (*e.g.*, by direct policy optimization), the optimal policy remains unknown during optimization, thus regret as the decision signal is often intractable to the **creator** – regret approximation is needed. We design the proxy below for creator's regret approximation (see § G for more), also as a metric for prompt informativeness:

> **Definition 2 (Informativeness)** *We estimate the informativeness of a prompt* $\mathbf{x}$ *in preference optimization by the (absolute) worst-case optimal advantage, approximating regret in Eq. 3:*
>
> $$|\hat{\text{Regret}}(\mathbf{x}, \pi_{\boldsymbol{\theta}})| \leftarrow info_{\boldsymbol{\theta}}(\mathbf{x}) := r(\mathbf{x}, \mathbf{y}_+) - r(\mathbf{x}, \mathbf{y}_-), \tag{11}$$
>
> *where*
>
> $$\mathbf{y}_+ := \arg\max_{\mathbf{y}_i} r(\mathbf{x}, \mathbf{y}), \quad \mathbf{y}_- := \arg\min_{\mathbf{y}_i} r(\mathbf{x}, \mathbf{y}), \tag{12}$$
>
> *and* $\{\mathbf{y}_i\}_{i=1}$ *is a set of responses sampled from* $\pi_{\boldsymbol{\theta}}(\cdot \mid \mathbf{x})$ *and* $r(\cdot, \cdot)$ *is the reward oracle.*

We use the informativeness proxy to guide the creator for prompt distribution adaptation, which has a few useful properties under different interpretations, as in § 3.4. In this way, we define a mechanism that the solver learns to improve, as the creator keeps challenging the solver on its weaknesses.

### 3.3 THE PRACTICAL ALGORITHM

We now illustrate **eva** of Algorithm 1, with practical implementations specified in § A.

#### 3.3.1 THE CREATOR STEP: ESTIMATE, SAMPLE THEN EVOLVE

Plainly, the creator finds most useful prompts and generate variants of them for preference optimization. One may relate this to *evolution strategies* (Schwefel, 1977) which find the most promising species, then mutate and crossover, or to *curriculum RL* (Parker-Holder et al., 2022) which finds environments with high-regret levels, then edits within some distance. As in Section 3.2, we do not seek a differentiable creator in this work. The creator is implemented in three steps as in Figure 3.

**Step 1:** `info`(·) – *estimate the informativeness.* For each $\mathbf{x}$ in the prompt set $\mathcal{X}_t$, we generate responses, annotate rewards and estimate a informativeness metric to $\mathbf{x}$ by Eq. 11 (see also Table 2).

**Step 2:** `sample`(·) – *weighted sampling for an informative subset.* By using the informativeness metric as the weight, we sample an informative prompt subset $\mathcal{X}_t^{\text{info}}$ to be evolved later. This is similar to finding high-regret levels in curriculum RL.

**Step 3:** `evolve`(·) – *evolving for a proximal region of high-advantage prompts.* Our algorithm is agnostic to and does not rely on any specific evolving method. We take EvolInstruct (Xu et al., 2023a) as an off-the-shelf method, which conducts in-depth (*i.e.*, adding constraints, deepening, concretising, complicating) and in-breadth evolving (*i.e.*, mutation) for prompts. Specifically, we iterate over each prompt in the $\mathcal{X}_t^{\text{info}}$, where each one is evolved to multiple variations, then optionally mix the newly generated prompts with a uniformly sampled buffer from $\mathcal{X}_t$ to create $\mathcal{X}_t'$.

#### 3.3.2 THE SOLVER STEP: SOLVE THEN OPTIMIZE

This step is the classical preference optimization (Rafailov et al., 2023), where responses are generated and the gradient descent is performed. Take the pointwise reward model setting as an example, for every prompt, we sample $n$ responses with reward annotated for each; we take the responses with the maximal and the minimal reward to construct the preference pairs, then optimize upon. This implicitly minimizes the regret to the KL-optimal policy, which we present in more details at § G.

Put together, **eva** can unify existing iterative optimization pipeline (Tran et al., 2023) with a new creator module, which can either share the same network as the solver policy or operate independently.

### 3.4 Understanding the Informativeness Proxy in Different Intuitive Ways

**Learning potential.** Our metric intuitively identifies the learning potential of a prompt by measuring the gap between the best and worst response to it from the solver. We reason, that prompts eliciting *both* high-reward and low-reward outcomes, reflect *learnable* tasks where the model is capable of improving but has not yet mastered, thereby implying learning potential (*cf.*, Jiang et al. (2021b)).

**Worst-case guarantees.** The minimax-regret objective, by design, leads to solvers that perform robustly across the prompt space, thus gives the worst-case guarantee. While exact equilibrium may not be attainable with approximation, our empirical results in § 4.2.1 demonstrate robustness.

**Auto-curricula for the players.** We visualize curriculum induced by **eva** in § E. With the stochastic policy, the advantage may be heuristically understood as the reward difference between *a base solver* and *a reference solver*. Rather than optimizing separate solvers (Dennis et al., 2020), we sample multiple times from the same policy to create the pair. In this way, the creator is incentivized to produce new prompts that are just out of the comfort zone of solvers (Chaiklin et al., 2003):

- For overly challenging prompts, both solutions perform poorly, leading to a low proxy.
- For overly easy prompts, the base solution already performs well, again giving a low proxy.
- The optimal strategy is to find prompts that are just beyond the solver's current capability.

**Auto-curricula inherent to Contrastive Optimization.** Contrastive preference optimization generalizes DPO and a family of algorithms (*c.f.*, Hejna et al. (2023); Rafailov et al. (2023); Tang et al. (2024)), many of whose losses monotonically decrease as the contrastive ratio increases. Here, by Eq. 2 and Eq. 6, the *contrastive ratio* can be written via the *advantage-based proxy*:

$$A_{\min}^{\star}(\mathbf{x}) = \beta \cdot \Delta_{\boldsymbol{\theta}^{\star};\,\mathrm{ref}}^{\mathbf{x}}. \tag{13}$$

By our proxy, we implicitly incentivize the creator to generate prompts that *bring the most contrastive responses*, which decrease the loss the most. This matches the curriculum learning literature, which prioritizes (in **eva**, *generatively* prioritizes) examples with smaller losses for better convergence and generalization (Bengio et al., 2009). We hence suggest the *Contrastive Curriculum Hypothesis*: in contrastive preference optimization, prioritizing prompts with higher contrastive ratio improves sample efficiency and generalization. We show initial empirical results on this in § 4.2.1 and § 4.2.4.

## 4 Experiments

**Datasets and models for training.** We use **UltraFeedback** (Cui et al., 2023) as the training dataset, which contains diverse high-quality prompts that are primarily human-generated. We use the instruction-finetuned GEMMA-2-9B (Team et al., 2024b) as the primary model, which is a strong baseline for models of its size. Detailed experimental setting can be found in § A.

**Evaluation settings.** We choose: (i) **AlpacaEval 2.0** (Dubois et al., 2024), which assesses general instruction following with 805 questions; (ii) **MT-Bench** (Zheng et al., 2023), which evaluates multi-turn instruction following with 80 hard questions in 8 categories; (iii) **Arena-Hard** (Li et al., 2024b), which is derived from 200K user queries on Chatbot Arena with 500 challenging prompts across 250 topics. We use gpt-4-1106 as the judge and gpt-4-0314 as the baseline for win rate.

**Optimization algorithms.** We focus on direct preference optimization and consider the following:

- **With reference policy**: DPO (Rafailov et al., 2023), SPPO (Wu et al., 2024).
- **Without reference policy**: SimPO (Meng et al., 2024), ORPO (Hong et al., 2024).

**Reward models as preference oracles.** We use ARMORM-8B (Wang et al., 2024) as our default reward model as the human-preference proxy, and consider the following for ablation studies:

- **Pointwise**: ARMORM-8B (Wang et al., 2024), SKYWORKRM-27B (Liu and Zeng, 2024).
- **Pairwise**: PAIRRM-0.4B (Jiang et al., 2023), PAIRRM-8B (Dong et al., 2024).

## 4.1 MAIN RESULTS

In general, **eva** brings notable gains in alignment without relying on any human-crafted data, thus offering more efficiency. In the base setup, building on the one-iteration finetuned model ($\theta_{0\to1}$), **eva** adds a creator to self-evolve the prompt set of the initial iteration and uses any preference optimization algorithm for an additional open-ended RLHF iteration, resulting in $\theta_{1\to\tilde{1}}$[2].

**eva achieves self-improvement.** As shown in red rows in Table 1, **eva** yields notable performance improvement over $\theta_{0\to1}$ across different optimization algorithms, especially on the harder Arena-Hard benchmark, which is recognized to be more challenging and distinguishable among others due to the complexity of its prompts and its fairer scoring system (Li et al., 2024b; Meng et al., 2024). Specifically, **eva** brings $8.4\%$ gain with SimPO as the solver, and $8.5\%$ gain with DPO as the solver, surpassing its 27B version and matching `claude-3-opus-240229` as reported on the AH leaderboard, while using fully self-automated prompt generation for alignment. Interestingly, **eva** brings the least gains on AlpacaEval 2.0, a simpler evaluation benchmark. This indicates **eva** improves the most for challenging tasks.

**eva can surpass human-crafted prompts.** We further show that **eva**-prompt-trained models ($\theta_{1\to\tilde{1}}$) can match and even outperform those trained on additional new prompts from UltraFeedback ($\theta_{1\to2}$) (which we denoted as human prompts), while being much cheaper and more efficient. Additionally, on MT-Bench, training with new human prompts typically show decreased performance in the first turn and only modest gains in the second turn. In contrast, **eva** notably enhances second-turn performance. We hypothesize that **eva** evolves novel, learnable prompts that include characteristics of second-turn questions, reflecting emergent skills like handling follow-up interactions.

| Model Family ($\to$) | GEMMA-2-9B-IT | | | | | |
|---|---|---|---|---|---|---|
| **Benchmark** ($\to$) | **Arena-Hard** | **MT-Bench** | | | **AlpacaEval 2.0** | |
| **Method** ($\downarrow$) **/ Metric** ($\to$) | **WR** (%) | **avg. score** | **1st turn** | **2nd turn** | **LC-WR** (%) | **WR** (%) |
| $\theta_0$: SFT | 41.3 | 8.57 | 8.81 | 8.32 | 47.11 | 38.39 |
| $\theta_{0\to1}$: DPO | 51.6 | 8.66 | 9.01 | 8.32 | 55.01 | 51.68 |
| $\theta_{1\to\tilde{1}}$: + eva | **60.1** (+8.5) | **8.90** | **9.04** | **8.75** (+0.43) | 55.35 | 55.53 |
| $\theta_{1\to2}$: + new human prompts | 59.8 | 8.64 | 8.88 | 8.39 | **55.74** | **56.15** |
| $\theta_{0\to1}$: SPPO | 55.7 | 8.62 | 9.03 | 8.21 | 51.58 | 42.17 |
| $\theta_{1\to\tilde{1}}$: + eva | **58.9** (+3.2) | **8.78** | **9.11** | **8.45** (+0.24) | **51.86** | **43.04** |
| $\theta_{1\to2}$: + new human prompts | 57.7 | 8.64 | 8.90 | 8.39 | 51.78 | 42.98 |
| $\theta_{0\to1}$: SimPO | 52.3 | 8.69 | 9.03 | 8.35 | 54.29 | 52.05 |
| $\theta_{1\to\tilde{1}}$: + eva | **60.7** (+8.4) | **8.92** | **9.08** | **8.77** (+0.42) | **55.85** | **55.92** |
| $\theta_{1\to2}$: + new human prompts | 54.6 | 8.76 | 9.00 | 8.52 | 54.40 | 55.72 |
| $\theta_{0\to1}$: ORPO | 54.8 | 8.67 | 9.04 | 8.30 | 52.17 | 49.50 |
| $\theta_{1\to\tilde{1}}$: + eva | **60.3** (+5.5) | **8.89** | **9.07** | **8.71** (+0.41) | **54.39** | **50.88** |
| $\theta_{1\to2}$: + new human prompts | 57.2 | 8.74 | 9.01 | 8.47 | 54.00 | **51.21** |

Table 1: **Main results.** Our **eva** achieves notable alignment gains and can surpass human prompts on major benchmarks across a variety of representative direct preference optimization algorithms.

## 4.2 ABLATION STUDIES

We conduct in-depth ablation studies on **eva**, with findings below to be elaborated on later:

- § 4.2.1 - **informativeness metric**: our *regret*-based metric outperforms other alternatives.

- § 4.2.2 - **sample-then-evolve procedure**: our method outperforms greedy selection.

- § 4.2.3 - **scaling w/ reward models**: the alignment gain of **eva** scales with reward models.

- § 4.2.4 - **continual training** : our method has monotonic gain with incremental training; the *evolved data and schedule* by **eva** serves as an *implicit regularizer* for better local minima.

---

[2]Unless stated otherwise, each iteration uses 10K prompts (*i.e.*, 1/6 partition from UltraFeedback in classical training). We denote $\theta_{t\to t+1}$ as the model trained with new human prompts based on the $t$-th checkpoint, and $\theta_{t\to\tilde{t}}$ as the model trained with evolved prompts from the $t$-th checkpoint w/o adding any new human prompts.

### 4.2.1 THE CHOICE OF INFORMATIVENESS METRICS: INFO($\cdot$)

| Metric | info($\mathbf{x}$) | Related Approximation |
|---|---|---|
| $A_{\min}^\star$: worst-case optimal advantage | $\left|\min_{\mathbf{y}} r(\mathbf{x}, \mathbf{y}) - \max_{\mathbf{y}'} r(\mathbf{x}, \mathbf{y}')\right|$ | minimax regret (Savage, 1951) |
| $A_{\text{avg}}^\star$: average optimal advantage | $\left|\frac{1}{N}\sum_{\mathbf{y}} r(\mathbf{x}, \mathbf{y}) - \max_{\mathbf{y}'} r(\mathbf{x}, \mathbf{y}')\right|$ | Bayesian regret (Banos, 1968) |
| $A_{\text{dts}}^\star$: dueling optimal advantage | $\left|\max_{\mathbf{y} \neq \mathbf{y}^\star} r(\mathbf{x}, \mathbf{y}) - \max_{\mathbf{y}'} r(\mathbf{x}, \mathbf{y}')\right|$ | min-margin regret (Wu and Liu, 2016) |

Table 2: The reward-advantage-based metrics that serve as the informativeness proxies for prompts.

| Model Family ($\rightarrow$) | GEMMA-2-9B-IT | | | | | |
|---|---|---|---|---|---|---|
| Benchmark ($\rightarrow$) | **Arena-Hard** | **MT-Bench** | | | **AlpacaEval 2.0** | |
| Method ($\downarrow$) / Metric ($\rightarrow$) | WR (%) | avg. score | 1st turn | 2nd turn | LC-WR (%) | WR (%) |
| $\theta_{0 \rightarrow 1}$: DPO | 51.6 | 8.66 | 9.01 | 8.32 | 55.01 | 51.68 |
| $\theta_{1 \rightarrow \tilde{1}}$: + **eva** (uniform) | 57.5 | 8.71 | 9.02 | 8.40 | 53.43 | 53.98 |
| $\theta_{1 \rightarrow \tilde{1}}$: + **eva** (var($\boldsymbol{r}$)) | 54.8 | 8.66 | 9.13 | 8.20 | 54.58 | 52.55 |
| $\theta_{1 \rightarrow \tilde{1}}$: + **eva** (avg($\boldsymbol{r}$)) | 58.5 | 8.76 | 9.13 | 8.40 | 55.01 | 55.47 |
| $\theta_{1 \rightarrow \tilde{1}}$: + **eva** (1/avg($\boldsymbol{r}$)) | 56.7 | 8.79 | 9.13 | 8.45 | 55.04 | 54.97 |
| $\theta_{1 \rightarrow \tilde{1}}$: + **eva** (1/$A_{\min}^\star$) | 52.3 | 8.64 | 8.96 | 8.31 | 53.84 | 52.92 |
| $\theta_{1 \rightarrow \tilde{1}}$: + **eva** ($A_{\text{avg}}^\star$) (our variant) | 60.0 | 8.85 | 9.08 | 8.61 | **56.01** | **56.46** |
| $\theta_{1 \rightarrow \tilde{1}}$: + **eva** ($A_{\text{dts}}^\star$) (our variant) | 60.0 | 8.86 | **9.18** | 8.52 | 55.96 | 56.09 |
| $\theta_{1 \rightarrow \tilde{1}}$: + **eva** ($A_{\min}^\star$) (our default) | **60.1** (+8.5) | **8.90** | 9.04 | **8.75** (+0.43) | 55.35 | 55.53 |

Table 3: **Choice of informativeness metric matters.** Our metric by *advantage* achieves the best performances, comparing with others as weights to sample for evolving. See also § F for visualization.

**Advantage as the informativeness metric outperforms baselines.** As in Table 3, **eva** offers an effective curriculum by the advantage-based proxy as the informativeness metric (bottom row):

- Comparing w/ *uniform evolving* (brown): Existing baselines generate prompts in a uniform manner (Yuan et al., 2024) w/o informativeness measure (*cf.*, the principle of insufficient reason (Keynes, 1921; Tobin et al., 2017)). **eva** concretely outperforms, corroborating Das et al. (2024) that uniform learners can suffer sub-optimality gaps.

- Comparing w/ *other heuristics* (blue): Prior practices (Team et al., 2023) tried heuristics like prioritizing prompts w/ the most variance in its rewards or w/ the lowest/highest average. We find our advantage based methods (red) outperforms those heuristics; see § F for more.

- Comparing w/ the *inverse advantage* (purple): Contrary to curriculum learning, a line of works conjecture that examples w/ higher losses may be prioritized (Jiang et al., 2019; Kawaguchi and Lu, 2020), which can be done by inverting our metric. We find it significantly *hurt* the alignment gain, corroborating Mindermann et al. (2022) that those examples are often noisy, unlearnable or irrelevant, meaning our curriculum is effective and practical.

- Among our *advantage variants* (red): We designed variants of our default advantage-based metric, as in Table 2; the default $A_{\min}^\star$ remains competitive among its peers. Together, the advantage-based principle provides a robust guideline for prompt sampling and evolving.

The lesson is that we must be selective about which are the promising to evolve, otherwise unlearnable, noisy or naïve prompts may hinder learning. Our regret-inspired metric represents a solid baseline.

### 4.2.2 THE EFFECT OF THE SAMPLE-THEN-EVOLVE PROCEDURE

**The design of evolve($\cdot$) in eva is effective.** As in Table 4, we show:

- Removing the evolve($\cdot$) step: if we only do subset sampling or ordered selection, we still have gain, but not as much as w/ evolving (*e.g.*, **eva** brings $4.8\%$ additional wins on AH).

- Altering the sample($\cdot$) step: if we greedily select prompts by the metric instead of using them as weights for importance sampling, the performance will be weaker as we evolve.

This shows that simply adaptive training within a fixed prompt distribution is unsatisfactory; our open-ended RLHF with *generative* prompt exploration gives a substantial headroom for self-improvement.

| Benchmark ($\rightarrow$) | Arena-Hard | MT-Bench | | | AlpacaEval 2.0 | |
|---|---|---|---|---|---|---|
| Method ($\downarrow$) / Metric ($\rightarrow$) | WR (%) | avg. score | 1st turn | 2nd turn | LC-WR (%) | WR (%) |
| $\theta_{0\rightarrow1}$: DPO | 51.6 | 8.66 | 9.01 | 8.32 | 55.01 | 51.68 |
| $\theta_{1\rightarrow\tilde{1}}$:   [no evolve]-greedy | 56.1 | 8.68 | 8.98 | 8.38 | 54.11 | 53.66 |
| $\theta_{1\rightarrow\tilde{1}}$:   [no evolve]-sample | 55.3 | 8.69 | 9.00 | 8.38 | 54.22 | 54.16 |
| $\theta_{1\rightarrow\tilde{1}}$:   + **eva**-greedy (our variant) | 59.5 | 8.72 | 9.06 | 8.36 | 54.52 | 55.22 |
| $\theta_{1\rightarrow\tilde{1}}$:   + **eva**-sample (our default) | **60.1** | **8.90** | 9.04 | **8.75** | **55.35** | **55.53** |

Table 4: **Effect of evolving.** The blue are those training w/ only the informative subset and w/o evolving); we denote -sample for the default weighted sampling procedure in Algo 1, while using -greedy for the variant from the classical active data selection procedure (*cf.*, a recent work (Muldrew et al., 2024) and a pre-LLM work (Kawaguchi and Lu, 2020)), which selects data by a high-to-low ranking via the metric greedily. We show evolving brings a remarkable alignment gain (the red v.s. the blue); and as we evolve, sampling is more robust than being greedy (*cf.*, Russo et al. (2018)).

### 4.2.3 SCALING POINTWISE AND PAIRWISE REWARD MODELS

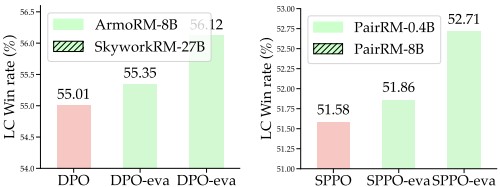

Figure 4 presents the length-controlled win rate of **eva** on AlpacaEval using pointwise and pairwise reward models of varying scales. The results give a clear trend: as the quality of reward models improve, **eva** brings higher alignment gain. The scaling observation shows the effectiveness of **eva** in exploiting more accurate reward signals to choose informative prompts for better alignment. One takeaway is interaction w/ the external world is essential for intelligence. The more accurate reward signals observed, the better the agent incentivize themselves to improve (*cf.*, Silver et al. (2021)).

Figure 4: **eva scales with quality of reward models**, under pointwise RMs w/ DPO (*left*) and pairwise RMs w/ SPPO (*right*). Note SPPO handles general preferences thus requires pairwise RMs, and DPO relies on the Bradley-Terry assumption, for which pointwise RMs are suitable.

### 4.2.4 **EVA** IMPROVES BOTH SAMPLE EFFICIENCY AND GENERALIZATION

We then continuously run the default *incremental training* (*i.e.*, trainining from the last checkpoint w/ the evolved set in each iteration), as in Fig 5, **eva** presents *monotonic performance gain* over iterations, and surpasses that trained w/ new human prompts, implying the generalization benefit[3].

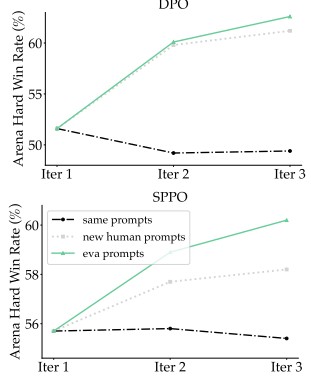

The solutions found by **eva** cannot be recovered by training longer w/ a fixed distribution (the dashed), nor by naïvely sourcing new prompts w/o examining informativeness (the gray dotted), thus our generative data schedule is effective.

In Table 5, we ablate **eva** in *scratch training*, *i.e.*, training w/ the full set (the evolved and the original data). **eva** is competitive in incremental training, thus *learns more effective* with *less data* – a nice bonus via minimax regret (Jiang et al., 2021a).

| Benchmark ($\rightarrow$) | Arena-Hard | MT-Bench | AlpacaEval 2.0 |
|---|---|---|---|
| Method ($\downarrow$) / Metric ($\rightarrow$) | WR (%) | avg. score | LC-WR (%) |
| $\theta_0$: SFT | 41.3 | 8.57 | 47.11 |
| $\theta_{0\rightarrow1}$: DPO | 51.6 | 8.66 | 55.01 |
| $\theta_{0\rightarrow\tilde{1}}$:   **eva** (scratch) | 59.8 | 8.88 | 54.59 |
| $\theta_{1\rightarrow\tilde{1}}$:   **eva** (incremental) | **60.1** | **8.90** | **55.35** |

Figure 5: **Continual training.** **eva** stays robust w/ more iterations in incremental training.

Table 5: **Ablation on incremental v.s. scratch training.**

---

[3]Behaviors of the dashed/dotted lines relate to *loss of plasticity* (Ash and Adams, 2019; Dohare et al., 2023; Abbas et al., 2023; Xue et al., 2024). Classical works resolve it by the optimization view (*e.g.*, weight perturbing), whereas **eva** provides a fresh data view, potentially mimicking an **implicit regularizer for better generalization**.

## 5 RELATED WORKS

**Self-improving algorithms and iterative optimization.** This line of work focuses on iteratively generating samples from the response policy and continuously re-training the policy by selected self-generated samples. Major works include ReST (Gulcehre et al., 2023; Singh et al., 2023), STaR (Zelikman et al., 2022), RFT (Yuan et al., 2023), RAFT (Dong et al., 2023), self-improving LLMs (Huang et al., 2022; Yuan et al., 2024); in the context of preference optimization, iterative DPO (Xu et al., 2023b; Tajwar et al., 2024; Tran et al., 2023; Xiong et al., 2024; Pang et al., 2024) has proven effective. Most works focus on self-training by improving in $\mathcal{Y} \mid \mathcal{X}$, while we *jointly optimize* both responses and prompts via generative exploration in the $(\mathcal{X}, \mathcal{Y})$ space. Among them, we also distinctly present a game-theoretic framework with the minimax-regret principle as the guidance.

**Prompt synthesis for language models.** Existing works include Self-Instruct (Wang et al., 2022), WizardLM (Xu et al., 2023a; Luo et al., 2023), Self-Align (Sun et al., 2024), Glan (Li et al., 2024a), EvoPrompt (Guo et al., 2023), Magpie (Xu et al., 2024) and others (Long et al., 2024). `eva` is an orthogonal contribution since any synthesis method can be plugged in as the `evolve(·)` for the creator. Importantly, our work presents a new reward-related metric to endow prompt the notion of informativeness, with new implications as in § 3.4. We also focus on preference optimization algorithms, while those existing works primarily use synthesized prompts in an SFT-only way.

**Self-play and curriculum RL.** Agents trained on a fixed data distribution are often brittle and may struggle to adapt to the real world (Hughes et al., 2024a). Self-play (Samuel, 1959; Goodfellow et al., 2014; Silver et al., 2016) addresses this by having the agent learn through self-interaction, thus creating more diverse experiences and automatic curricula. In asymmetric self-play, the paradigm centers on "*Alice proposing a task, and Bob doing it*" (Sukhbaatar et al., 2017; Samvelyan et al., 2023; Beukman et al., 2024a; Dennis et al., 2020). We revive the classical asymmetric self-play principle (Sutton et al., 2011) in optimizing language models. Unlike traditional curriculum RL (Parker-Holder et al., 2022), which usually renders environments from specified levels (Dennis et al., 2020), our approach is *generative* by nature, as we directly generate contexts from the auto-regressive language models.

**Self-play in RLHF.** A growing line of research frames RLHF as a *symmetric* self-play game, where both players are response players (Munos et al., 2023; Wu et al., 2024; Choi et al., 2024; Rosset et al., 2024). However, these methods still rely on a fixed prompt distribution thus is sub-optimal. In contrast, we solve this by *asymmetric* self-play, enabling evolving prompt distributions for more generalizable language agents. During our work, we notice one concurrent paper adopting the asymmetric two-player setup (Zheng et al., 2024), however (i) it applies to adversarial attack tasks instead of general alignment benchmarks, (ii) it is incompatible w/ direct preference optimization, and (iii) it relies on the maxmin principle (which is known to be producing unlearnable environments (Dennis et al., 2020)) instead of the minimax *regret* principle (Fan, 1953; Savage, 1951) as we do. We also first precisely defined the new problem of open-ended RLHF, which generalizes over classical RLHF.

## 6 CONCLUDING REMARKS

**Limitations and future directions.** `eva` defines a new paradigm for alignment, opening up many new directions, *e.g.*, (i) extending to differentiable creator policies, combining w/ other `evolve(·)` methods; (ii) evolving for more iterations w/ on-policy solvers like RLOO (Ahmadian et al., 2024); (iii) investigating exploration bonuses for distribution diversity and coverage, and the self-consuming loop (Gerstgrasser et al., 2024); (iv) extending the game with more players for full automation (*e.g.*, rewarders, critics, rewriters, verifiers, retrievers); (v) extending from alignment to reasoning (*e.g.*, auto-conjecturing for theorem proving (Poesia et al., 2024) can be cast as an asymmetric game), or from the bandits to the trajectories w/ process reward models and hierarchical tree search for creator and solver generations; (vii) further scaling up w/ million-level prompts for post-training.

**Conclusions.** `eva` is a new, simple and scalable framework for aligning language models, and can be plugged into any existing alignment pipeline. The primary takeaway may be that RLHF can be made open-ended: (i) self-evolving joint data distributions can bring significant gain (as shown across various optimization algorithms), and (ii) reward advantage acts as an effective metric informing the collection and creation of *future* prompts for alignment. `eva` presents a new view of alignment by framing it as an asymmetric game between a creator generating *new* and *learnable* prompts and a solver producing preferred responses. `eva` also *incentivizes agents to create problems* rather than to simply *solve problems*, which is a key feature of intelligence, yet model trainers often neglect.

# APPENDIX

The appendix is organized as follows:

- § A - **Details On Reproducibility**
- § B - **Plug-In Loss Functions Used In Main Results**
- § C - **Extended Results for Experiments in the Main Paper**
- § D - **Additional Experiments**
- § G - **Illustration on Methodology**
- § E and § J - **Illustrations on Prompts, Responses and Relevant Distributions**
- § H and § I - **Additional Literature Review**

## A    DETAILS ON REPRODUCIBILITY

Our code is built based on many open-source packages, and we sincerely thank every developer and contributor of these projects for their efforts and contributions to the community.

**Code release.**    We hope to open-source all codes, generated data and trained models, *upon approval* – before then, we are more than happy to provide any clarification to help re-implement `eva` and replicate our results. In general, our code base is made to be simple to use for practitioners, requiring **only a creator module addition** within the commonly adopted Alignment Handbook pipeline.

**Hyperparameter settings.**    We follow the original hyperparameter settings as in (Hong et al., 2024; Meng et al., 2024; Wu et al., 2024), default to be:

| Hyperparameter ($\downarrow$) / Loss ($\rightarrow$) | DPO | ORPO | SimPO | SPPO |
|---|---|---|---|---|
| learning rate | 5e-7 | 5e-7 | 8e-7 | 5e-7 |
| learning rate scheduler | cosine | cosine | cosine | linear |
| $\beta$ | 0.05 | / | 10 | 0.001 |
| $\gamma$ | / | / | 5 | / |
| $\lambda$ | / | 0.5 | / | / |
| no. epochs per iter | 2 | 1 | 1 | 6 |
| warmup ratio per iter | 0.1 | 0.1 | 0.1 | 0.1 |
| effective batch size | 8 | 8 | 32 | 8 |
| max length | 2048 | 2048 | 2048 | 1024 |
| max prompt length | 1024 | 1024 | 1024 | 512 |
| optimizer | adamw | adamw | adamw | rmsprop |

**Iterative Training Settings.**    By default (Tran et al., 2023; Yuan et al., 2024), we train with equal-size prompt subset in each iteration. Unless otherwise specified, we use 10K prompts from the UltraFeedback dataset (Cui et al., 2023) per iteration. The incremental training proceeds as follows:

- $\theta_0$ : Base SFT model.
- $\theta_{0 \rightarrow 1}$ : initialize with $\theta_0$; then train with the prompt split $\mathcal{X}_1$ by self-generated responses from the initial model $\theta_0$.
- $\theta_{1 \rightarrow 2}$ : initialize with $\theta_{0 \rightarrow 1}$; trained with the prompt split $\mathcal{X}_2$ via by self-generated responses from the initial model $\theta_{0 \rightarrow 1}$.

For evolving prompts (*e.g.*, evolving $\mathcal{X}_1$ to $\mathcal{X}_{\tilde{1}}$), with the calculated informativeness metric for each prompt, we normalize them as the weight to do weighted sampling for a 25% informative subset to get $\mathcal{X}_1^{\texttt{info}}$. We then iterate over in $\mathcal{X}_1^{\texttt{info}}$ and call `EvolInstrut` (Xu et al., 2023a) as the plug-in evolving method (with the number of evolutions as 4) using the default mutation templates for (i) in-depth evolving (constraints, deepening, concretizing, increased reasoning steps) and (ii) in-breadth evolving (extrapolation) as implemented in `tasks/evol_instruct/utils.py` of `distilabel==1.3.2`. Next we uniformly select 80% prompts from this evolved dataset and 20% from the original dataset (*i.e.*, the buffer) to form $\mathcal{X}_{\tilde{1}}$. We do not seek extensive parameter search (*e.g.*, the number of evolutions, the evolving ratio) in this stage and encourage future works on exploring this and other plug-in evolving methods. For solver we generate 6 responses per prompt.

**Software environments.** All experiments are conducted on 8xNVIDIA H100 SXM GPUs. Our codebase primarily relies on `transformers==4.40.0`. For the response generation of GEMMA models at the training stage, we use `vllm==0.5.4` with `flashinfer` backend for `CUDA 12.4` and `torch 2.4`. For evolving prompts, we use `distilabel==1.3.2`, and use `LiteLLM` to serve Gemini (default to be `gemini-1.5-pro`) and transformers models (default to be `gemma-2-9b-it`). For evaluation on all benchmarks, we use `sglang==0.2.10` and `openai==1.35.14`, with `gpt-4-1106-preview` as the judge model and `gpt-4-0314-preview` as the baseline model. Specifically for AlpacaEval 2.0, we use `alpaca_eval_gpt4_turbo_fn` as the annotator config. We use 42 as the random seed.

## B    PLUG-IN LOSS FUNCTIONS USED IN MAIN RESULTS

| With Reference Model | |
| --- | --- |
| DPO (Rafailov et al., 2023) | $\ell_\beta(\pi_\theta) = -\log\left[\sigma\left(\beta \cdot \Delta^{\mathbf{x}}_{\pi_\theta;\pi_{ref}}\right)\right] := -\log\left[\sigma\left(\beta \cdot \log\frac{\pi_\theta(\mathbf{y}_+|\mathbf{x})}{\pi_{ref}(\mathbf{y}_+|\mathbf{x})} - \beta \cdot \log\frac{\pi_\theta(\mathbf{y}_-|\mathbf{x})}{\pi_{ref}(\mathbf{y}_-|\mathbf{x})}\right)\right]$ |
| SPPO (Wu et al., 2024) | $\ell_\beta(\pi_\theta) = -\log\left[\sigma\left(\left(\beta \cdot \log\frac{\pi_\theta(\mathbf{y}_+|\mathbf{x})}{\pi_{ref}(\mathbf{y}_+|\mathbf{x})} - \frac{1}{2}\right)^2 + \left(\beta \cdot \log\frac{\pi_\theta(\mathbf{y}_-|\mathbf{x})}{\pi_{ref}(\mathbf{y}_-|\mathbf{x})} + \frac{1}{2}\right)^2\right)\right]$ |

| Without Reference Model | |
| --- | --- |
| SimPO (Meng et al., 2024) | $\ell_{\beta,\alpha}(\pi_\theta) = -\log\left[\sigma\left(\beta \cdot \Delta^{\mathbf{x}}_{\pi_\theta^{1/|\mathbf{y}|};1} - \alpha\right)\right] := -\log\left[\sigma\left(\frac{\beta}{|\mathbf{y}_+|}\log\pi_\theta(\mathbf{y}_+|\mathbf{x}) - \frac{\beta}{|\mathbf{y}_-|}\log\pi_\theta(\mathbf{y}_-|\mathbf{x}) - \alpha\right)\right]$ |
| ORPO (Hong et al., 2024) | $\ell_\lambda(\pi_\theta) = -\log\left[\sigma\left(\lambda \cdot \Delta^{\mathbf{x}}_{odds_\theta;1}\right)\right] := -\log\left[\sigma\left(\lambda \cdot \log\frac{odds_\theta(\mathbf{y}_+|\mathbf{x})}{odds_\theta(\mathbf{y}_-|\mathbf{x})}\right)\right]$, where $odds_\theta = \frac{\pi_\theta}{1-\pi_\theta}$ |

Table 6: Direct preference alignment algorithms used in the main experiments. In parameter tuning, we include an additional negative log-likelihood loss for chosen responses (*i.e.*, $\frac{\gamma}{|\mathbf{y}_+|}\log\pi_\theta(\mathbf{y}_+|\mathbf{x})$).

## C    ADDITIONAL EXPERIMENTAL RESULTS FOR THE MAIN PAPER

In general, **eva** maintains the accuracy on downstream tasks and is robust on those reasoning-heavy tasks, and the scaling with reward models is more prominent on AlpacaEval, possibly due to the training sources for such reward models.

| Method (↓) / Dataset (→) | MUSR-TA | TruthfulQA-Gen | WMDP | GSM8K | GSM-Plus | MMLU-Pro |
| --- | --- | --- | --- | --- | --- | --- |
| $\theta_0$: SFT | 38.80 | 34.76 | 58.62 | 24.64 | 18.62 | 52.08 |
| $\theta_{0\to1}$: DPO | 38.40 | 34.76 | 58.45 | 24.56 | 18.50 | 52.63 |
| $\theta_{1\to\tilde{1}}$:   + eva | 38.40 | 34.15 | 58.40 | 24.26 | 17.96 | **53.03** |
| $\theta_{0\to1}$: SPPO | 40.80 | 34.15 | 58.72 | 24.79 | 18.42 | 52.70 |
| $\theta_{1\to\tilde{1}}$:   + eva | **41.20** | 34.64 | **58.94** | **25.40** | **18.88** | 52.47 |

Table 7: **Performance on Downstream tasks.**

| Model Family (→) | GEMMA-2-9B-IT | | | | | |
| --- | --- | --- | --- | --- | --- | --- |
| Benchmark (→) | | MT-Bench | | Arena-Hard | AlpacaEval 2.0 | |
| Method (↓) / Metric (→) | avg. score | 1st turn | 2nd turn | WR (%) | LC (%) | WR (%) |
| $\theta_{0\to1}$: DPO | 8.66 | 9.01 | 8.32 | 51.6 | 55.01 | 51.68 |
| $\theta_{1\to\tilde{1}}$:   + eva-i (ARMO-8B) | **8.90** | 9.04 | 8.75 | 60.1 | 55.35 | 55.53 |
| $\theta_{1\to\tilde{1}}$:   + eva-i (SKYWORKRM-27B) | 8.75 | 9.07 | 8.43 | **60.3** | **56.12** | **56.40** |

Table 8: **Effect of (pointwise) reward models.**

| Model Family (→) | GEMMA-2-9B-IT | | | | | |
| --- | --- | --- | --- | --- | --- | --- |
| Benchmark (→) | | MT-Bench | | Arena-Hard | AlpacaEval 2.0 | |
| Method (↓) / Metric (→) | avg. score | 1st turn | 2nd turn | WR (%) | LC (%) | WR (%) |
| $\theta_{0\to1}$: SPPO | 8.62 | 9.03 | 8.21 | 55.7 | 51.58 | 42.17 |
| $\theta_{1\to\tilde{1}}$:   + eva-i (PAIRRM-0.4B) | 8.78 | **9.11** | 8.45 | 58.9 | 51.86 | 43.04 |
| $\theta_{1\to\tilde{1}}$:   + eva-i (PAIRRM-8B) | **8.89** | 9.08 | **8.70** | **60.2** | **52.71** | **44.52** |

Table 9: **Effect of (pairwise) reward models.**

# D ADDITIONAL EXPERIMENTAL RESULTS (AS EXTENSIONS)

## D.1 EXPERIMENTS ON DIFFERENT `evolve(·)` METHODS

As an addition to Table 1, we have experimented with three different evolve(·) methods, including:

- **SelfInstruct** (Wang et al., 2022): Given seed prompts, variations are created based on criteria such as verb diversity and style blending (mixing interrogative and imperative styles). Unlike EvolInstruct (Xu et al., 2023a), which generates prompt variations sequentially, this approach generates independently. We follow the one-shot implementation in `self_instruct.py` of `distilabel==1.4.1` and modified the instruction on conciseness so that newly generated prompts have similar lengths compared to the seed prompts.
- **EvolQuality** and **EvolComplexity** (Liu et al., 2023b): The two methods use the same evolutionary approach (*i.e.*, sequentially generating), but with slightly different meta-instructions for prompt generation, where EvolQuality asks to improve the quality (*i.e.*, helpfulness, relevance, etc) of the seed prompt and EvolComplexity asks to improve the complexity (*i.e.*, increased reasoning steps, etc) of the seed prompt. We follow the implementation in `evol_quality/utils.py` and `evol_complexity/utils.py` of `distilabel==1.4.1`.

| Model Family ($\rightarrow$) | GEMMA-2-9B-IT | |
| --- | --- | --- |
| Benchmark ($\rightarrow$) | **Arena-Hard** | |
| Method ($\downarrow$) / Metric ($\rightarrow$) | **WR** (%) | **avg. len** |
| $\theta_0$: SFT | 41.3 | 544 |
| $\theta_{0\rightarrow1}$: DPO | 51.6 | 651 |
| $\theta_{1\rightarrow\bar{1}}$:  + **eva** (evolve(·) = EvolInstruct) | 60.1 | 733 |
| $\theta_{1\rightarrow\bar{1}}$:  + **eva** (evolve(·) = EvolQuality) | 58.7 | 721 |
| $\theta_{1\rightarrow\bar{1}}$:  + **eva** (evolve(·) = EvolComplexity) | **60.6** | 749 |
| $\theta_{1\rightarrow\bar{1}}$:  + **eva** (evolve(·) = SelfInstruct) | 57.2 | 725 |

Table 10: Results of using different evolving methods.

**eva is effective under different evolving methods.** As shown in Table 10, our method brings strong performance gain without training with additional human prompts. Among the experimented methods, we find EvolComplexity shows better results.

We believe the main strength of such method is its **simplicity**. Viewing the evolving process as $\mathbf{x}' \leftarrow p_{\boldsymbol{\theta}}(\cdot \mid \mathbf{x}, \texttt{meta\_prompt})$, one can easily tune the meta prompt in natural language for improved performance. However, such simplicity comes at a price: (i) the main weakness is that the default method does not take **environmental feedback** into account (*e.g.*, rewards received, verbal critique on responses, etc) and relies on the pre-defined meta prompt, thus the evolving may be less directional; we encourage practitioners to consider incorporating more richer feedback during evolving (one way to formulate this is by generative optimization (Yuksekgonul et al., 2024; Cheng et al., 2024; Nie et al., 2024)); (ii) another weakness is that existing method is single-shot (*i.e.*, we evolve based on a single $\mathbf{x}$ each time), thus the **diversity** of the generation may be limited – we anticipate future works improving this with multi-shot evolving by graph-based sampling. In this regard, the evolving process can be viewed as $\{\mathbf{x}'\}_{i=1}^{N} \leftarrow p_{\boldsymbol{\theta}}(\cdot \mid \{\mathbf{x}\}_{i=1}^{M}, \texttt{meta\_prompt}, \texttt{env\_feedback})$.

## D.2 EXPERIMENTS ON NUMBER OF ITERATIONS

As an addition to § 4.2.4, we have experimented with the following settings:

- 10K prompts per iteration with 3 iterations.
- 20K prompts per iteration with 3 iterations (*i.e.*, all seed prompts are used).
- 60K prompts per iteration with 2 iterations (*i.e.*, all seed prompts are used).

Due to time constraints, we did not perform an extensive hyper-parameter search; however, we believe the results presented below sufficiently demonstrate the performance gains achieved by **eva**.

| Model Family ($\rightarrow$) | GEMMA-2-9B-IT | |
|---|---|---|
| Benchmark ($\rightarrow$) | Arena-Hard | |
| Method ($\downarrow$) / Metric ($\rightarrow$) | WR (%) | avg. len |
| $\theta_0$: SFT | 41.3 | 544 |
| $\theta_{0\rightarrow1}$: DPO (10k) | 51.6 | 651 |
| $\theta_{1\rightarrow2}$: DPO (10k) | 59.8 | 718 |
| $\theta_{2\rightarrow3}$: DPO (10k) | 61.2 | 802 |
| $\theta_{1\rightarrow\tilde{1}}$:   + **eva** (10k) | 60.1 | 733 |
| $\theta_{\tilde{1}\rightarrow\tilde{2}}$:   + **eva** (10k) | 62.0 | 787 |
| $\theta_{\tilde{2}\rightarrow\tilde{3}}$:   + **eva** (10k) | 62.2 | 774 |

Table 11: Results of using 10k prompts per iteration (DPO + length-penalized NLL loss).

| Model Family ($\rightarrow$) | GEMMA-2-9B-IT | |
|---|---|---|
| Benchmark ($\rightarrow$) | Arena-Hard | |
| Method ($\downarrow$) / Metric ($\rightarrow$) | WR (%) | avg. len |
| $\theta_0$: SFT | 41.3 | 544 |
| $\theta_{0\rightarrow1}$: DPO (20k) | 53.2 | 625 |
| $\theta_{1\rightarrow2}$: DPO (20k) | 47.0 | 601 |
| $\theta_{2\rightarrow3}$: DPO (20k) | 46.8 | 564 |
| $\theta_{1\rightarrow\tilde{1}}$:   + **eva** (20k) | 59.5 | 826 |
| $\theta_{\tilde{1}\rightarrow\tilde{2}}$:   + **eva** (20k) | 60.0 | 817 |
| $\theta_{\tilde{2}\rightarrow\tilde{3}}$:   + **eva** (20k) | 61.4 | 791 |

Table 12: Results of using 20k prompts per iteration (DPO + length-penalized NLL loss).

| Model Family ($\rightarrow$) | GEMMA-2-9B-IT | |
|---|---|---|
| Benchmark ($\rightarrow$) | Arena-Hard | |
| Method ($\downarrow$) / Metric ($\rightarrow$) | WR (%) | avg. len |
| $\theta_0$: SFT | 41.3 | 544 |
| $\theta_{0\rightarrow1}$: DPO (60k) | 58.9 | 717 |
| $\theta_{1\rightarrow\tilde{1}}$:   + **eva** (60k) | 59.6 | 725 |
| $\theta_{\tilde{1}\rightarrow\tilde{1}'}$:   + **eva** (60k) | 61.9 | 792 |

Table 13: Results of using 60k prompts per iteration (DPO + length-penalized NLL loss).

**eva can bring robust gains with multiple iterations.**   As shown in Table 11, 12, and 13 below, our method presents persistent performance gain over iterations, and concretely surpasses the performance by default DPO training with true human prompts.

However, there exist diminishing marginal gains in iterative off-policy training. We ground **eva** in the iterative (off-policy) preference alignment paradigm due to its efficiency and ease of integration. However, such paradigms inherently face diminishing returns, where performance gains decrease with successive iterations, as previously observed in (Wu et al., 2024; Setlur et al., 2024; Yuan et al., 2024; Nikishin et al., 2022). While the generative data schedule in **eva** mitigates these challenges and extends beyond default training with human prompts (see also §4.2.4), the gains can weaken over iterations. We summarize potential reasons as: (i) the **off-policy signal decay** – as the number of examples increases, signals from the off-policy data become weaker due to distributional shift; (ii) the **loss of plasticity**, where the agent's ability to learn good policies decreases in continuing training with more iterations (Nikishin et al., 2022); (iii) the **ability of the solver** – as we evolve more harder prompts, it is harder for the solver to produce preferred response (thus more explicit reasoning techniques may be needed); (iv) the **ability of the reward model** to correctly provide reward signals to responses and thus informativeness signals to prompts, as there may exists distributional mismatch.

Thus, we envision future work to build on **eva** by: (i) exploring its integration with **on-policy RLHF** (*e.g.*, instead of evolving prompts in iterations, one may evolve in batches); (ii) **enhancing solver capabilities**, such as sampling more responses during inference or leveraging meta-instructions to guide deeper reasoning; (iii) online training of RM to co-evolve with the creator and the solver.

### D.2.1 BONUS EXPERIMENTS ON `rewriter`(·) IN THE LOOP

Though beyond the current package, we present the basic idea here for practitioners to build upon. The motivation comes from the hypotheses derived from § D.2: as the prompts gets harder by evolving, there may be greater demands on the solver's capabilities *compared to earlier iterations*. As such, the solver may not be naively treated the same. One may address this by either inference-time scaling on responses or introducing meta-instructions to explicitly enhance the solver's reasoning.

We design a proof-of-concept experiment *w.r.t* the latter by adding **rewriter** in **eva**'s solver step. Previously, as in Algo. 1 and § 3.3.2, for each prompt $\mathbf{x}$, we generate multiple responses, and choose the best as $\mathbf{y}_+$ and the worst as $\mathbf{y}_-$ for preference optimization. Now, we add one more rewriting step that attempts to enhance $\mathbf{y}_+$ to be $\mathbf{y}'_+$, by applying a rewriting instruction (Liu et al., 2023b) that asks the solver to alter $\mathbf{y}_+$ with imporved helpfulness, relevance, reasoning depths, creativity and details while keeping the similar length. We then train with $(\mathbf{x}, \mathbf{y}'_+, \mathbf{y}_-)$ for preference optimization. Table 14 shows that adding the rewriter yields concrete performance gains over the default training method, while keeping the training budget and slightly increasing cost for offline data generation.

| Model Family (→) | GEMMA-2-9B-IT | |
|---|---|---|
| Benchmark (→) | **Arena-Hard** | |
| Method (↓) / Metric (→) | **WR** (%) | **avg. len** |
| $\theta_0$: SFT | 41.3 | 544 |
| $\theta_{0\rightarrow1}$: DPO | 51.6 | 651 |
| $\theta_{1\rightarrow\bar{1}}$:    + **eva** | **60.1** | 733 |
| $\theta_{1\rightarrow\bar{1}}$:    + **eva** with **rewriter** | **61.9** | 741 |

Table 14: Results of adding **rewriter** in the **solver** step.

## E CURRICULUM VISUALIZATION OVER ITERATIONS

We now present initial observations supporting the intuition in § 3.4, where **eva** brings auto-curricula and the creator is incentivized to create new prompts that are both learnable and worth-learning.

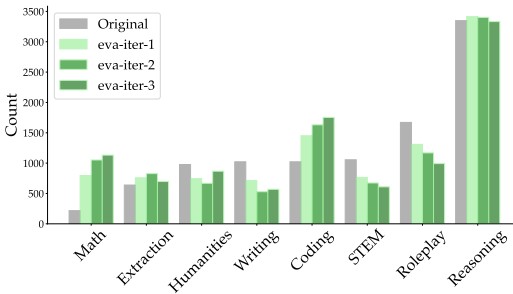

Figure 6: **Training distributions.** The prompt distribution of Table 11 for evolved prompts by zero-shot classification. **eva** creates a curriculum that prioritizes math / coding prompts over iterations.

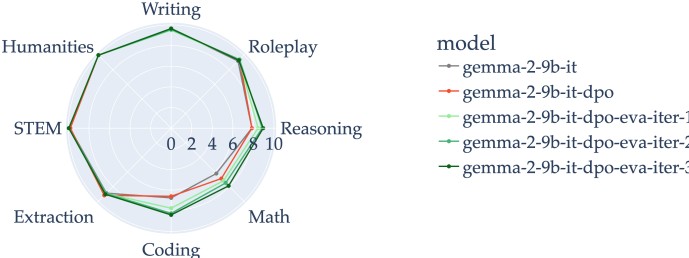

Figure 7: **Benchmark performance.** The radar figure for ratings on MT-Bench (Zheng et al., 2023), where each category contains ten problems. **eva** prioritizes and gradually improves on coding, math and reasoning over iterations, implicitly reflecting a learned curriculum.

## E.1 COMPLEXITY AND QUALITY OF PROMPTS OVER ITERATIONS

| Prompt Set (↓) / Metric (→) | Complexity (1-5) | Quality (1-5) |
|---|---|---|
| UltraFeedback (seed) | 2.90 | 3.18 |
| UltraFeedback-**eva**-Iter-1 | 3.84 | 3.59 |
| UltraFeedback-**eva**-Iter-2 | 3.92 | 3.63 |
| UltraFeedback-**eva**-Iter-3 | **3.98** | **3.73** |

Table 15: **eva** improves prompt quality and complexity.

As in Table 15, there is a gradual improvement of prompt complexity and quality over iterations with **eva**. We sample 10K prompts per iteration, and use the below prompts modified from Liu et al. (2023b) for the complexity and quality evaluation, with `gemini-1.5-flash` as the scorer:

```
Rank the following questions according to their quality. Your evaluation should consider the following
↪  factors: Helpfulness, Relevance, Accuracy, Depth, Creativity, and Level of detail.
Score each response from 1 to 5: 1: Poor quality, 2: Below average, 3: Average, 4: Good, 5: Excellent.
```

```
Ranking the following questions according to their difficulty and complexity. Use a fixed scoring system: 1:
↪  Very simple, 2: Simple, 3: Moderate,  4: Difficult, 5: Very difficult
```

## F VISUALIZATION ON PROMPT SELECTION METRIC

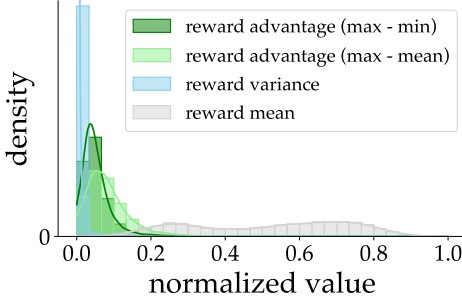
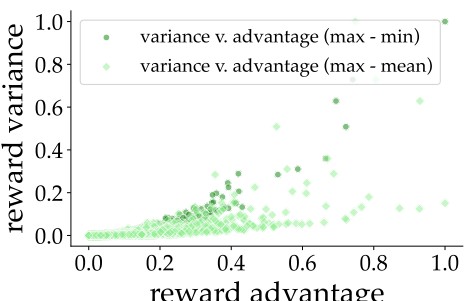

Figure 8: The probability density distributions of informativeness metrics compared in Table 3 – they show different patterns.

Figure 9: The correlation plot for reward advantage (ours) and reward variance – they are only *weakly* correlated.

In **eva**, we assign each prompt an informativeness value, which the creator will use as the weight to sample from the seed prompts for prompt synthesis. In § 4.2.1, we have shown that traditional methods like reward mean and reward variance are less effective as our advantage-based informativeness proxy. The intuition is simple: advantage/regret-based proxy aligns better with the preference optimization objective. We here further illustrate that they are statistically different from other choices:

- Figure 8: The distribution of informativeness values shows that reward variance is heavily concentrated at lower values, reward mean is more uniformly scattered, and reward advantage achieves a better balance, providing a broader yet also focused sampling range.
- Figure 9: The *weak correlation* between reward variance and reward advantage shows that variance *cannot* serve as a substitute for advantage as a proxy for informativeness.

We have discussed the contrastive curriculum hypothesis in § 3.4 to support using reward advantage in the sense that the induced samples tend to decrease the loss the most in the contrastive optimization. Furthermore, assuming the optimization algorithm can converge to the *more optimal* responses, neither reward mean nor variance directly capture the learning potential of such responses – one may easily construct cases with identical variance yet differ much in reward range – thus variance fails to distinguish such scenarios. By contrast, reward advantage estimate inherently captures the relative improvement towards better response, and is sensitive to differences in reward range; variants of advantage estimate are commonly used in literature, and we discuss underlying principles in § G.

## G    EXTENDED ILLUSTRATION ON THE METHODOLOGY

This is an extended version of § 3. In § G.1, we re-present the open-ended RLHF principle in Definition 1, and discuss the intuition under the KL regularization. In § G.2, we show heuristic approaches in open-ended learning to approximate this objective, with a focus on minimax game formulation. In § G.3, we formalize the regret objective in our RLHF setting, and discuss the regret minimization for the solver and the regret maximization for the creator.

### G.1    THE CONCEPTUAL OPEN-ENDED RLHF FORMULATION

Classical RLHF optimizes over a static prompt set:

$$\max_{\boldsymbol{\theta}} \; \mathbb{E}_{\mathbf{x} \sim \mathcal{D}, \mathbf{y} \sim \pi_{\boldsymbol{\theta}}(\cdot | \mathbf{x})} \Big[ r(\mathbf{x}, \mathbf{y}) \Big] - \mathbb{E}_{\mathbf{x} \sim \mathcal{D}} \Big[ \beta \cdot \mathbb{D}_{\mathrm{KL}} \Big[ \pi_{\boldsymbol{\theta}}(\mathbf{y} | \mathbf{x}) \parallel \pi_{\mathrm{SFT}}(\mathbf{y} | \mathbf{x}) \Big] \Big].$$

We propose to drop the static prompt set assumption, and jointly update the prompt distribution via a creator policy for Open-Ended RLHF, with the ideal objective below:

$$\max_{\boldsymbol{\phi}, \boldsymbol{\theta}} \; \mathbb{E}_{\mathbf{x} \sim \pi_{\boldsymbol{\phi}}(\cdot), \mathbf{y} \sim \pi_{\boldsymbol{\theta}}(\cdot | \mathbf{x})} \Big[ r(\mathbf{x}, \mathbf{y}) \Big] - \beta \cdot \mathbb{D}_{\mathrm{KL}} \Big[ \pi_{\boldsymbol{\phi}}(\mathbf{x}) \cdot \pi_{\boldsymbol{\theta}}(\mathbf{y} | \mathbf{x}) \parallel p_{\mathrm{ref}}(\mathbf{x}) \cdot \pi_{\mathrm{SFT}}(\mathbf{y} | \mathbf{x}) \Big].$$

This generalizes RLHF (Eq. 1). We can rewrite Eq. 7 with modified coefficients for precision:

$$\max_{\boldsymbol{\phi}, \boldsymbol{\theta}} \; \mathbb{E}_{\mathbf{x} \sim \pi_{\boldsymbol{\phi}}(\cdot)} \Big[ \underbrace{\mathbb{E}_{\mathbf{y} \sim \pi_{\boldsymbol{\theta}}(\cdot | \mathbf{x})} \Big[ r(\mathbf{x}, \mathbf{y}) \Big] - \beta_1 \cdot \mathbb{D}_{\mathrm{KL}} \big[ \pi_{\boldsymbol{\theta}}(\mathbf{y} | \mathbf{x}) \parallel \pi_{\mathrm{SFT}}(\mathbf{y} | \mathbf{x}) \big]}_{\text{solver}} \Big] - \beta_2 \cdot \underbrace{\mathbb{D}_{\mathrm{KL}} \Big[ \pi_{\boldsymbol{\phi}}(\mathbf{x}) \parallel p_{\mathrm{ref}}(\mathbf{x}) \Big]}_{\text{creator}}.$$

The newly proposed $p_{\mathrm{ref}}$ represents an *underspecified*, potentially intractable probability distribution over possible tasks in the **open-ended world** (instatiated *via* prompts) – it is *not* the initial static training prompt distribution (which is only the seed set for the creator to evolve upon); it can be seen serve as a conceptual guide to steer the prompt distribution.

To further clarify, there are two types of regularization in open-ended RLHF:

- $\mathbb{D}_{\mathrm{KL}} \big[ \pi_{\theta}(\mathbf{y} | \mathbf{x}) \parallel \pi_{\mathrm{SFT}}(\mathbf{y} | \mathbf{x}) \big]$: this is the classical regularization on the response policy, ensuring that no matter how the training distribution over prompts evolves during optimization, the response policy remained anchored to the supervised fine-tuned (SFT) policy.
  - This KL (and preference optimization) is **explicitly achieved** in plug-in algorithms (*e.g.*, DPO) in Algo. 1. We later show how it relates to **solver's regret minimization**.
- $\mathbb{D}_{\mathrm{KL}} \big[ \pi_{\phi}(\mathbf{x}) \parallel p_{\mathrm{ref}}(\mathbf{x}) \big]$: this probability matching term captures the intuition on optimizing $\pi_{\phi}(\mathbf{x})$ to approach the conceptualized $p_{\mathrm{ref}}(\mathbf{x})$, in the sense that a language model optimizes itself by adapting its training distributions with newly generated prompts for self-training to develop increasingly general capabilities, directing its learning towards informative, new tasks (Jiang, 2023), instead being constrained in a static, pre-defined set of tasks.
  - This conceptual KL is **implicitly achieved** by the creator step in the current **eva** setting by training on a *sequence of informative prompt sets*. We later show how it relates to **creator's regret maximization**. As illustrated in § 3.3.1, we start from the seed prompt set, choose those high-regret prompts and generate variations upon them by `EvolInstruct`, then mixing with a buffer of the original set to form the new training distribution at each iteration. This approach resembles classical open-ended learning in § G.2, and we hope it can serve as a small step for future works to build upon.
  - A common misunderstanding among readers may be to confuse the open-ended reference $p_{\mathrm{ref}}(\mathbf{x})$ with the initial seed prompt distribution $\mathcal{D}$, which is static. In contrast, $p_{\mathrm{ref}}(\mathbf{x})$ represents a broader space of tasks (*e.g.*, user prompts in the real wild world), as a conceptual target derived from the *underspecified distribution* (Dennis et al., 2020), *i.e.*, an environment with free parameters that control. Let's use an illustrative example with Fig. 6: the prompt distribution may be defined along several dimensions (*e.g.*, the number or complexity of coding problems); a potential creator can be designed to modify these dimensions, steering the initial $\mathcal{D}$ to new training distributions, by certain decision rules (*e.g.*, minimax regret, which offers worst-case guarantees) that forms *a sequence of informative prompts* for training.

This joint optimization objective only serves as a general principle. In the next, we discuss how existing works **implicitly achieve** the open-ended learning objective through **two-player games**.

### G.2 Approaching Open-Ended Learning by Unsupervised Environment Design

#### G.2.1 The Asymmetric Game Formulation for Unsupervised Environment Design

While we cannot directly train the agent with the intractable $p_{ref}(\mathbf{x})$ of the open-ended world, it is possible to curate a **curriculum of prompt distributions** to improve over the static distribution and support the *continual training* of the policy $\pi_{\boldsymbol{\theta}}(\cdot|\mathbf{x})$, for it to keep improving and succeed over the full task space, thus conceptually approaching $p_{ref}(\mathbf{x})$. This is often framed as an **asymmetric two-player game**.

Dennis et al. (2020) first formally define this problem as Unsupervised Environment Design (UED). The idea is that while the real-world environments are inexhaustible and hard to tract, there may exist some free parameters (*e.g.*, height and roughness in a maze) which one may control to generate new environments; UED then concerns about designing a distribution of those free parameters (*i.e.*, settings) to create new fully specified environments, that can be used to train the agents.

In this setup, one player, the **creator**, generates new environments based on some specific decision rules (see the following), while the other player, the **solver**, optimizes its policy within these training environments, and the process continues iteratively. Common **heuristic strategies** include:

- **Randomization**: environments are generated uniformly and independently of the solver's current policy. This method is simple but less effective (Tobin et al., 2017).
- **Maximin**: the creator generates environments that minimize the solver's maximum possible reward, which can often lead to unsolvable scenarios (Khirodkar and Kitani, 2018).
- **Minimax regret**: The creator targets environments that maximize the solver's *regret*, defined as the difference between the optimal return achievable and that of the solver's current policy (Beukman et al., 2024b). The regret is often conceived as the **creator's utility**.

Among them[4], the minimax regret approach presents a sweet spot where the creator can create hard yet solvable environments, and is often empirically better. The minimax regret strategy also implies that the agent's policy is trained to perform well under all levels/settings, thus enjoys a worst-case guarantee. However, while it is often straightforward for the solver to minimize the regret (*e.g.*, through direct policy optimization, as we discuss in § G.3), the optimal policy remains *unknown* during the optimization process, thus regret as the decision signal is often intractable to the creator – which requires *approximation* (as an amusing side note, this is described as the Achilles' heel of those curriculum RL methods by Parker-Holder et al. (2022)).

#### G.2.2 Approximating the Regret and Generating New Environments

In general, the **creator** design in this line of research contains two steps:

1. **identifying high-regret levels** using different (often heuristic) regret approximation;
2. **generating new environments** by making variations or retrieving from buffers on those high-regret levels.

We hereby review major works on regret approximation and environment generation as follows:

Dennis et al. (2020) propose joint training for the creator and two competing solvers.

- **Regret approximation**: here, two solver policies are trained, with the regret approximated as the **difference in their returns**. During each optimization step, one solver *maximizes* this regret, the other *minimizes* it, and the creator maximizes it.
- **Environment generation**: the system directly sample the parameter from the creator policy and use that to specify the environment.

---

[4]We have implemented variants of these in § 4.2.1, and show minimax regret is empirically better.

Jiang et al. (2021b) propose to random sampling on high-regret levels.

- **Regret approximation**: as a heuristic, the authors use *positive value loss*, which is a function of Generalized Advantage Estimate (Schulman et al., 2015) (which itself is a function of the TD error – the difference between the expected and the actual returns) as the creator's utility.
- **Environment generation**: the creator have a rolloing buffer of highest-regret levels by random searching on relevant configurations.

Jiang et al. (2021a) further propose a double-creator setting based on (Jiang et al., 2021b), where one creator is actively generating new environments, and the other is retrieving from the buffer.

Parker-Holder et al. (2022) propose to sample high-regret levels and generate new environments by making *edits* on existing ones. The regret approximation is the same as (Jiang et al., 2021b) – the positive value loss. For the environment generation, the authors suggest a general editing/mutation mechanism, where the creator chooses from high-regret levels and make small variations within an edit distance, which by heuristics will lead to the discovery of more high-regret environments. There is an additional filtering step: they do not directly train on the newly generated levels, but evaluate on those levels first, then add only the high-regret ones to the training buffer.

Note the solvers are often directly trained with PPO (Schulman et al., 2017) under the environments.

### G.3 REGRET FORMULATION FOR OPEN-ENDED RLHF

Next, we discuss the regret minimization and maximization in our setting for alignment. Specifically,

- **Regret minimization for the solver**: we avoid calculating regret and use direct policy optimization (*e.g.*, DPO) to equivalently achieve regret minimization.
- **Regret maximization for the creator**: similarly to (Jiang et al., 2021b; Parker-Holder et al., 2022), we first find an approximation of regret, then curate new environments for the solver to train on by (i) sampling from a replay buffer of existing prompts, and (ii) making variations (through `EvolInstruct` (Xu et al., 2023a)) on those high-regret prompts. Specifically, we use **advantage-based estimates of the current policy**, as summarized in Table 2.

This asymmetric two-player game serves as one potential modeling choice to implicitly achieve the open-ended RLHF principle that we proposed in Definition 1. We look forward to exploring more principled solutions in the future.

**Preliminaries.** Let $r(\cdot, \cdot)$ be an oracle reward model. The (unregularized) optimal policy is:

$$\pi^\star = \arg\max_\pi \mathbb{E}_{\mathbf{x} \sim \mathcal{D}, \mathbf{y} \sim \pi(\cdot|\mathbf{x})} \Big[ r(\mathbf{x}, \mathbf{y}) \Big].$$

We have the *optimal advantage* / the *negated regret* as:

$$A^\star(\mathbf{x}, \mathbf{y}) = r(\mathbf{x}, \mathbf{y}) - \mathbb{E}_{\mathbf{y}' \sim \pi^\star(\cdot|\mathbf{x})} \Big[ r(\mathbf{x}, \mathbf{y}') \Big]$$
$$= r(\mathbf{x}, \mathbf{y}) - V^\star(\mathbf{x}, \mathbf{y}).$$

Classical preference-based RL assumes a *reward*-based preference model, that is:

$$P\left(\mathbf{y}^+ \succeq \mathbf{y}^-\right) = \frac{\exp\big(r\left(\mathbf{x}, \mathbf{y}^+\right)\big)}{\exp\big(r\left(\mathbf{x}, \mathbf{y}^+\right)\big) + \exp\big(r\left(\mathbf{x}, \mathbf{y}^-\right)\big)}.$$

As a side note (Hejna et al., 2023), this is equivalent to the *advantage/regret*-based preference model, due to the bandit setup in RLHF:

$$P\left(\mathbf{y}^+ \succeq \mathbf{y}^-\right) = \frac{\exp\big(r\left(\mathbf{x}, \mathbf{y}^+\right) - V^\star(\mathbf{x}, \mathbf{y})\big)}{\exp\big(r\left(\mathbf{x}, \mathbf{y}^+\right) - V^\star(\mathbf{x}, \mathbf{y})\big) + \exp\big(r\left(\mathbf{x}, \mathbf{y}^-\right) - V^\star(\mathbf{x}, \mathbf{y})\big)}$$
$$= \frac{\exp\big(A^\star\left(\mathbf{x}, \mathbf{y}^+\right)\big)}{\exp\big(A^\star\left(\mathbf{x}, \mathbf{y}^+\right)\big) + \exp\big(A^\star\left(\mathbf{x}, \mathbf{y}^-\right)\big)}.$$

In our current setting, we assume there is an oracle preference model for the preference pair labeling.

**KL-regularized regret.** In the RLHF setting at fixed prompt distribution, the objective is:

$$\max_{\pi_{\boldsymbol{\theta}}} \mathbb{E}_{\mathbf{x} \sim \pi_{\phi}(\cdot), \mathbf{y} \sim \pi_{\boldsymbol{\theta}}(\cdot | \mathbf{x})} \left[ r(\mathbf{x}, \mathbf{y}) \right] - \mathbb{E}_{\mathbf{x} \sim \mathcal{D}} \left[ \beta \cdot \pi_{\phi}(\cdot)_{\mathrm{KL}} \left[ \pi_{\boldsymbol{\theta}}(\mathbf{y} | \mathbf{x}) \parallel \pi_{\mathrm{SFT}}(\mathbf{y} | \mathbf{x}) \right] \right].$$

The optimal policy of the above KL-constrained objective is:

$$\pi_{\mathrm{KL}}^{\star}(\mathbf{y} | \mathbf{x}) = \frac{1}{Z(\mathbf{x})} \pi_{\mathrm{SFT}}(\mathbf{y} | \mathbf{x}) \exp\left( \frac{1}{\beta} \cdot r(\mathbf{x}, \mathbf{y}) \right),$$

where the partition function is defined as $Z(\mathbf{x}) = \sum_{\mathbf{y}} \pi_{\mathrm{ref}}(\mathbf{y} | \mathbf{x}) \exp\left( \frac{1}{\beta} r(\mathbf{x}, \mathbf{y}) \right)$.

We can now formally define the *regret* with regard to $\pi_{\mathrm{KL}}^{\star}(\cdot | \mathbf{x})$ as:

$$\mathrm{Regret}_{\mathrm{KL}}(\mathbf{x}, \pi_{\boldsymbol{\theta}}) = \mathbb{E}_{\mathbf{y} \sim \pi_{\boldsymbol{\theta}}(\cdot | \mathbf{x})} \left[ r(\mathbf{x}, \mathbf{y}) \right] - \mathbb{E}_{\mathbf{y} \sim \pi_{\mathrm{KL}}^{\star}(\cdot | \mathbf{x})} \left[ r(\mathbf{x}, \mathbf{y}) \right]. \tag{14}$$

**Regret Minimization for the Solver.** It is rather straightforward/trivial to understand the objective of the solver $\pi_{\boldsymbol{\theta}}(\cdot | \mathbf{x})$ as regret minimization, since the goal is to align the policy $\pi_{\boldsymbol{\theta}}(\cdot | \mathbf{x})$ with the KL-optimal solution $\pi_{\mathrm{KL}}^{\star}(\cdot | \mathbf{x})$, which directly minimizes the KL-regularized regret by design. This formulation allows flexibility in the plug-in preference optimization algorithms for the solver's step in Algorithm 1, and ensures *the alignment problem is well-defined*. In practice, we use Direct Preference Optimization (DPO) and its variants, which approximate the KL-optimal solution by iteratively adjusting $\pi_{\boldsymbol{\theta}}$ to reflect preference differences.

**Regret Maximization for the Creator.** As discussed previously, while it is often trivial for the solver to minimize the regret through direct policy optimization, the optimal policy remains unknown during the optimization process, thus we cannot directly calculate the regret – we must approximate it when using it as the utility for the creator. Similarly to heuristics proposed by prior works (Jiang et al., 2021b;a; Parker-Holder et al., 2022), we use the advantage-based estimate:

$$|\widehat{\mathrm{Regret}}(\mathbf{x}, \pi_{\boldsymbol{\theta}})| \leftarrow \mathrm{info}_{\boldsymbol{\theta}}(\mathbf{x}) := r(\mathbf{x}, \mathbf{y}_{+}) - r(\mathbf{x}, \mathbf{y}_{\mathrm{baseline}}), \tag{15}$$

where

$$\mathbf{y}_{+} := \arg\max_{\mathbf{y}_i} r(\mathbf{x}, \mathbf{y}), \tag{16}$$

$$\mathbf{y}_{\mathrm{baseline}} := \arg\min_{\mathbf{y}_i} r(\mathbf{x}, \mathbf{y}) \text{ or } \mathbf{y}_{\mathrm{baseline}} := \mathrm{avg}_{\mathbf{y}_i} r(\mathbf{x}, \mathbf{y}), \tag{17}$$

and $\{\mathbf{y}_i\}_{i=1}$ is a set of responses sampled from $\pi_{\boldsymbol{\theta}}(\cdot | \mathbf{x})$ and $r(\cdot, \cdot)$ is the reward oracle. We use $\arg\min_{\mathbf{y}_i} r(\mathbf{x}, \mathbf{y})$ by default due to its simplicity and efficiency (see also § 3.4 for additional interpretation) and consistent strong empirical gains we observed in vast experiments. As the policy optimizes, the proxy should approximate the true regret better. We leave exploration of other informativeness proxy designs in **eva** to future work.

For new environment generation, as illustrated in § 3.3.1, we start from the seed prompt set, choose those high-regret prompts and generate variations upon them by `EvolInstruct`, then mixing with a buffer of the original set to form the new training distribution at each iteration.

## H EXTENDED LITERATURE REVIEW FOR OPEN-ENDED LEARNING

The design of our game-theoretic framework for language model post-training is inspired from many prior works in open-ended learning. The central idea of open-ended learning is *not* to optimize for a *specific, static* distribution, but to develop an agent that can *generalize* well across *unseen, novel* environments, which are the environments that the agent has not been explicitly trained on. To achieve this, unsupervised environment design proposes to generate environments that present a curriculum of *increasing complexity* for the agent to evolve, which ensures that the agent's learning is not *narrow*, but broad enough to handle the diversity of complexity of future environments. In such curriculum, as the agent solves simpler environments, it moves on to more difficult ones, thus progressively builds more sophisticated strategies. Furthermore, by adopting a *minimax regret* framework, this approach adds a layer of robustness by minimizing the agent's performance gap in worst-case (*i.e.*, most adversarial) environments. In addition to distinctions discussed in § 5, we here list several foundational works in

this line, and encourage the LLM community to explore with more rigor and depth: Schmidhuber (1991) presents an initial investigation into open-ended learning via self-supervised curiosity-driven exploration; Wang et al. (2019) emphasize co-evolution of environments and agent policies by training a population of agents that adapt to and solve progressively complex challenges; Dennis et al. (2020) formally introduce the notion of Unsupervised Environment Design (UED), where a protagonist and antagonist agent pair simulates regret by competing in shared environments, driving the protagonist (the main learner) to adapt to increasingly challenging scenarios; Jiang et al. (2021b) introduce Prioritized Level Replay (PLR), which uses a rolling buffer of high-regret levels to dynamically adjust the training curriculum, and selects levels with the higher learning potential; Parker-Holder et al. (2022) further propose improvements by editing previously high-regret levels; Hughes et al. (2024b) present a formal definition for open-ended system with respect to *novelty* and *learnability*, that generalizes various systems, *e.g.*, AlphaGo (Silver et al., 2016), AdA (Team et al., 2021), etc.

# I  EXTENDED LITERATURE REVIEW IN BI-LEVEL RLHF

Bi-level optimization refers to optimization problems where the cost function is defined *w.r.t.* the optimal solution to another optimization problem (Grosse, 2022). There is a recent line of works applying bi-level optimization to RLHF. While they all rely on a fixed dataset of prompts, **eva** propose to dynamically update the prompt set, as in § 1. We present a detailed comparison of **eva** with Ding et al. (2024); Shen et al. (2024); Makar-Limanov et al. (2024). We sincerely thank the anonymous reviewer for the kind references, and welcome suggestions for any other works we may have missed.

Ding et al. (2024) formulate iterative online RLHF as a bi-level optimization problem, where the upper-level represents the reward learning, and the lower-level represents the policy optimization. Leveraging reward re-parameterization tricks in Rafailov et al. (2023), Ding et al. (2024) reduces the problem to a single-level objective with regard to the policy. The differences of this work and our work lie in the prompt distribution and preference oracle: (i) **eva** features by **dynamic prompt set generation for Open-Ended RLHF**, whereas (Ding et al., 2024) remains using a static prompt set; (ii) we assume the existence of the preference oracle (as discussed in § 4), while Ding et al. (2024) consider online training of reward models and ablate on self-rewarding by the current LLM policy. Our usage of a pre-trained reward model follows from industrial practices (Team et al., 2023; 2024b), which is also commonly used by prior works in academia (Meng et al., 2024; Wu et al., 2024).

Makar-Limanov et al. (2024) provide an interesting exploration on formulating RLHF as a leader-follower game, where the language model (LM) policy is the leader and the reward model (RM) policy is the follower, and the solution is **Stackelberg equilibrium** (von Stackelberg, 1934; Rajeswaran et al., 2020), where the *leader does not likewise best respond to the follower's strategy*. Here, following the curriculum RL literature (Dennis et al., 2020; Parker-Holder et al., 2022), we seek the **Nash equilibrium** (Nash et al., 1950) between the creator for prompt generation and the solver for response generation. In the current setting of **eva**, the goal is to search for an optimal solver policy with a best supporting prompt distribution, *and* an optimal prompt distribution with a best supporting solver policy. Nevertheless, the LM-RM iterative optimization may be added on top of **eva**'s framework, and we look forward to future works exploring the leader-follower re-formulation of **eva**.

Shen et al. (2024) present a rigorous theoretical work (though it does not directly involve practical post-training of large language models). The authors propose to reduce the bi-level problem to a single-level problem with a penalty-based reformulation, and apply it in the setting of LM-RM optimization within a *fixed* environment, whereas **eva** focuces on dynamic prompt generation and practically train large language models with extensive empirical experiments conducted. We believe it would be interesting to adapt similar first-order optimization techniques to solve Open-Ended RLHF.

In summary, existing bi-level RLHF works focus on online optimization of both the RM and the LM (as the response policy), all with **fixed** prompt/state distribution. **eva** presents an orthogonal direction on **dynamic** prompt generation for Open-Ended RLHF, with an empirical algorithm which attains state-of-the-art performance with large language models on a variety of benchmarks. It is possible to incorporate the online RM training within **eva** – we have shown in § 4.2.3 that **eva** scales with quality of reward models, thus integrating online RM training may further enhance performance and mitigate potential distributional mismatch problems as we evolves for more prompts. This direction may have not been widely adopted in real-world training of language models, due to concerns about practicality (Team et al., 2023; 2024a;b; Adler et al., 2024). We look forward to future works exploring *efficient* variations unifying **eva** and existing bi-level RM-LM frameworks.

## J EXAMPLES ON PROMPTS AND MODEL GENERATIONS

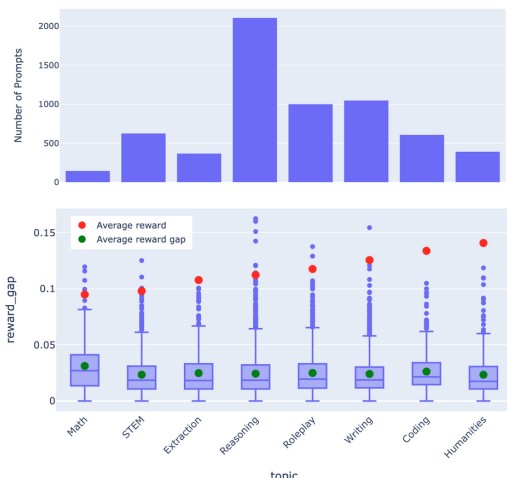

Figure 10: The initial prompt distribution of AlpacaEval by `bart-large-mnli` with 0-shot classification, which is imbalanced. For the reward distribution, the category with lowest average reward has the highest reward gap (*i.e.*, the default informativeness proxy), implying the potential to improve.

| | |
|---|---|
| initial prompt → | Write me the code for a distributed transaction manager.\nThink step by step and use pseudo code first.\nThen, define interfaces for all involved actors and entities.\nUse Rational Unified approach for this part.\n\nOnly then move on to the actual implementation, class-by-class, and method-by-method.\nMake the code be implemented in C# and follow SOLID principles. |
| evolved #1 → | Craft a suite of syntax for a distributed transaction coordinator. Start with a sequential breakdown in pseudocode format. Following that, establish the protocols for communication and interaction amongst the various participants and components, incorporating the Rational Unified Process methodology.\n\nProceed thereafter to the concrete creation, detailing each class and function. Ensure that the final C# code adheres to the precepts of SOLID and is annotated for clarification and maintainability purposes. |
| evolved #2 → | Devise a comprehensive set of directives and structures for a distributed transaction coordinator architecture. Initiate by formulating a step-by-step algorithmic decomposition in pseudocode. Subsequently, delineate the frameworks for dialog and cooperation between the numerous entities and elements, utilizing the tenets of the Rational Unified Process methodology.\n\nContinue to the actual generation of the code, meticulously detailing every class and corresponding method. Guarantee that the culminating C# code is in strict compliance with SOLID principles and is supplemented with descriptive commentary to enhance future clarity and upkeep, while also validating the code against a set of unit tests to ensure robust functionality. |
| evolved #3 → | Commence by constructing an algorithm in pseudocode that meticulously breaks down the sequential stages for a distributed transaction coordinator system. Progress by outlining the communication protocols and collaborative mechanisms among the myriad components, adhering to the Rational Unified Process approach.\n\nAdvance to crafting the code in C#, ensuring each class and method is elaborated with precision, aligning with SOLID design principles. Integrate thorough comments for heightened maintainability and conduct comprehensive unit testing for assured operational integrity. |
| evolved #4 → | Design a detailed flowchart that represents the intricate steps of an orchestration engine for managing microservices in a cloud-native environment. Begin by detailing the service discovery, load balancing, and fault tolerance strategies, while incorporating the DevOps philosophy.\n\nProceed by developing the implementation in Java, focusing on clean, modular code following object-oriented programming best practices. Accompany your code with clear documentation for easy understanding and perform rigorous integration testing to guarantee robustness. |

Table 16: Examples of evolved prompts from AlpacaEval. We suggest practitioners to include generated responses with oracle rewards and self-critique in the context for better prompt evolving, which is helpful for mitigating the effect of potential self-consuming loops in synthetic data training.

| **1st-turn prompt:** When a number is divided by 10, the remainder is 4. What is the remainder when twice the number is divided by 4? | |
|---|---|
| **Answer from gemma-2-9b-it-dpo** | **Answer from gemma-2-9b-it-dpo-eva** |
| Here's how to solve this problem: 1. Understand the first statement: * "When a number is divided by 10, the remainder is 4" means the number can be represented as 10*k + 4, where k is an integer. 2. Double the number: * Twice the number is 2*(10*k + 4) = 20*k + 8 3. Divide by 4: * (20*k + 8) divided by 4 is (20*k)/4 + 8/4 = 5*k + 2 4. Find the remainder: * Since 5*k is a multiple of 4, the remainder when twice the number is divided by 4 is 2. Let me know if you'd like to try another example! | Here's how to solve this problem: 1. Understand the first statement: – "When a number is divided by 10, the remainder is 4" means the number can be represented as 10*k + 4, where k is an integer. 2. Double the number: – Twice the number is 2*(10*k + 4) = 20*k + 8 3. Divide by 4: – When (20*k + 8) is divided by 4, the 20*k term is perfectly divisible by 4. – The remainder will be the same as the remainder when 8 is divided by 4, which is 0. Therefore, the remainder when twice the number is divided by 4 is 0. |
| **2nd-turn prompt:** What about when twice the number is divided by 5? | |
| **Answer from gemma-2-9b-it-dpo** | **Answer from gemma-2-9b-it-dpo-eva** |
| Here's how to solve that: 1. We know: Twice the number is 20*k + 8 (from the previous calculation). 2. Divide by 5: (20*k + 8) divided by 5 is (20*k)/5 + 8/5 = 4*k + 1.6 3. Find the remainder: Since 4*k is a multiple of 5, the remainder when twice the number is divided by 5 is 1. Let me know if you have any more number theory puzzles! | Here's how to solve that: 1. Twice the number: We know twice the number is 20*k + 8 (from the previous steps). 2. Divide by 5: * 20*k is divisible by 5. * 8 divided by 5 leaves a remainder of 3. Therefore, the remainder when twice the number is divided by 5 is 3. |
| **Judgment** (by gpt-4) | |
| **Judgment for gemma-2-9b-it-dpo** | **Judgment for gemma-2-9b-it-dpo-eva** |
| The assistant's response is incorrect. The assistant incorrectly calculated the remainder when twice the number is divided by 5. The assistant calculated the decimal part of the division operation, not the remainder. The correct calculation should be (20n + 8) % 5 = (20n % 5) + (8 % 5) = 0 + 3 = 3. Therefore, the remainder when twice the number is divided by 5 is 3, not 1 as the assistant stated.  The corrected response would be: 1. We know: Twice the number is 20*k + 8 (from the previous calculation). 2. Divide by 5: (20*k + 8) divided by 5 gives us a remainder of (20*k % 5) + (8 % 5) = 0 + 3 = 3. 3. Find the remainder: The remainder when twice the number is divided by 5 is 3.  Rating: [[3]], "score": 3, "turn": 2, "tstamp": 1722737011.7768948 | The assistant's answer is correct and matches the reference answer. The assistant correctly explains that twice the number is 20*k + 8 and that when this is divided by 5, the remainder is 3. The assistant's explanation is clear and easy to understand, making it helpful for the user. There are no mistakes to correct.  Rating: [[10]], "score": 10, "turn": 2, "tstamp": 1722846518.5543818 |

Table 17: Comparison of model generations on examples from MT-Bench.

**1st-turn prompt:** Imagine you are writing a blog post comparing two popular smartphone models. Develop an outline for the blog post, including key points and subheadings to effectively compare and contrast the features, performance, and user experience of the two models. Please answer in fewer than 200 words.

**Answer from gemma-2-9b-it-dpo**

## Smartphone Showdown: [Model A] vs. [Model B]\n\n**I. Introduction**\n* Briefly introduce the two smartphone models being compared, highlighting their popularity and target audience.\n* State the purpose of the comparison: to help readers decide which phone best suits their needs.\n\n**II. Design and Display**\n* Physical dimensions, weight, and build quality comparison.\n* Screen size, resolution, and display technology (e.g., OLED, LCD).\n* Design aesthetics and user preferences.\n\n**III. Performance and Specs**\n* Processor, RAM, and storage comparison.\n* Benchmark scores and real-world performance analysis.\n* Battery life and charging capabilities.\n\n**IV. Camera Capabilities**\n* Rear and front camera specs (megapixels, aperture, features).\n* Image quality comparison in various lighting conditions.\n* Video recording capabilities.\n\n**V. Software and User Experience**\n* Operating system and user interface comparison.\n* App ecosystem and availability.\n* Software features and customization options.\n\n**VI. Price and Value**\n* Compare the price points of both models.\n* Analyze the value proposition based on features and performance.\n\n**VII. Conclusion**\n* Summarize the key differences and similarities.\n* Recommend the best phone for specific user profiles (e.g., budget-conscious, photography enthusiasts, gamers).

**Answer from gemma-2-9b-it-dpo-eva**

## iPhone 14 Pro vs. Samsung Galaxy S23: Which Flagship Reigns Supreme?\n\n**I. Introduction**\n* Briefly introduce both smartphones and their position in the market.\n* State the purpose of the comparison: to help readers decide which phone best suits their needs.\n\n**II. Design and Display**\n* Physical dimensions, weight, and build quality comparison.\n* Screen size, resolution, refresh rate, and display technology (OLED vs. Super AMOLED).\n* Design aesthetics and unique features (e.g., notch vs. punch-hole, color options).\n\n**III. Performance and Camera**\n* Processor comparison (Apple A16 Bionic vs. Snapdragon 8 Gen 2).\n* RAM and storage options.\n* Camera specs: megapixels, lens types, aperture, and special features (e.g., telephoto, macro).\n* Image quality comparison based on sample photos and user reviews.\n\n**IV. Software and User Experience**\n* Operating systems: iOS 16 vs. Android 13.\n* User interface comparison: intuitiveness, customization options, and app ecosystems.\n* Special features: Face ID vs. in-display fingerprint sensor, voice assistants.\n\n**V. Battery Life and Charging**\n* Battery capacity and real-world usage comparison.\n* Fast charging speeds and wireless charging capabilities.\n\n**VI. Price and Value**\n* Compare starting prices and different storage configurations.\n* Assess the overall value for money based on features and performance.\n\n**VII. Conclusion**\n* Summarize the key differences and strengths of each phone.\n* Recommend the best phone for specific user types (e.g., photography enthusiasts, gamers, budget-conscious consumers).

**2nd-turn prompt:** Take your previous response and rephrase it as a limerick.

**Answer from gemma-2-9b-it-dpo**

Two phones, a showdown so grand,\n[Model A] and [Model B], hand in hand.\nDesign, specs, and price,\nWe'll analyze nice,\nTo help you choose, understand?\n\nLet me know if you'd like me to try filling in the bracketed parts with specific model names!

**Answer from gemma-2-9b-it-dpo-eva**

\nTwo phones, a Pro and a Galaxy,\nFor flagships, they truly are spacy.\nWith cameras and screens,\nAnd processors keen,\nWhich one wins? Read on, and be savvy! \nLet me know if you'd like another limerick!

**Judgment** (by gpt-4)

**Judgment for gemma-2-9b-it-dpo**

The assistant's response is creative and relevant to the user's request. The user asked the assistant to rephrase the previous response, which was a detailed outline for a blog post, into a limerick. The assistant successfully did so, maintaining the essence of the original response in a limerick format. The assistant also offered to further customize the limerick by filling in the bracketed parts with specific model names, showing a willingness to provide additional detail if needed. However, the limerick does not fully capture the depth and detail of the original response, which is understandable given the constraints of the limerick format.

Rating: [[8]], "score": 8, "turn": 2, "tstamp": 1722737024.0327556

**Judgment for gemma-2-9b-it-dpo-eva**

The assistant's response is creative and relevant to the user's request. The assistant successfully rephrased the previous response, which was a detailed outline for a blog post comparing two smartphone models, into a limerick. The limerick is playful and engaging, and it maintains the essence of the original response, which is to compare the two phones and help the reader make an informed decision. The assistant also offers to create another limerick if the user wants, showing a willingness to continue the interaction.

Rating: [[9]], "score": 9, "turn": 2, "tstamp": 1722846542.9437995

Table 18: (Continued) comparison of model generations on examples from MT-Bench.

**1st-turn prompt:** Each problem consists of three statements. Based on the first two statements, the third statement may be true, false, or uncertain.\n1. Oranges cost more than apples.\n2. Oranges cost less than bananas.\n3. Bananas cost more than apples and bananas cost more than orange.\nIf the first two statements are true, then the third statement is?

**Answer from gemma-2-9b-it-dpo**

Here's how to break down the logic:\n\n* **Statement 1:** Oranges > Apples (Oranges cost more than apples)\n* **Statement 2:** Oranges < Bananas (Oranges cost less than bananas)\n\nThis means the price order is: Bananas > Oranges > Apples\n\n* **Statement 3:** Bananas > Apples AND Bananas > Oranges\n\nStatement 3 is **partially true**.\n\n* The part "Bananas > Apples" is true based on our price order.\n* The part "Bananas > Oranges" is also true based on our price order.\n\nLet me know if you'd like to try another one!

**Answer from gemma-2-9b-it-dpo-eva**

Here's how to break down the logic:\n\n* **Statement 1:** Oranges > Apples (Oranges cost more than apples)\n* **Statement 2:** Oranges < Bananas (Oranges cost less than bananas)\n\nThese statements tell us that bananas are the most expensive, then oranges, and apples are the least expensive.\n\n* **Statement 3:** Bananas > Apples AND Bananas > Oranges\n\nSince bananas are indeed more expensive than both apples and oranges, statement 3 is **true**.\n\nLet me know if you'd like to try another one!

**2nd-turn prompt:** If the third statement is true. Is the first statement true, false, or uncertain? Please explain.

**Answer from gemma-2-9b-it-dpo**

If the third statement is true, the first statement is **false**.\n\nHere's why:\n\n* **Statement 3:** States that bananas cost more than both apples AND oranges.\n\nThis means the price order is: Bananas > Oranges > Apples.\n\n* **Statement 1:** States that oranges cost more than apples.\n\nIf bananas are the most expensive, and oranges are second, then oranges cannot be more expensive than apples.\n\nLet me know if you have any other logic puzzles!

**Answer from gemma-2-9b-it-dpo-eva**

If the third statement is true, then the first statement is **uncertain**.\nHere's why:\n\n* **Statement 3:** Bananas > Apples AND Bananas > Oranges\n\nThis tells us bananas are the most expensive, but it doesn't explicitly state the relative prices of apples and oranges.\n\n* **Statement 1:** Oranges > Apples\n\nThis statement claims oranges are more expensive than apples.\n\nStatement 3 doesn't contradict statement 1, but it also doesn't confirm it. It's possible that oranges *are* more expensive than apples, but statement 3 alone doesn't give us enough information to be sure.

**Judgment** (by gpt-4)

**Judgment for gemma-2-9b-it-dpo**

The assistant's response is incorrect. The assistant states that if the third statement is true, the first statement is false. However, the third statement only establishes that bananas cost more than both apples and oranges. It does not provide information about the relative costs of apples and oranges. Therefore, the first statement could be true (if oranges cost more than apples), false (if oranges cost less than apples), or uncertain (if oranges and apples cost the same). The assistant's explanation is flawed because it incorrectly assumes that the order of costs must be Bananas > Oranges > Apples, when in fact the order could also be Bananas > Apples = Oranges or Bananas > Apples > Oranges.\n\nCorrected Assistant's Answer: If the third statement is true, then bananas cost more than apples and bananas cost more than oranges. This means that oranges could still cost more than apples, making the first statement true. However, it's also possible that oranges and apples have the same cost, which would make the first statement false. Therefore, based on the third statement alone, the first statement is uncertain.

Rating: [[2]], "score": 2, "turn": 2, "tstamp": 1722737017.2808955

**Judgment for gemma-2-9b-it-dpo-eva**

The assistant's response is correct and helpful. The assistant explains that the first statement is uncertain if the third statement is true, which aligns with the reference answer. The assistant provides a clear explanation of why this is the case, stating that the third statement doesn't contradict the first statement, but it also doesn't confirm it. Therefore, the assistant's response is accurate and informative.\nThe assistant's response: "If the third statement is true, then the first statement is **uncertain**. Here's why: Statement 3: Bananas > Apples AND Bananas > Oranges. This tells us bananas are the most expensive, but it doesn't explicitly state the relative prices of apples and oranges. Statement 1: Oranges > Apples. This statement claims oranges are more expensive than apples. Statement 3 doesn't contradict statement 1, but it also doesn't confirm it. It's possible that oranges *are* more expensive than apples, but statement 3 alone doesn't give us enough information to be sure.

Rating: [[10]], "score": 10, "turn": 2, "tstamp": 1722846527.2339694

Table 19: (Continued) comparison of model generations on examples from MT-Bench.

**1st-turn prompt:** Implement a function to find the median of two sorted arrays of different sizes with O(1) space complexity and O(n) time complexity.

**Answer from gemma-2-9b-it-dpo**

python\n\def find_median_sorted_arrays(nums1, nums2):\n """\n Finds the median of two sorted arrays of different sizes.\n\n Args:\n nums1: The first sorted array.\n nums2: The second sorted array.\n Returns:\n The median of the two sorted arrays.\n """\n\n m, n = len(nums1), len(nums2)\n if m > n:\n nums1, nums2, m, n = nums2, nums1, n, m # Ensure m <= n\n\n imin, imax, half_len = 0, m, (m + n + 1) // 2\n\n while imin <= imax:\n i = (imin + imax) // 2\n j = half_len - i\n if i < m and nums2[j - 1] > nums1[i]:\n imin = i + 1 # i is too small, increase it\n elif i > 0 and nums1[i - 1] > nums2[j]:\n imax = i - 1 # i is too big, decrease it\n else:\n # i is perfect\n if i == 0:\n max_left = nums2[j - 1]\n elif j == 0:\n max_left = nums1[i - 1]\n else:\n max_left = max(nums1[i - 1], nums2[j - 1])\n\n if (m + n) % 2 == 1:\n return max_left\n\n if i == m:\n min_right = nums2[j]\n elif j == n:\n min_right = nums1[i]\n else:\n min_right = min(nums1[i], nums2[j])\n\n return (max_left + min_right) / 2.0\n\n# Example usage\nnums1 = [1, 3]\nnums2 = [2]\nmedian = find_median_sorted_arrays(nums1, nums2)\nprint(f"The median of {nums1} and {nums2} is: {median}")\n\n\n**Explanation:**\n1. **Handle Unequal Lengths:**\n - If `nums1` is longer than `nums2`, swap them to ensure `nums1` is always the shorter array. This simplifies the logic.\n2. **Binary Search:**\n - The core idea is to use binary search on the shorter array (`nums1`).\n - `imin` and `imax` define the search range for the partition point `i` in `nums1`.\n - `half_len` calculates the desired total number of elements on the left side of the partition (to find the median).\n3. **Partitioning:**\n - In each iteration, `i` divides `nums1` into two parts: `nums1[0:i]` and `nums1[i:m]`.\n - `j` is calculated to divide `nums2` so that the total number of elements on the left side is `half_len`.\n4. **Adjusting the Partition:**\n - We compare elements at the boundaries of the partitions to see if `i` is in the correct position:\n - If `nums2[j - 1] > nums1[i]`, `i` is too small, so we move `imin` to `i + 1`.\n - If `nums1[i - 1] > nums2[j]`, `i` is too large, so we move `imax` to `i - 1`.\n5. **Finding the Median:**\n - Once `i` is in the correct position, we find:\n - `max_left`: The maximum element in the left partitions of both arrays.\n - `min_right`: The minimum element in the right partitions of both arrays.\n - If the total length is odd, `max_left` is the median.\n - If the total length is even, the median is the average of `max_left` and `min_right`.\n\n**Time Complexity:** O(log(min(m, n))) because we perform binary search on the shorter array.\n\n**Space Complexity:** O(1) as we only use a constant amount of extra space for variables.

**Answer from gemma-2-9b-it-dpo-eva**

python\n\def find_median_sorted_arrays(nums1, nums2):\n """\n Finds the median of two sorted arrays of different sizes.\n\n Args:\n nums1: The first sorted array.\n nums2: The second sorted array.\n\n Returns:\n The median of the two sorted arrays.\n """\n\n m, n = len(nums1), len(nums2)\n if m > n:\n nums1, nums2, m, n = nums2, nums1, n, m # Ensure m <= n\n\n imin, imax, half_len = 0, m, (m + n + 1) // 2\n\n while imin <= imax:\n i = (imin + imax) // 2\n j = half_len - i\n if i < m and nums2[j - 1] > nums1[i]:\n imin = i + 1 # i is too small, increase it\n elif i > 0 and nums1[i - 1] > nums2[j]:\n imax = i - 1 # i is too big, decrease it\n else:\n # i is perfect\n if i == 0:\n max_left = nums2[j - 1]\n elif j == 0:\n max_left = nums1[i - 1]\n else:\n max_left = max(nums1[i - 1], nums2[j - 1])\n\n if (m + n) % 2 == 1:\n return max_left\n\n if i == m:\n min_right = nums2[j]\n elif j == n:\n min_right = nums1[i]\n else:\n min_right = min(nums1[i], nums2[j])\n\n return (max_left + min_right) / 2.0\n\n# Example usage\nnums1 = [1, 3]\nnums2 = [2]\nmedian = find_median_sorted_arrays(nums1, nums2)\nprint(f"The median of {nums1} and {nums2} is: {median}")\n\n\n**Explanation:**\n1. **Handle Unequal Lengths:**\n - If `nums1` is longer than `nums2`, swap them to ensure `nums1` is always the shorter array. This simplifies the logic.\n2. **Binary Search:**\n - The core idea is to use binary search on the shorter array (`nums1`).\n - `imin` and `imax` define the search range for the partition point `i` in `nums1`.\n - `half_len` calculates the desired total number of elements on the left side of the partition (to find the median).\n3. **Partitioning:**\n - In each iteration, `i` divides `nums1`, and `j` divides `nums2` such that the total number of elements on the left side is `half_len`.\n4. **Adjusting the Search Range:**\n - If `nums2[j - 1] > nums1[i]`, it means `i` is too small, so we need to move the partition point in `nums1` to the right (`imin = i + 1`).\n - If `nums1[i - 1] > nums2[j]`, `i` is too large, so we move it to the left (`imax = i - 1`).\n5. **Finding the Median:**\n - When the correct partition is found (`i` is in the right position), we find:\n - `max_left`: The maximum element on the left side of the partition.\n - `min_right`: The minimum element on the right side of the partition.\n - If the total number of elements is odd, `max_left` is the median.\n - If even, the median is the average of `max_left` and `min_right`.\n\n**Time Complexity:** O(log(min(m, n))) because we are performing binary search on the shorter array.\n\n**Space Complexity:** O(1) as we are only using a constant amount of extra space for variables.

**Judgment** (by gpt-4)

**Judgment for gemma-2-9b-it-dpo**

The assistant's answer is correct and provides a detailed explanation of the code. The assistant correctly states that the time complexity is O(log(min(m, n))) and the space complexity is O(1), which aligns with the reference answer. The assistant also provides an example usage of the function, which is helpful for understanding how to use it. The assistant's code is a different implementation than the reference answer, but it achieves the same result and meets the same complexity requirements.

However, the assistant's answer does not address the user's original question about achieving O(n) time complexity. The assistant should have clarified that it's not possible to find the median of two sorted arrays of different sizes with O(1) space complexity and O(n) time complexity, as stated in the reference answer.

Rating: [[8]], "score": 8, "turn": 1, "tstamp": 1722737514.2011044

**Judgment for gemma-2-9b-it-dpo-eva**

The assistant's answer is correct and provides a detailed explanation of the code. The assistant correctly states that the time complexity is O(log(min(m, n))) and the space complexity is O(1), which aligns with the reference answer. The assistant also provides a step-by-step explanation of the code, which is very helpful for understanding the logic behind the solution. The assistant's code is also correct and will correctly find the median of two sorted arrays of different sizes. Therefore, there are no corrections needed for the assistant's response.

Rating: [[10]], "score": 10, "turn": 1, "tstamp": 1722846533.772981

Table 20: (Continued) comparison of model generations on examples from MT-Bench.

**1st-turn prompt:**      You are given two sorted lists of size m and n.  Implement a function to find the kth smallest element in the union of the two lists with linear complexity.

**Answer from gemma-2-9b-it-dpo**

```python
\n\def find_kth_smallest(list1, list2, k):\n """\n Finds the kth smallest element in the union of two sorted lists.\n\n Args:\n  list1: The first sorted list.\n  list2: The second sorted list.\n  k: The desired kth smallest element.\n\n Returns:\n The kth smallest element in the union of the two lists.\n """\n\n  m, n = len(list1), len(list2)\n  i, j = 0, 0\n  min_heap = [] \n # Build a min-heap with the first elements of both lists\n  for x in range(min(m, n)):\n    heapq.heappush(min_heap, (list1[x], x, 0)) # (value, list_index, position_in_list)\n    heapq.heappush(min_heap, (list2[x], x, 1))\n\n # Extract k smallest elements from the heap\n  for _ in range(k - 1):\n    _, _, _ = heapq.heappop(min_heap)\n\n # The top element of the heap is the kth smallest\n  return heapq.heappop(min_heap)[0]\n\n# Example usage:\nlist1 = [2, 5, 8, 12]\nlist2 = [1, 3, 7, 9, 10]\nk = 5\nkth_smallest = find_kth_smallest(list1, list2, k)\nprint(f"The {k}th smallest element is: {kth_smallest}")
```

\n\n\n1. **Initialization:**\n  – 'm' and 'n' store the lengths of the input lists.\n  – 'i' and 'j' are pointers to track the current position in each list.\n  – 'min_heap' is a min-heap used to efficiently store and retrieve the smallest elements encountered so far.\n\n2. **Building the Min-Heap:**\n  – The code iterates through the first 'min(m, n)' elements of both lists.\n  – For each element, a tuple '(value, list_index, position_in_list)' is pushed onto the 'min_heap'.  This tuple helps us keep track of the element's value, the list it belongs to, and its original position.\n\n3. **Extracting k-1 Smallest Elements:**\n  – The 'for' loop runs 'k – 1' times, effectively extracting the 'k-1' smallest elements from the 'min_heap'.\n\n4. **Finding the kth Smallest:**\n  – After the loop, the top element of the 'min_heap' contains the 'k'th smallest element.\n  – 'heapq.heappop(min_heap)' removes and returns this element.\n\n**Time Complexity:**\n- Building the min-heap: O(m + n)\n- Extracting k-1 elements: O(k * log(k))\n- Overall:  **O(m + n + k * log(k))**, which is approximately **O(m + n)** since 'k' is usually much smaller than 'm + n'.\n\n\n\nLet me know if you have any other questions!

**Answer from gemma-2-9b-it-dpo-eva**

```python
\n\def find_kth_smallest(list1, list2, k):\n """\n Finds the kth smallest element in the union of two sorted lists.\n\n Args:\n  list1: The first sorted list.\n  list2: The second sorted list.\n  k: The desired rank of the smallest element.\n\n Returns:\n  The kth smallest element in the union of the lists.\n """\n\n  i = 0 # Index for list1\n  j = 0 # Index for list2\n  merged = [] # Store the merged elements\n\n  while i < len(list1) and j < len(list2):\n    if list1[i] <= list2[j]:\n      merged.append(list1[i])\n      i += 1\n    else:\n      merged.append(list2[j])\n      j += 1\n\n # Add any remaining elements from list1\n  while i < len(list1):\n    merged.append(list1[i])\n    i += 1\n\n # Add any remaining elements from list2\n  while j < len(list2):\n    merged.append(list2[j])\n    j += 1\n\n  return merged[k - 1] # Return the kth element (index k-1)\n\n# Example usage:\nlist1 = [2, 5, 8, 12]\nlist2 = [1, 3, 7, 9, 10]\nk = 5\nkth_smallest = find_kth_smallest(list1, list2, k)\nprint(f"The {k}th smallest element is: {kth_smallest}")
```

\n\n\n**Explanation:**\n\n1. **Initialization:**\n  – 'i' and 'j' are index pointers for 'list1' and 'list2' respectively, starting at 0.\n  – 'merged' is an empty list to store the merged sorted elements.\n\n2. **Merging:**\n  – The 'while' loop continues as long as both 'i' and 'j' are within the bounds of their respective lists.\n  – In each iteration, it compares the elements at 'list1[i]' and 'list2[j]'.\n  – The smaller element is appended to 'merged', and the corresponding index ('i' or 'j') is incremented.\n\n3. **Adding Remaining Elements:**\n  – After one list is exhausted, the remaining elements from the other list are directly appended to 'merged'.\n\n4. **Returning the kth Element:**\n  – Finally, 'merged[k - 1]' is returned, which represents the kth smallest element in the union (remember that list indices start from 0).\n\n\n**Linear Complexity:**\n\n- The algorithm iterates through each element in both lists at most once during the merging process.\n- The time complexity is therefore O(m + n), which is linear in the total number of elements (m + n), making it efficient for large lists.

**Judgment** (by gpt-4)

**Judgment for gemma-2-9b-it-dpo**

The assistant's answer is incorrect.  The user asked for a solution with linear complexity, but the assistant provided a solution with a time complexity of O(m + n + k * log(k)), which is not linear.  The assistant's solution uses a min-heap, which is not necessary and adds unnecessary complexity to the solution.

Rating: [[2]], "score": 2, "turn": 1, "tstamp": 1722737030.6411152

**Judgment for gemma-2-9b-it-dpo-eva**

The assistant's answer is correct and helpful.  It provides a Python function that implements the two-pointer approach to find the kth smallest element in the union of two sorted lists.  The assistant's function is slightly different from the reference, but it achieves the same goal.  The assistant's function merges the two lists into a new list and then returns the kth element from the merged list.  The assistant also provides a detailed explanation of how the function works and confirms that the time complexity is linear, which is what the user asked for.  However, the assistant's function uses extra space to store the merged list, which is not necessary and not as efficient as the reference.

Rating: [[8]], "score": 8, "turn": 1, "tstamp": 1722846543.867601

Table 21: (Continued) comparison of model generations on examples from MT-Bench.

| | 2nd-turn prompt: Does there exist an algorithm with better time complexity? If so, implement it. |
|---|---|

**Answer from gemma-2-9b-it-dpo**

**Answer from gemma-2-9b-it-dpo-eva**

You are right to ask! While the previous algorithm has linear time complexity, we can achieve an even better solution with **O(log(min(m, n)))** time complexity using a slightly more sophisticated approach.\n\nThe key idea is to leverage the sorted nature of the lists and use a **binary search** strategy.\n\nHere's the implementation:\n\n\`\`\`python\n\def find_kth_smallest_optimized(list1, list2, k):\n """\n Finds the kth smallest element in the union of two sorted lists with O(log(min(m, n))) time complexity.\n Args:\n list1: The first sorted list.\n list2: The second sorted list.\n k: The desired rank of the smallest element.\n Returns:\n The kth smallest element in the union of the lists.\n """\n m, n = len(list1), len(list2)\n if m > n:\n list1, list2, m, n = list2, list1, n, m # Ensure list1 is the shorter one\n\n left, right = 0, m\n while left <= right:\n partitionX = (left + right) // 2\n partitionY = k - partitionX\n\n maxLeftX = list1[partitionX - 1] if partitionX > 0 else float('-inf')\n minRightX = list1[partitionX] if partitionX < m else float('inf')\n\n maxLeftY = list2[partitionY - 1] if partitionY > 0 else float('-inf')\n minRightY = list2[partitionY] if partitionY < n else float('inf')\n\n if maxLeftX <= minRightY and maxLeftY <= minRightX:\n if partitionX == 0:\n return list2[partitionY - 1]\n elif partitionY == 0:\n return list1[partitionX - 1]\n else:\n return max(list1[partitionX - 1], list2[partitionY - 1])\n elif maxLeftX > minRightY:\n right = partitionX - 1 # Too many elements on the right side of list1\n else:\n left = partitionX + 1 # Too many elements on the right side of list2\n\n**Explanation:**\n\n1. **Shorten Lists:** If `list1` is longer than `list2`, swap them to ensure `list1` is always the shorter one. This simplifies the logic.\n2. **Binary Search:**\n - `left` and `right` pointers define the search space within `list1`.\n - `partitionX` is the middle point in `list1`, and `partitionY` is calculated to divide the remaining elements in `list2` such that `partitionX + partitionY = k`.\n - `maxLeftX`, `minRightX`, `maxLeftY`, and `minRightY` represent the largest element to the left and smallest element to the right of the partitions in both lists.\n3. **Comparison and Adjustment:**\n - If the `maxLeftX` is less than or equal to `minRightY` and `maxLeftY` is less than or equal to `minRightX`, it means the kth smallest element lies within the current partitions. We return the larger of `maxLeftX` and `maxLeftY`.\n - If `maxLeftX` is greater than `minRightY`, it indicates that we need to search in the left subarray of `list1` (i.e., decrease `right`).\n - Otherwise, we search in the right subarray of `list1` (i.e., increase `left`).\n4. **Termination:** The binary search continues until `left` crosses `right`, indicating the correct partition has been found.\n\n**Time Complexity:**\n\n- The binary search divides the search space in half at each step.\n- Therefore, the time complexity is O(log(min(m, n))), which is significantly faster than linear time for large lists where one list is much smaller than the other.

You are absolutely right! There's a more efficient algorithm with a time complexity of **O(m + n)** without using a heap.\n\nHere's the implementation:\n\n\`\`\`python\n\def find_kth_smallest_optimized(list1, list2, k):\n """\n Finds the kth smallest element in the union of two sorted lists\n with O(m + n) time complexity.\n\n Args:\n list1: The first sorted list.\n list2: The second sorted list.\n k: The desired kth smallest element.\n\n Returns:\n The kth smallest element in the union of the two lists.\n """\n m, n = len(list1), len(list2)\n i, j = 0, 0\n result = []\n\n # Merge the lists while keeping track of the kth smallest\n while len(result) < k:\n if i < m and j < n:\n if list1[i] <= list2[j]:\n result.append(list1[i])\n i += 1\n else:\n result.append(list2[j])\n j += 1\n elif i < m:\n result.append(list1[i])\n i += 1\n else:\n result.append(list2[j])\n j += 1\n\n return result[k - 1]\n\n\n# Example usage:\nlist1 = [2, 5, 8, 12]\nlist2 = [1, 3, 7, 9, 10]\nk = 5\nkth_smallest = find_kth_smallest_optimized(list1, list2, k)\nprint(f"The {k}th smallest element is: {kth_smallest}")\n\n\n**Explanation:**\n\n1. **Initialization:**\n - `m` and `n` store the lengths of the lists.\n - `i` and `j` are pointers for `list1` and `list2`, respectively.\n - `result` is a list to store the merged and sorted elements.\n\n2. **Merging and Tracking:**\n - The `while` loop continues until `k` elements are in the `result` list.\n - In each iteration, it compares the current elements from `list1` and `list2`.\n - The smaller element is appended to `result`, and the corresponding pointer (`i` or `j`) is incremented.\n - If one list is exhausted, the remaining elements from the other list are appended to `result`.\n\n3. **Returning the kth Smallest:**\n - Finally, `result[k - 1]` contains the `k`th smallest element.\n\n**Time Complexity:**\n- The algorithm iterates through both lists once, resulting in a time complexity of **O(m + n)**.\n\n\nLet me know if you have any more questions!

**Judgment** (by gpt-4)

**Judgment for gemma-2-9b-it-dpo**

**Judgment for gemma-2-9b-it-dpo-eva**

The assistant's first response to the user's question is incorrect. The user asked for a function to find the kth smallest element in the union of two sorted lists with linear complexity. The assistant provided a function that uses a min-heap, which has a time complexity of O(m + n + k + log(k)). This is not linear complexity, as the user requested. The assistant's second response to the user's question is also incorrect. The user asked if there exists an algorithm with better time complexity than linear. The assistant provided a function that has a time complexity of O(m + n), which is linear, not better than linear. The assistant should have provided a function that uses a binary search approach, which has a time complexity of O(log(m) + log(n)), as the reference answer correctly does.

Rating: [[2]], "score": 2, "turn": 2, "tstamp": 1722737031.5033472

The assistant's response is correct and helpful. The assistant provided a Python function that uses a binary search approach to find the kth smallest element in the union of two sorted lists. The assistant also provided a detailed explanation of how the function works, including the time complexity of the function. The assistant's answer matches the reference answer in terms of the algorithm used and the explanation provided. The assistant also correctly identified that the binary search approach has a better time complexity than the linear approach. However, the assistant's code has a minor issue. The assistant's code does not handle the case when `k` is greater than the sum of the lengths of the two lists. In such a case, the code will raise an `IndexError`. This can be fixed by adding a check at the beginning of the function to return `None` if `k` is greater than the sum of the lengths of the two lists.

Rating: [[9]], "score": 9, "turn": 2, "tstamp": 1722846556.6828268

Table 22: (Continued) comparison of model generations on examples from MT-Bench.

## REBUTTAL TO REVIEWER AC2P

**Summary.** We sincerely thank the reviewer for the appreciation of **eva** and the constructive feedback. We have made every effort to throughly address the concerns. Specifically, we have:

- added experiments on implementing **different evolving methods** and discussed relevant strengths and weaknesses in § D.1;
- added **visualization on the learning curriculum** in § E;
- provided detailed discussion on **scaling up eva** with million-level data on larger-scale seed sets and/or inference-time scaling for synthesizing prompts.

> **Q1 (Choice of the Evolving Method)**: Could you explain more about the particular choice of evolution algorithm used in your implementation of eva and different potential strengths and weaknesses related to this choice?

**TL;DR:** We use EvolInstruct (Xu et al., 2023a) as it is among the most easy-to-implement methods. We added new experiments w/ other methods, including SelfInstruct (Wang et al., 2022), EvolQuality and EvolComplexity (Liu et al., 2023b), and show that **eva** remains to be effective in § D.1.

**Answer:** As an addition to Table 1, we have experimented with three different `evolve(·)` methods:

- **SelfInstruct** (Wang et al., 2022): Given seed prompts, variations are created based on criteria such as verb diversity and style blending (mixing interrogative and imperative styles). Unlike EvolInstruct (Xu et al., 2023a), which generates prompt variations sequentially, this approach generates independently. We follow the one-shot implementation in `self_instruct.py` of `distilabel==1.4.1` and modified the instruction on conciseness so that newly generated prompts have similar lengths compared to the seed prompts.
- **EvolQuality** and **EvolComplexity** (Liu et al., 2023b): The two methods use the same evolutionary approach (*i.e.*, sequential generation), but with slightly different meta-instructions for prompt generation, where EvolQuality asks to improve the quality (*i.e.*, helpfulness, relevance, etc) of the seed prompt and EvolComplexity asks to improve the complexity (*i.e.*, increased reasoning steps, etc) of the seed prompt. We follow the implementation in `evol_quality/utils.py` and `evol_complexity/utils.py` of `distilabel==1.4.1`.

| **Model Family** ($\rightarrow$) | GEMMA-2-9B-IT | |
|---|---|---|
| **Benchmark** ($\rightarrow$) | **Arena-Hard** | |
| **Method** ($\downarrow$) **/ Metric** ($\rightarrow$) | **WR** (%) | **avg. len** |
| $\theta_0$: SFT | 41.3 | 544 |
| $\theta_{0\rightarrow1}$: DPO | 51.6 | 651 |
| $\theta_{1\rightarrow\bar{1}}$:    **+ eva** (evolve(·) = EvolInstruct) | 60.1 | 733 |
| $\theta_{1\rightarrow\bar{1}}$:    **+ eva** (evolve(·) = EvolQuality) | 58.7 | 721 |
| $\theta_{1\rightarrow\bar{1}}$:    **+ eva** (evolve(·) = EvolComplexity) | **60.6** | 749 |
| $\theta_{1\rightarrow\bar{1}}$:    **+ eva** (evolve(·) = SelfInstruct) | 57.2 | 725 |

Table 23: Results of using different evolving methods.

**eva is effective under different evolving methods.** As shown in Table 10, our method brings strong performance gain without training with additional human prompts. Among the experimented methods, we find EvolComplexity shows better results.

We believe the main strength of such method is its **simplicity**. Viewing the evolving process as $\mathbf{x}' \leftarrow p_{\boldsymbol{\theta}}(\cdot \mid \mathbf{x}, \texttt{meta\_prompt})$, one can easily tune the meta prompt in natural language for improved performance. However, such simplicity comes at a price: (i) the main weakness is that the default method does not take **environmental feedback** into account (*e.g.*, rewards received, verbal critique on responses, etc) and relies on the pre-defined meta prompt, thus the evolving may be less directional; we encourage practitioners to consider incorporating richer feedback during

evolving (one way to formulate this is by generative optimization (Yuksekgonul et al., 2024; Cheng et al., 2024; Nie et al., 2024)); (ii) another weakness is that existing method is single-shot (*i.e.*, we evolve based on a single $\mathbf{x}$ each time), thus the **diversity** of the generation may be limited – we anticipate future works improving this with multi-shot evolving by graph-based sampling or including diversity-related rewards in generation. In this regard, the evolving process can be viewed as $\{\mathbf{x}'\}_{i=1}^{N} \leftarrow p_{\boldsymbol{\theta}}(\cdot \mid \{\mathbf{x}\}_{i=1}^{M}, \texttt{meta\_prompt}, \texttt{env\_feedback})$.

> **Q2 & Q3** (**Empirical Evidence on Learning Progress and Curriculum**): Do you see empirical evidence of your intuition about learning progress discussed in section 3.4? It seems like some of these claims are directly testable. Could you visualize the curriculum learned in your experiments with eva? It would be very nice to get an intuition for why performance improves and what the heuristic prioritizes over time.

**Answer:** We thank the reviewer for the constructive suggestions on empirically validating the intuition. We have revised the manuscript with additional visualization on potential curriculum learned in § E. In general, we observe the creator prioritizes learning in math and coding, which brings gradual improvement on benchmark performance on relevant categories over iterations. We have attached the bar plot and radar figure here for the reviewer's reference:

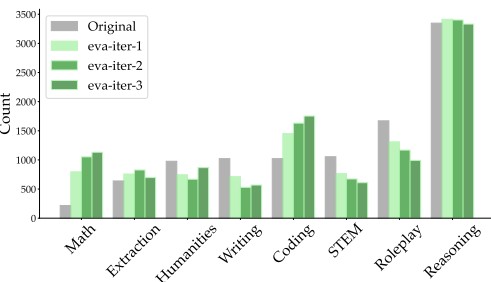

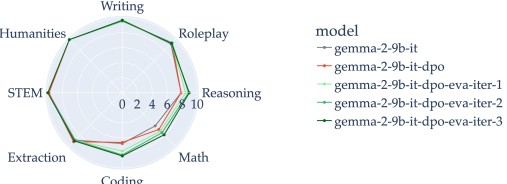

Figure 11: **Training distributions.** The prompt distribution of Table 11 for evolved prompts by zero-shot classification. **eva** creates a curriculum that prioritizes math / coding prompts.

Figure 12: **Benchmark performance.** The radar figure for ratings on MT-Bench (Zheng et al., 2023), where each category contains ten problems. **eva** prioritizes and gradually improves on coding, math and reasoning over iterations, implicitly reflecting a learned curriculum.

We further evaluate the complexity and quality of the prompt distributions. As in Table 24, there is a gradual improvement of prompt complexity and quality over iterations with **eva**.

| Prompt Set (↓) / Metric (→) | Complexity (1-5) | Quality (1-5) |
|---|---|---|
| UltraFeedback (seed) | 2.90 | 3.18 |
| UltraFeedback-**eva**-Iter-1 | 3.84 | 3.59 |
| UltraFeedback-**eva**-Iter-2 | 3.92 | 3.63 |
| UltraFeedback-**eva**-Iter-3 | **3.98** | **3.73** |

Table 24: **eva** improves prompt quality and complexity.

Implementation notes: We sample 10K prompts per iteration, and use the below prompts modified from Liu et al. (2023b) for the complexity and quality evaluation, with `gemini-1.5-flash` as the generative scorer:

```
Rank the following questions according to their quality. Your evaluation should consider the following
↪  factors: Helpfulness, Relevance, Accuracy, Depth, Creativity, and Level of detail.
Score each response from 1 to 5: 1: Poor quality, 2: Below average, 3: Average, 4: Good, 5: Excellent.
```

```
Ranking the following questions according to their difficulty and complexity. Use a fixed scoring system: 1:
↪  Very simple, 2: Simple, 3: Moderate,  4: Difficult, 5: Very difficult
```

**Q4 (Scaling):** When discussing future directions, the authors write further scaling up w/ million-level data. Can you clarify what this means? Seems like some important context is missing?

**TL;DR:** We consider (i) applying **eva** when the seed set contains million-level or more prompts; or (ii) using **eva** to robustly generate million-level or more prompts when the seed set is limited.

**Answer:** (i) The current paper uses the UltraFeedback (Cui et al., 2023) as the seed prompt set, which is a ten-thousands level dataset; in training practically useful large language models (Brown et al., 2020; Team et al., 2024a; Singh et al., 2023), the seed prompt set is usually much larger than such a level. We believe it is an interesting direction to explore the data scaling properties of **eva** on larger seed prompt sets, in combination with our on-policy variants. (ii) On the other hand, when the seed prompt set contains only limited data (this issue is particularly concerning in hard reasoning tasks like math (Yang et al., 2024)), can we still follow the data generating curriculum and synthesize million-level prompts/problems to help training, and how to robustly verify the generated prompts/problems? Beyond training-time scaling, there is also a recent trend in inference-time scaling (Snell et al., 2024), however these works only consider scaling in the $\mathcal{Y}$ space, not the $\mathcal{X}$ or the joint $(\mathcal{X}, \mathcal{Y})$ space. We believe synthetically scaling up the joint $(\mathcal{X}, \mathcal{Y})$ space to a much more larger magnitude under **eva**'s game-theoretic design presents a new direction worth investigating.

**Final remarks.** We thank the reviewer once again for spending time providing constructive feedback that helps improve **eva**. Please let us know if there is any other concerns or questions, and we are more than grateful to have the opportunity to learn from and discuss with you.

## REBUTTAL TO REVIEWER ZXTK

**Summary.** We sincerely thank the reviewer for all the constructive feedback helping improving the `eva` method. In response, we have provided:

- **experiments on more iterations** in § D.2;
- **extended discussions on the regret objective and the proxy** in § G;
- **evidence on distinction of advantage-based metrics and variance-based ones** in § F;
- **revised illustration of the method** in § 3, and **evidence on evolving prompt distributions** (to answer it cannot "cheat by selecting easier prompts") in § E.

We appreciate the chance to address the reviewer's comments and have made every effort to thoroughly address the concerns and revise our manuscript accordingly. We hope that these revisions meet the reviewer's expectations and would be grateful if the reviewer could kindly consider revise the score.

> **W1 (Running for More Iterations)**: The number of iterations in the main results is 2, with only one EVA step in each experiment, which is a little different from what the demonstration in Figure 3 shows. If the `eva` step is performed multiple times, would the results be better or worse? What is performance like when you access all data in UltraFeedback?

**TL;DR:** We added experimental result on running more iterations with more data, and `eva` remains to be effective. We have added § D.2 in the manuscript to incorporate the reviewer's suggestion.

**Rebuttal:** As an addition to § 4.2.4, we have experimented with the following settings:

- 10K prompts per iteration with 3 iterations.
- 20K prompts per iteration with 3 iterations (*i.e.*, all seed prompts are used).
- 60K prompts per iteration with 2 iterations (*i.e.*, all seed prompts are used).

Due to time constraints, we did not perform an extensive hyper-parameter search; however, we believe the results presented below sufficiently demonstrate the performance gains achieved by `eva`.

| Model Family ($\rightarrow$) | GEMMA-2-9B-IT | |
|---|---|---|
| **Benchmark** ($\rightarrow$) | **Arena-Hard** | |
| **Method** ($\downarrow$) **/ Metric** ($\rightarrow$) | **WR** (%) | **avg. len** |
| $\theta_0$: SFT | 41.3 | 544 |
| $\theta_{0\rightarrow1}$: DPO (10k) | 51.6 | 651 |
| $\theta_{1\rightarrow2}$: DPO (10k) | 59.8 | 718 |
| $\theta_{2\rightarrow3}$: DPO (10k) | 61.2 | 802 |
| $\theta_{1\rightarrow\tilde{1}}$:    + `eva` (10k) | 60.1 | 733 |
| $\theta_{\tilde{1}\rightarrow\tilde{2}}$:    + `eva` (10k) | 62.0 | 787 |
| $\theta_{\tilde{2}\rightarrow\tilde{3}}$:    + `eva` (10k) | 62.2 | 774 |

Table 25: Results of using 10k prompts per iteration.

| Model Family ($\rightarrow$) | GEMMA-2-9B-IT | |
|---|---|---|
| **Benchmark** ($\rightarrow$) | **Arena-Hard** | |
| **Method** ($\downarrow$) **/ Metric** ($\rightarrow$) | **WR** (%) | **avg. len** |
| $\theta_0$: SFT | 41.3 | 544 |
| $\theta_{0\rightarrow1}$: DPO (20k) | 53.2 | 625 |
| $\theta_{1\rightarrow2}$: DPO (20k) | 47.0 | 601 |
| $\theta_{2\rightarrow3}$: DPO (20k) | 46.8 | 564 |
| $\theta_{1\rightarrow\tilde{1}}$:    + `eva` (20k) | 59.5 | 826 |
| $\theta_{\tilde{1}\rightarrow\tilde{2}}$:    + `eva` (20k) | 60.0 | 817 |
| $\theta_{\tilde{2}\rightarrow\tilde{3}}$:    + `eva` (20k) | 61.4 | 791 |

Table 26: Results of using 20k prompts per iteration.

| Model Family ($\rightarrow$) | GEMMA-2-9B-IT | |
|---|---|---|
| Benchmark ($\rightarrow$) | Arena-Hard | |
| Method ($\downarrow$) / Metric ($\rightarrow$) | WR (%) | avg. len |
| $\theta_0$: SFT | 41.3 | 544 |
| $\theta_{0\rightarrow1}$: DPO (60k) | 58.9 | 717 |
| $\theta_{1\rightarrow\tilde{1}}$:   + eva (60k) | 59.6 | 725 |
| $\theta_{\tilde{1}\rightarrow\tilde{1}'}$:   + eva (60k) | 61.9 | 792 |

Table 27: Results of using 60k prompts per iteration.

**eva can bring robust gains with multiple iterations.** As shown in Table 25, 26, and 27, our method presents persistent performance gain over iterations, and concretely surpasses the performance by default DPO training with true human prompts.

However, there exist diminishing marginal gains in iterative off-policy training. We ground eva in the iterative (off-policy) preference alignment paradigm due to its efficiency and ease of integration. However, such paradigms inherently face diminishing returns, where performance gains decrease with successive iterations, as previously observed in (Wu et al., 2024; Setlur et al., 2024; Yuan et al., 2024; Nikishin et al., 2022). While the generative data schedule in eva mitigates these challenges and extends beyond default training with human prompts (see also §4.2.4), the gains can weaken over iterations. We summarize potential reasons as: (i) the **off-policy signal decay** – as the number of examples increases, signals from the off-policy data become weaker due to distributional shift; (ii) the **loss of plasticity**, where the agent's ability to learn good policies decreases in continuing training with more iterations (Nikishin et al., 2022); (iii) the **ability of the solver** – as we evolve more harder prompts, it is harder for the solver to produce preferred response (thus more explicit reasoning techniques may be needed); (iv) the **ability of the reward model** to correctly provide reward signals to responses and thus informativeness signals to prompts, as there may exists distributional mismatch.

Thus, we envision future work to build on eva by: (i) exploring its integration with **on-policy RLHF** (*e.g.*, instead of evolving prompts in iterations, one may evolve in batches); (ii) **enhancing solver capabilities**, such as sampling more responses during inference or leveraging meta-instructions to guide deeper reasoning; (iii) online training of RM to co-evolve with the creator and the solver.

**Bonus experiments on adding rewriter in the solver step.** This is beyond the current paper, and we present the basic idea here for practitioners to build upon eva. The motivation comes from the hypotheses derived from § D.2: as the prompts gets harder by evolving, there may be greater demands on the solver's capabilities *compared to earlier iterations*. As such, the solver may not be naively treated the same. One may address this by either inference-time scaling on responses or introducing meta-instructions to explicitly enhance the solver's reasoning.

We hereby design a proof-of-concept experiment *w.r.t* the latter by adding **rewriter** in eva's solver step. Previously, as in Algo. 1 and § 3.3.2, for each prompt $\mathbf{x}$, we generate multiple responses, and choose the best as $\mathbf{y}_+$ and the worst as $\mathbf{y}_-$ for preference optimization. Now, we add one more rewriting step that attempts to enhance $\mathbf{y}_+$ to be $\mathbf{y}'_+$, by applying a rewriting instruction (Liu et al., 2023b) that asks the solver to alter $\mathbf{y}_+$ with imporved helpfulness, relevance, reasoning depths, creativity and details while keeping the similar length. We then train with $(\mathbf{x}, \mathbf{y}'_+, \mathbf{y}_-)$ for preference optimization. Table 14 shows that adding the rewriter yields concrete performance gains over the default method, while keeping training budgets and only slightly increasing response generation cost.

| Model Family ($\rightarrow$) | GEMMA-2-9B-IT | |
|---|---|---|
| Benchmark ($\rightarrow$) | Arena-Hard | |
| Method ($\downarrow$) / Metric ($\rightarrow$) | WR (%) | avg. len |
| $\theta_0$: SFT | 41.3 | 544 |
| $\theta_{0\rightarrow1}$: DPO | 51.6 | 651 |
| $\theta_{1\rightarrow\tilde{1}}$:   + eva | **60.1** | 733 |
| $\theta_{1\rightarrow\tilde{1}}$:   + eva with rewriter | **61.9** | 741 |

Table 28: Results of adding rewriter in the solver step.

> **W2 (Connection in Minimax Regret and The Algorithm)**: The connection between the minimax regret objective and the algorithm is a somehow vague. The regret concerns the performance gap with the optimal policy. It's not reflected by the informativeness proxy.

**TL;DR:** We have added § G to address the reviewer's concern in detail. (i) In the current algorithm, the solver explicitly minimizes the regret by plug-in preference optimization algorithms (*e.g.*, DPO), while the creator implicitly maximizes the regret by first finding high-regret prompts and generate variations as new prompt distributions for training. (ii) The informativeness proxy is an advantage-based estimate of the regret; similar variants have been used in prior literature like Jiang et al. (2021b); Parker-Holder et al. (2022); as the policy optimizes, the proxy can approximate the true regret better.

**Rebuttal:** For the rebuttal to be self-contained, we extract contents from § G.3 here. We feel § G offers a better overview – it would be great if you could take some time to review § G when feasible.

**KL-regularized regret.** In the RLHF setting at fixed prompt distribution, the objective is:

$$\max_{\pi_{\boldsymbol{\theta}}} \ \mathbb{E}_{\mathbf{x} \sim \pi_\phi(\cdot), \mathbf{y} \sim \pi_{\boldsymbol{\theta}}(\cdot \mid \mathbf{x})} \left[ r(\mathbf{x}, \mathbf{y}) \right] - \mathbb{E}_{\mathbf{x} \sim \mathcal{D}} \left[ \beta \cdot \pi_\phi(\cdot)_{\mathrm{KL}} \left[ \pi_{\boldsymbol{\theta}}(\mathbf{y} \mid \mathbf{x}) \parallel \pi_{\mathrm{SFT}}(\mathbf{y} \mid \mathbf{x}) \right] \right].$$

The optimal policy of the above KL-constrained objective is:

$$\pi_{\mathrm{KL}}^{\star}(\mathbf{y} \mid \mathbf{x}) = \frac{1}{Z(\mathbf{x})} \pi_{\mathrm{SFT}}(\mathbf{y} \mid \mathbf{x}) \exp \left( \frac{1}{\beta} \cdot r(\mathbf{x}, \mathbf{y}) \right),$$

where the partition function is defined as $Z(\mathbf{x}) = \sum_{\mathbf{y}} \pi_{\mathrm{ref}}(\mathbf{y} \mid \mathbf{x}) \exp \left( \frac{1}{\beta} r(\mathbf{x}, \mathbf{y}) \right)$.

We can now formally define the *regret* with regard to $\pi_{\mathrm{KL}}^{\star}(\cdot \mid \mathbf{x})$ as:

$$\mathrm{Regret}_{\mathrm{KL}}(\mathbf{x}, \pi_{\boldsymbol{\theta}}) = \mathbb{E}_{\mathbf{y} \sim \pi_{\boldsymbol{\theta}}(\cdot \mid \mathbf{x})} \left[ r(\mathbf{x}, \mathbf{y}) \right] - \mathbb{E}_{\mathbf{y} \sim \pi_{\mathrm{KL}}^{\star}(\cdot \mid \mathbf{x})} \left[ r(\mathbf{x}, \mathbf{y}) \right]. \tag{18}$$

**Regret Minimization for the Solver.** It is rather straightforward/trivial to understand the objective of the solver $\pi_{\boldsymbol{\theta}}(\cdot \mid \mathbf{x})$ as regret minimization, since the goal is to align the policy $\pi_{\boldsymbol{\theta}}(\cdot \mid \mathbf{x})$ with the KL-optimal solution $\pi_{\mathrm{KL}}^{\star}(\cdot \mid \mathbf{x})$, which directly minimizes the KL-regularized regret by design. This formulation allows flexibility in the plug-in preference optimization algorithms for the solver's step in Algorithm 1, and ensures *the alignment problem is well-defined*. In practice, we use Direct Preference Optimization (DPO) and its variants, which approximate the KL-optimal solution by iteratively adjusting $\pi_{\boldsymbol{\theta}}$ to reflect preference differences.

**Regret Maximization for the Creator.** As discussed previously, while it is often trivial for the solver to minimize the regret through direct policy optimization, the optimal policy remains unknown during the optimization process, thus we cannot directly calculate the regret – we must approximate it when using it as the utility for the creator. Similarly to heuristics proposed by prior works (Jiang et al., 2021b;a; Parker-Holder et al., 2022), we use the advantage-based estimate:

$$|\hat{\mathrm{Regret}}(\mathbf{x}, \pi_{\boldsymbol{\theta}})| \leftarrow \mathrm{info}_{\boldsymbol{\theta}}(\mathbf{x}) := r(\mathbf{x}, \mathbf{y}_+) - r(\mathbf{x}, \mathbf{y}_{\mathrm{baseline}}), \tag{19}$$

where

$$\mathbf{y}_+ := \arg\max_{\mathbf{y}_i} r(\mathbf{x}, \mathbf{y}), \tag{20}$$

$$\mathbf{y}_{\mathrm{baseline}} := \arg\min_{\mathbf{y}_i} r(\mathbf{x}, \mathbf{y}) \text{ or } \mathbf{y}_{\mathrm{baseline}} := \mathrm{avg}_{\mathbf{y}_i} r(\mathbf{x}, \mathbf{y}), \tag{21}$$

and $\{\mathbf{y}_i\}_{i=1}$ is a set of responses sampled from $\pi_{\boldsymbol{\theta}}(\cdot \mid \mathbf{x})$ and $r(\cdot, \cdot)$ is the reward oracle. We use $\arg\min_{\mathbf{y}_i} r(\mathbf{x}, \mathbf{y})$ by default due to its simplicity and efficiency (see also § 3.4 for additional interpretation) and consistent strong empirical gains we observed in vast experiments. As the policy optmizes, the proxy should approximate the regret better. We leave exploration of other informativeness proxy designs in **eva** to future work.

For new environment generation, as illustrated in § 3.3.1, we start from the seed prompt set, choose those high-regret prompts and generate variations upon them by `EvolInstruct`, then mixing with a buffer of the original set to form the new training distribution at each iteration.

**Q1 (Advantage v.s. Variance):** The informativeness proxy seems to be similar to the variance of the rewards because they all concern the diversity of the generated responses. However, in lines 393-395, the results show using variance leads to poor performance. How to interpret this?

**Rebuttal:** To explain, (i) variance does not directly capture the **learning potential** in preference optimization, while advantage-based informativeness proxy is better aligned to the learning objective; (iii) we **empirically show** that variance and advantage are only **weakly correlated** thus will likely result in different sampling. We have added § F to incorporate the reviewer's suggestion.

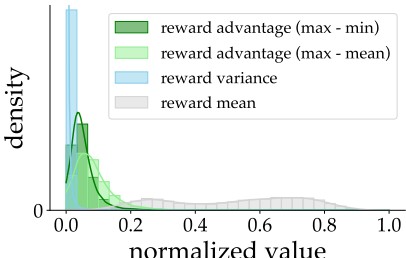
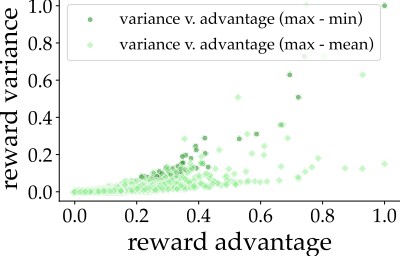

Figure 13: The probability density distributions of informativeness metrics in Table 3 – they show different patterns.

Figure 14: The correlation plot for reward advantage (ours) and reward variance – they are only *weakly* correlated.

In **eva**, we assign each prompt an informativeness value, which the creator will use as the weight to sample from the seed prompts for prompt synthesis. In § 4.2.1, we have shown that traditional methods like reward mean and reward variance are less effective as our advantage-based informativeness proxy. The intuition is simple: advantage/regret-based proxy aligns better with the preference optimization objective. We here further illustrate that they are statistically different from other choices:

- Figure 13: The distribution of informativeness values shows that reward variance is heavily concentrated at lower values, reward mean is more uniformly scattered, and reward advantage achieves a better balance, providing a broader yet also focused sampling range.
- Figure 14: The *weak correlation* between reward variance and reward advantage shows that variance *cannot* serve as a substitute for advantage as a proxy for informativeness.

We have discussed the contrastive curriculum hypothesis in § 3.4 to support using reward advantage in the sense that the induced samples tend to decrease the loss the most in the contrastive optimization. Furthermore, assuming the optimization algorithm can converge to the *more optimal* responses, neither reward mean nor variance directly capture the learning potential of such responses – one may easily construct cases with identical variance yet differ much in reward range – thus variance fails to distinguish such scenarios. By contrast, reward advantage estimate inherently captures the relative improvement towards better response, and is sensitive to differences in reward range; variants of advantage estimate are commonly used in literature, and we discuss underlying principles in § G.

**Q2 (Open-Ended Objective):** In Line 1399 (authors' note: should be around Line 880 - 886 in the main OpenReview PDF), why is this objective ideal? Optimizing the prompt distribution seems like cheating by selecting easier prompts for a higher reward.

**TL;DR:** No – the whole design (we impose the KL to the open-ended reference in principle, and its approximation by the creator through regret maximization) is to avoid cheating! We also present empirical evidence showing **eva** helps create more complex prompts.

**Answer:** To start with, we quote the conceptual objective below:

$$\max_{\phi, \theta} \mathbb{E}_{\mathbf{x} \sim \pi_\phi(\cdot), \mathbf{y} \sim \pi_\theta(\cdot | \mathbf{x})} \Big[ r(\mathbf{x}, \mathbf{y}) \Big] - \beta \cdot \mathbb{D}_{\mathrm{KL}} \Big[ \pi_\phi(\mathbf{x}) \cdot \pi_\theta(\mathbf{y} \mid \mathbf{x}) \parallel p_{\mathrm{ref}}(\mathbf{x}) \cdot \pi_{\mathrm{SFT}}(\mathbf{y} \mid \mathbf{x}) \Big].$$

Another way to express the principled objective (with refined coefficients) is:

$$\max_{\phi, \theta} \mathbb{E}_{\mathbf{x} \sim \pi_\phi(\cdot)} \Big[ \underbrace{\mathbb{E}_{\mathbf{y} \sim \pi_\theta(\cdot | \mathbf{x})} \Big[ r(\mathbf{x}, \mathbf{y}) \Big] - \beta_1 \cdot \mathbb{D}_{\mathrm{KL}} \big[ \pi_\theta(\mathbf{y} \mid \mathbf{x}) \parallel \pi_{\mathrm{SFT}}(\mathbf{y} \mid \mathbf{x}) \big]}_{\text{solver} \sim \text{``regret minimization''}} \Big] - \underbrace{\beta_2 \cdot \mathbb{D}_{\mathrm{KL}} \Big[ \pi_\phi(\mathbf{x}) \parallel p_{\mathrm{ref}}(\mathbf{x}) \Big]}_{\text{creator} \sim \text{``regret maximization'' (implicit)}} .$$

**Conceptually**, the cheating will happen when the reference distribution is narrow or wrongly defined. It is important that in our case $p_{\text{ref}}(\mathbf{x})$ represents an *underspecified*, potentially intractable probability distribution over possible tasks (instantiated *via* prompts) in the wild, as a realizable **open-ended reference** that covers the full diversity and complexity of tasks agents may encounter, *not* the initial static prompt set $\mathcal{D}$. The joint regularization towards $\pi_{\text{ref}}(\mathbf{x}, \mathbf{y})$ captures the objective for agents to generalize on alignment in $p_{\text{ref}}(\mathbf{x})$ with broader open-ended prompts, while being close to the SFT policy $\pi_{\text{SFT}}(\mathbf{y}|\mathbf{x})$. In brief, the definition of the conceptual $p_{\text{ref}}(\mathbf{x})$ and the **regularization** avoids collapsing to distributions with easier prompts.

**Practically**, we do not directly optimize this principle, rather we design a creator-solver game to implicitly and iteratively achieve this. It is important that we use *regret* as the objective and its approximation by the estimate of the *optimal reward advantage*, which avoids selecting easy prompts by design as well. See also § 3.4 on auto-curricula and learning potential for prompt selection, and § G for more connection between the objective and the algorithm.

For **empirical evidence**, as in Table 29, there is a gradual improvement of prompt complexity and quality over iterations with **eva**. We also observe the creator auto-prioritizes learning in problems like math and coding in Fig. 15, which are initially hard for it as in Fig. 16. Thus the creator also practically does not select easier prompts in the **eva** game. Details can be found in § E.

| Prompt Set (↓) / Metric (→) | Complexity (1-5) | Quality (1-5) |
|---|---|---|
| UltraFeedback (seed) | 2.90 | 3.18 |
| UltraFeedback-**eva**-Iter-1 | 3.84 | 3.59 |
| UltraFeedback-**eva**-Iter-2 | 3.92 | 3.63 |
| UltraFeedback-**eva**-Iter-3 | **3.98** | **3.73** |

Table 29: **eva** improves prompt quality and complexity.

In addition, the whole literature of curriculum RL, open-ended learning and so on are about designing the right metric for the agents to learn increasingly complex and general capabilities, and we summarize at § H for the reviewer's reference.

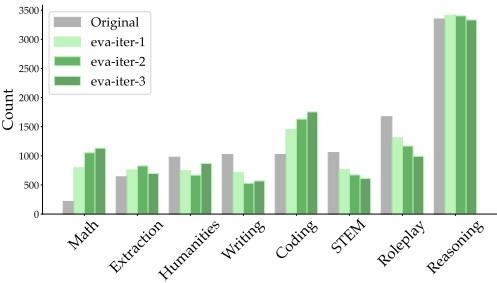

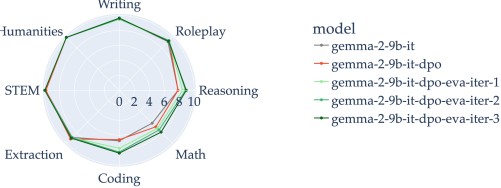

Figure 15: **Training distributions.** The prompt distribution of Table 11 for evolved prompts by zero-shot classification. **eva** creates a curriculum that prioritizes math / coding prompts.

Figure 16: **Benchmark performance.** The radar figure for ratings on MT-Bench (Zheng et al., 2023), where each category contains ten problems. **eva** prioritizes and gradually improves on coding, math and reasoning over iterations, implicitly reflecting a learned curriculum.

**Remarks.** We thank Reviewer zxTK for the constructive feedback. We have made careful efforts to address all the weaknesses and questions raised. We would be grateful if the reviewer may kindly consider again the rating for **eva**, also with regard to its strong performance gain, theeasy-to-implement method, as well as the new principle and the tractable algorithm. If there are additional concerns, we are more than happy to discuss and revise our manuscript further.

## REBUTTAL TO REVIEWER I9KX

**Summary.** We thank the reviewer for the thoughtful and detailed feedback. In response, we have:

- provided a point-by-point rebuttal fully addressing each suggested weakness and question;
- revised definitions for the regret and the informativeness proxy in § 3;
- added a detailed review on Bi-Level RLHF and open-ended learning in § I and § H;
- added detailed illustration on the method, from the principle to the asymmetric game setting, then to the regret minimization by the solver and maximization by the creator in § G;
- added new experiments on empirical gains and prompt evaluation of **eva** in § D and § E.

To avoid repetition, we reorganize the questions and group related answers into a single response.

> **Q5 (Intuition on Open-Ended RLHF)**: Can you provide intuitions behind equation 7, on the KL divergence between the joint policy for both prompt and response? Is it even tractable to estimate or approximate this KL?
>
> **W2 (Regret and KL)**: The optimization is over $\pi$ in Eq. 9 for solving the minimax regret. However, its not absolutely clear how the KL divergence plays a role here and how it is ensured that the response and prompt distributions are close to reference. Without that, the alignment problem is ill-defined. Please provide concrete justifications in theory and empirical results.
>
> **W1 (Proxy Tractability)**: How is Eq. 10 tractable and being solved? Any heuristic of sampling and approximating should result in sub-optimality which is not clear where its accounted.

**TL;DR:** We have added § G to fully address related concerns. Regarding specific concerns:

- **Q5 - Intuition.** The joint regularization towards $\pi_{\text{ref}}(\mathbf{x}, \mathbf{y})$ captures the objective for agents to generalize on alignment in $p_{\text{ref}}(\mathbf{x})$ with broader open-ended prompts, while being close to $\pi_{\text{SFT}}(\mathbf{y}|\mathbf{x})$. Note that $p_{\text{ref}}(\mathbf{x})$ is the *underspecified* open-ended reference, *not* the initial static prompt set $\mathcal{D}$. We can reformulate the principle to Eq. 8, while the KL on SFT response policy is tractable, we need to approximate the KL on the open-ended reference $p_{\text{ref}}(\mathbf{x})$. One way to achieve this heuristically is by iteratively creating a *sequence* of prompt distributions.

- **Q2 - KL.** We have revised Eq. 9 so that the regret is the difference in the reward of the current policy and the KL-optimal policy (thanks for catching this). For the solver, by design, preference optimization would be equivalent to regret minimization, thus the alignment problem remains to be correctly defined. For the creator, the distribution matching to the open-ended reference is implicitly achieved by prompt curriculum construction, and we present empirical evidence in § E to justify that prompts are evolving towards broader tasks with higher complexity.

- **W1 - Proxy.** We have revised Definition 2 for better readability. It is estimated by sampling multiple responses from the stochastic policy and calculating the reward range (or other advantage-based proxy). This approximation will result in sub-optimality for creator's regret maximization process, and we present more discussion in § G.2.

**Rebuttal:** For the rebuttal to be self-contained, we extract contents from § G here.

First, we re-present the open-ended RLHF principle, and discuss the intuition under the KL regularization. Next, we show heuristic approaches in open-ended learning to approximate this objective, with a focus on minimax game formulation. Finally, we formalize the regret objective in our RLHF setting, and discuss the regret minimization for the solver and the regret maximization for the creator.

### J.1 THE CONCEPTUAL OPEN-ENDED RLHF FORMULATION

Classical RLHF optimizes over a static prompt set:

$$\max_{\boldsymbol{\theta}} \; \mathbb{E}_{\mathbf{x}\sim\mathcal{D},\mathbf{y}\sim\pi_{\boldsymbol{\theta}}(\cdot|\mathbf{x})}\Big[\, r(\mathbf{x},\mathbf{y}) \,\Big] - \mathbb{E}_{\mathbf{x}\sim\mathcal{D}}\Big[\beta \cdot \mathbb{D}_{\text{KL}}\Big[\, \pi_{\boldsymbol{\theta}}(\mathbf{y}\mid\mathbf{x}) \,\|\, \pi_{\text{SFT}}(\mathbf{y}\mid\mathbf{x}) \,\Big]\Big].$$

We propose to drop the static prompt set assumption, and jointly update the prompt distribution via a creator policy for Open-Ended RLHF, with the ideal objective below:

$$\max_{\boldsymbol{\phi},\boldsymbol{\theta}} \; \mathbb{E}_{\mathbf{x}\sim\pi_{\boldsymbol{\phi}}(\cdot),\mathbf{y}\sim\pi_{\boldsymbol{\theta}}(\cdot|\mathbf{x})} \Big[\, r(\mathbf{x},\mathbf{y}) \,\Big] - \beta \cdot \mathbb{D}_{\mathrm{KL}} \Big[\, \pi_{\boldsymbol{\phi}}(\mathbf{x}) \cdot \pi_{\boldsymbol{\theta}}(\mathbf{y}\mid\mathbf{x}) \,\|\, p_{\mathrm{ref}}(\mathbf{x}) \cdot \pi_{\mathrm{SFT}}(\mathbf{y}\mid\mathbf{x}) \,\Big].$$

This generalizes RLHF (Eq. 1). We can rewrite Eq. 7 with modified coefficients for precision:

$$\max_{\boldsymbol{\phi},\boldsymbol{\theta}} \; \mathbb{E}_{\mathbf{x}\sim\pi_{\boldsymbol{\phi}}(\cdot)} \Big[ \underbrace{\mathbb{E}_{\mathbf{y}\sim\pi_{\boldsymbol{\theta}}(\cdot|\mathbf{x})} \Big[\, r(\mathbf{x},\mathbf{y}) \,\Big] - \beta_1 \cdot \mathbb{D}_{\mathrm{KL}} \big[\pi_{\boldsymbol{\theta}}(\mathbf{y}\mid\mathbf{x}) \,\|\, \pi_{\mathrm{SFT}}(\mathbf{y}\mid\mathbf{x})\big]}_{\text{solver}} \Big] - \beta_2 \cdot \underbrace{\mathbb{D}_{\mathrm{KL}} \Big[\, \pi_{\boldsymbol{\phi}}(\mathbf{x}) \,\|\, p_{\mathrm{ref}}(\mathbf{x}) \,\Big]}_{\text{creator}}.$$

The newly proposed $p_{\mathrm{ref}}$ represents an *underspecified*, potentially intractable probability distribution over possible tasks in the **open-ended world** (instatiated *via* prompts) – it is *not* the initial static training prompt distribution (which is only the seed set for the creator to evolve upon); it can be seen serve as a conceptual guide to steer the prompt distribution.

To further clarify, there are two types of regularization in open-ended RLHF:

- $\mathbb{D}_{\mathrm{KL}}\,[\pi_{\boldsymbol{\theta}}(\mathbf{y}\mid\mathbf{x}) \,\|\, \pi_{\mathrm{SFT}}(\mathbf{y}\mid\mathbf{x})]$: this is the classical regularization on the response policy, ensuring that no matter how the training distribution over prompts evolves during optimization, the response policy remained anchored to the supervised fine-tuned (SFT) policy.
    - This KL (and preference optimization) is **explicitly achieved** in plug-in algorithms (*e.g.*, DPO) in Algo. 1. We later show how it relates to **solver's regret minimization**.
- $\mathbb{D}_{\mathrm{KL}}\big[\, \pi_{\boldsymbol{\phi}}(\mathbf{x}) \,\|\, p_{\mathrm{ref}}(\mathbf{x}) \,\big]$: this probability matching term captures the intuition on optimizing $\pi_{\boldsymbol{\phi}}(\mathbf{x})$ to approach the conceptualized $p_{\mathrm{ref}}(\mathbf{x})$, in the sense that a language model optimizes itself by adapting its training distributions with newly generated prompts for self-training to develop increasingly general capabilities, directing its learning towards informative, new tasks (Jiang, 2023), instead being constrained in a static, pre-defined set of tasks.
    - This conceptual KL is **implicitly achieved** by the creator step in the current **eva** setting by training on a *sequence of informative prompt sets*. We later show how it relates to **creator's regret maximization**. As illustrated in § 3.3.1, we start from the seed prompt set, choose those high-regret prompts and generate variations upon them by EvolInstruct, then mixing with a buffer of the original set to form the new training distribution at each iteration. This approach resembles classical open-ended learning in § G.2, and we hope it can serve as a small step for future works to build upon.
    - A common misunderstanding among readers may be to confuse the open-ended reference $p_{\mathrm{ref}}(\mathbf{x})$ with the initial seed prompt distribution $\mathcal{D}$, which is static. In contrast, $p_{\mathrm{ref}}(\mathbf{x})$ represents a broader space of tasks (*e.g.*, user prompts in the real wild world), as a conceptual target derived from the *underspecified distribution* (Dennis et al., 2020), *i.e.*, an environment with free parameters that control. Let's use an illustrative example with Fig. 6: the prompt distribution may be defined along several dimensions (*e.g.*, the number or complexity of coding problems); a potential creator can be designed to modify these dimensions, steering the initial $\mathcal{D}$ to new training distributions, by certain decision rules (*e.g.*, minimax regret, which offers worst-case guarantees) that forms *a sequence of informative prompts* for training.

This joint optimization objective only serves as a general principle. In the next, we discuss how existing works **implicitly achieve** the open-ended learning objective through **two-player games**.

## J.2 Approaching Open-Ended Learning by Unsupervised Environment Design

### J.2.1 The Asymmetric Game Formulation for Unsupervised Environment Design

While we cannot directly train the agent with the intractable $p_{\mathrm{ref}}(\mathbf{x})$ of the open-ended world, it is possible to curate a **curriculum of prompt distributions** to improve over the static distribution and support the *continual training* of the policy $\pi_{\boldsymbol{\theta}}(\cdot|\mathbf{x})$, for it to keep improving and succeed over the full task space, thus conceptually approaching $p_{\mathrm{ref}}(\mathbf{x})$. This is often framed as an **asymmetric two-player game**.

Dennis et al. (2020) first formally define this problem as Unsupervised Environment Design (UED). The idea is that while the real-world environments are inexhaustible and hard to tract, there may exist some free parameters (*e.g.*, height and roughness in a maze) which one may control to generate new environments; UED then concerns about designing a distribution of those free parameters (*i.e.*, settings) to create new fully specified environments, that can be used to train the agents.

In this setup, one player, the **creator**, generates new environments based on some specific decision rules (see the following), while the other player, the **solver**, optimizes its policy within these training environments, and the process continues iteratively. Common **heuristic strategies** include:

- **Randomization**: environments are generated uniformly and independently of the solver's current policy. This method is simple but less effective (Tobin et al., 2017).
- **Maximin**: the creator generates environments that minimize the solver's maximum possible reward, which can often lead to unsolvable scenarios (Khirodkar and Kitani, 2018).
- **Minimax regret**: The creator targets environments that maximize the solver's *regret*, defined as the difference between the optimal return achievable and that of the solver's current policy (Beukman et al., 2024b). The regret is often conceived as the **creator's utility**.

Among them[5], the minimax regret approach presents a sweet spot where the creator can create hard yet solvable environments, and is often empirically better. The minimax regret strategy also implies that the agent's policy is trained to perform well under all levels/settings, thus enjoys a worst-case guarantee. However, while it is often straightforward for the solver to minimize the regret (*e.g.*, through direct policy optimization, as we discuss in § G.3), the optimal policy remains *unknown* during the optimization process, thus regret as the decision signal is often intractable to the creator – which requires *approximation* (as an amusing side note, this is described as the Achilles' heel of those curriculum RL methods by Parker-Holder et al. (2022)).

### J.2.2 Approximating the Regret and Generating New Environments

In general, the **creator** design in this line of research contains two steps:

1. **identifying high-regret levels** using different (often heuristic) regret approximation;
2. **generating new environments** by making variations or retrieving from buffers on those high-regret levels.

We hereby review major works on regret approximation and environment generation as follows:

Dennis et al. (2020) propose joint training for the creator and two competing solvers.

- **Regret approximation**: here, two solver policies are trained, with the regret approximated as the **difference in their returns**. During each optimization step, one solver *maximizes* this regret, the other *minimizes* it, and the creator maximizes it.
- **Environment generation**: the system directly sample the parameter from the creator policy and use that to specify the environment.

Jiang et al. (2021b) propose to random sampling on high-regret levels.

- **Regret approximation**: as a heuristic, the authors use *positive value loss*, which is a function of Generalized Advantage Estimate (Schulman et al., 2015) (which itself is a function of the TD error – the difference between the expected and the actual returns) as the creator's utility.
- **Environment generation**: the creator have a rolloing buffer of highest-regret levels by random searching on relevant configurations.

Jiang et al. (2021a) further propose a double-creator setting based on (Jiang et al., 2021b), where one creator is actively generating new environments, and the other is retrieving from the buffer.

Parker-Holder et al. (2022) propose to sample high-regret levels and generate new environments by making *edits* on existing ones. The regret approximation is the same as (Jiang et al., 2021b) – the

---

[5]We have implemented variants of these in § 4.2.1, and show minimax regret is empirically better.

positive value loss. For the environment generation, the authors suggest a general editing/mutation mechanism, where the creator chooses from high-regret levels and make small variations within an edit distance, which by heuristics will lead to the discovery of more high-regret environments. There is an additional filtering step: they do not directly train on the newly generated levels, but evaluate on those levels first, then add only the high-regret ones to the training buffer.

Note the solvers are often directly trained with PPO (Schulman et al., 2017) under the environments.

### J.3 REGRET FORMULATION FOR OPEN-ENDED RLHF

Next, we discuss the regret minimization and maximization in our setting for alignment. Specifically,

- **Regret minimization for the solver**: we avoid calculating regret and use direct policy optimization (*e.g.*, DPO) to equivalently achieve regret minimization.
- **Regret maximization for the creator**: similarly to (Jiang et al., 2021b; Parker-Holder et al., 2022), we first find an approximation of regret, then curate new environments for the solver to train on by (i) sampling from a replay buffer of existing prompts, and (ii) making variations (through `EvolInstruct` (Xu et al., 2023a)) on those high-regret prompts. Specifically, we use **advantage-based estimates of the current policy**, as summarized in Table 2.

This asymmetric two-player game serves as one potential modeling choice to implicitly achieve the open-ended RLHF principle that we proposed in Definition 1. We look forward to exploring more principled solutions in the future.

**KL-regularized regret.** In the RLHF setting at fixed prompt distribution, the objective is:

$$\max_{\pi_\theta} \mathbb{E}_{\mathbf{x} \sim \pi_\phi(\cdot), \mathbf{y} \sim \pi_\theta(\cdot | \mathbf{x})} \left[ r(\mathbf{x}, \mathbf{y}) \right] - \mathbb{E}_{\mathbf{x} \sim \mathcal{D}} \left[ \beta \cdot \pi_\phi(\cdot)_{\mathrm{KL}} \left[ \pi_\theta(\mathbf{y} | \mathbf{x}) \,\|\, \pi_{\mathrm{SFT}}(\mathbf{y} | \mathbf{x}) \right] \right].$$

The optimal policy of the above KL-constrained objective is:

$$\pi_{\mathrm{KL}}^\star(\mathbf{y} | \mathbf{x}) = \frac{1}{Z(\mathbf{x})} \pi_{\mathrm{SFT}}(\mathbf{y} | \mathbf{x}) \exp\left( \frac{1}{\beta} \cdot r(\mathbf{x}, \mathbf{y}) \right).$$

In our current setting, we assume there is an oracle preference model for the preference pair labeling. We can now formally define the *regret* with regard to $\pi_{\mathrm{KL}}^\star(\cdot | \mathbf{x})$ as:

$$\mathrm{Regret}_{\mathrm{KL}}(\mathbf{x}, \pi_\theta) = \mathbb{E}_{\mathbf{y} \sim \pi_\theta(\cdot | \mathbf{x})} \left[ r(\mathbf{x}, \mathbf{y}) \right] - \mathbb{E}_{\mathbf{y} \sim \pi_{\mathrm{KL}}^\star(\cdot | \mathbf{x})} \left[ r(\mathbf{x}, \mathbf{y}) \right]. \tag{22}$$

**Regret Minimization for the Solver.** It is rather straightforward/trivial to understand the objective of the solver $\pi_\theta(\cdot | \mathbf{x})$ as regret minimization, since the goal is to align the policy $\pi_\theta(\cdot | \mathbf{x})$ with the KL-optimal solution $\pi_{\mathrm{KL}}^\star(\cdot | \mathbf{x})$, which directly minimizes the KL-regularized regret by design. This formulation allows flexibility in the plug-in preference optimization algorithms for the solver's step in Algorithm 1, and ensures *the alignment problem is well-defined*. In practice, we use Direct Preference Optimization (DPO) and its variants, which approximate the KL-optimal solution by iteratively adjusting $\pi_\theta$ to reflect preference differences.

**Regret Maximization for the Creator.** As discussed previously, while it is often trivial for the solver to minimize the regret through direct policy optimization, the optimal policy remains unknown during the optimization process, thus we cannot directly calculate the regret – we must approximate it when using it as the utility for the creator. Similarly to heuristics proposed by prior works (Jiang et al., 2021b;a; Parker-Holder et al., 2022), we use the advantage-based estimate:

$$|\hat{\mathrm{Regret}}(\mathbf{x}, \pi_\theta)| \leftarrow \mathrm{info}_\theta(\mathbf{x}) := r(\mathbf{x}, \mathbf{y}_+) - r(\mathbf{x}, \mathbf{y}_{\mathrm{baseline}}), \tag{23}$$

where

$$\mathbf{y}_+ := \arg\max_{\mathbf{y}_i} r(\mathbf{x}, \mathbf{y}), \tag{24}$$

$$\mathbf{y}_{\mathrm{baseline}} := \arg\min_{\mathbf{y}_i} r(\mathbf{x}, \mathbf{y}) \text{ or } \mathbf{y}_{\mathrm{baseline}} := \mathrm{avg}_{\mathbf{y}_i} r(\mathbf{x}, \mathbf{y}), \tag{25}$$

and $\{\mathbf{y}_i\}_{i=1}$ is a set of responses sampled from $\pi_{\boldsymbol{\theta}}(\cdot \mid \mathbf{x})$ and $r(\cdot,\cdot)$ is the reward oracle. We use $\arg\min_{\mathbf{y}_i} r(\mathbf{x}, \mathbf{y})$ by default due to its simplicity and efficiency (see also § 3.4 for additional interpretation) and consistent strong empirical gains we observed in vast experiments. As the policy optimizes, the proxy should approximate the true regret better. We leave exploration of other informativeness proxy designs in **eva** to future work.

For new environment generation, as illustrated in § 3.3.1, we start from the seed prompt set, choose those high-regret prompts and generate variations upon them by `EvolInstruct`, then mixing with a buffer of the original set to form the new training distribution at each iteration.

> **W3 (Understanding the Iterative Algorithm)**: As described in Algorithm 1, informativeness is evaluated and a prompt subset is created based on current policy estimate and then the policy is updated based on the prompt subset. However, this causes an inter-dependence between the two which leads to nested structure, which is not clearly explained. Specifically, while computing the informativeness score for the prompts, it depends on $\theta^\star(x_{t-1})$, *i.e.*, optimal parameter for the previous distribution. Provide clear explaination on the same.

**TL;DR:** (i) We revised Algo. 1 with updated subscripts to reflect the training process – please take a look in our main paper. Given a current model checkpoint, we evaluate the prompt informativeness based on it, and evolve a new prompt set more informative to the current checkpoint, and use the new prompt set for continual training. (ii) We intend to use an iterative best-response framework to approximate equilibrium in expectation, balancing computational efficiency and practicality.

**Rebuttal:** The iterative updates in **eva**, as described in Algo. 1, are based on a best-response-to-best-response framework. Specifically, the creator updates the prompt distribution based on the solver's current policy, and the solver then optimizes its policy over the updated prompts, and the process repeats. This sequential structure approximates a Nash equilibrium in expectation over iterations, inspired by works such as Freund and Schapire (1999); Wu et al. (2024), which establish convergence to optimal policies on average through iterative optimization.

We intentionally avoid simultaneous joint optimization as it would significantly increase computational and memory overhead, making it less practical for integration into current RLHF pipelines.

> **W4 (Understanding Reward Models)**: While iterating, every new prompt distribution will require generating new response, how is the evaluation coming from which reward model? Is the ground reward available, if not please explain how the preference is obtained and how does it affect suboptimality? Also: **Q3 (RM Availability)**: What's the reward model availability? Is the true reward model available?

**TL;DR:** We assume a preference oracle provided by an **external, pre-trained reward model**, which is practically used in many real-world LLM training scenarios (Team et al., 2023).

**Rebuttal:** As discussed in the beginning of the experimental setting in § 4, we assume the availability of a pre-trained, fixed reward model. This approach is practically adopted in industry (Team et al., 2023; 2024a;b) and is also commonly used in academia works (Xu et al., 2023b; Meng et al., 2024; Wu et al., 2024). The reason is more on efficiency concerns. For example, in GEMMA-2 training, the reward model is *an order of magnitude larger* than the policy (Team et al., 2024b); it would thus be impractical or the gain may only be marginal if we update the reward model on-the-fly.

Nevertheless, it is possible to incorporate the online RM training within **eva** – we have shown in § 4.2.3 (ablation studies) that **eva** scales with quality of reward models, thus integrating online RM training may further enhance performance and address the potential distribution mismatch problem. We believe this is an interesting direction to pursue, and have listed it in § 6 (future works) on adding more players including rewarders in the self-play loop.

> **W5 (Improvement of Sub-Optimality)**: Overall, which expression/Theorem guides us in understanding the improvement of prior suboptimality is not clear? Can you please point out/highlight how the current method improves upon the prior suboptimality due to static prompt distribution?

**TL;DR:** The improvement of sub-optimality is guided by the minimax regret objective (Remark 1) through its iterative implementation. While this work does not explicitly derive suboptimality bounds, our approach has demonstrated **strong empirical gains** over the training by static distributions, as shown in § 4 (main experiments), § E (benchmark performance), and § D.2 (alignment gains over iterations).

**Rebuttal:** In general, the improvement of prior suboptimality due to static prompt distributions is guided by the minimax game outlined in Remark 1. This expression forms the basic foundation for our iterative algorithm, where the creator updates prompts to maximize informativeness (proxy for regret), and the solver minimizes regret (through direct preference optimization). This iterative process ensures the solver and creator adapt to each other, implicitly forming a curriculum and addressing sub-optimality in static prompts. We also added § G to help illustrate the intuition behind.

In general, the empirical results in § 4 (main results), § E (curriculum effect and benchmark improvement), and § D.2 (alignment gains over iterations) demonstrate that the dynamic prompt distribution improves solver performance and alignment metrics, thereby mitigating suboptimality. While the current package does not explicitly derive sub-optimality bounds (as would be typical in formal RL/bandit theory papers) and emphasizes practicality and usability as a methodology paper, we would love to learn if the reviewer has any suggestions for this as the future work.

> **W6 (Prompt Distribution):** It is extremely crucial to show the prompt distribution and demonstrate its perplexity to ensure its not generating some meaningless or irrelevant prompts, since its not very evident on the KL divergence in the prompt space and its relation with the informative measure. Please provide detailed explanation to clarify that.

**TL;DR:** We have (i) added experimental results in § E (prompt distribution visualization) and § J (prompt examples) to verify that **eva** evolves meaningful and relevant prompts with improved complexity and quality; (ii) added explanation in § G (detailed illustration of method) on the KL regularization in the prompt space and the implicit approximation by the creator.

**Rebuttal:** (This rebuttal also addresses Reviewer `ac2p`'s concerns on curriculum.)

We have revised the manuscript with additional visualization on potential curriculum learned in § E. In general, we observe the creator prioritizes learning in math and coding for the generated prompt distribution, which brings gradual improvement on benchmark performance on relevant categories over iterations. In other words, **eva** effectively shifts focus towards harder yet learnable categories. We have attached the bar plot and radar figure here for the reviewer's reference:

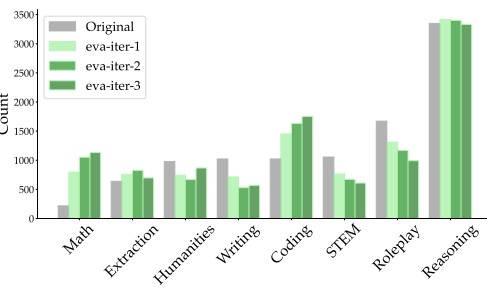
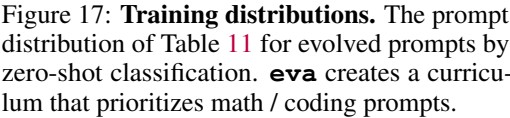

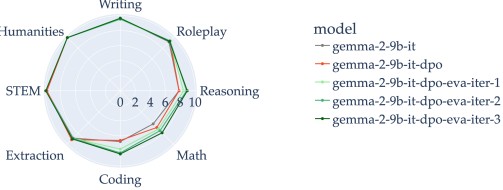

Figure 18: **Benchmark performance.** The radar figure for ratings on MT-Bench (Zheng et al., 2023), where each category contains ten problems. **eva** prioritizes and gradually improves on coding, math and reasoning over iterations, implicitly reflecting a learned curriculum.

Figure 17: **Training distributions.** The prompt distribution of Table 11 for evolved prompts by zero-shot classification. **eva** creates a curriculum that prioritizes math / coding prompts.

We have added Table 16 providing qualitative examples for evolved prompts. Also, as noted in § G, the solver maintains KL regularization during optimization, ensuring that the response distribution remain close to the reference policy; in the this work, we do not explicitly add KL regularization in the prompt distribution since we do not directly conduct parameter update for the creator (which we empirically find to bring training instability); rather, we use **meta instructions** and **buffer sampling** to constrain the prompt generations (as described in § 3.3, § A and § D.1), which is empirically very

effective, and introduces only minimal changes to existing pipeline thus can be easily applied. As noted in § 6, we look forward to future works on making the creator policy differentiable.

Regarding the relation with the informativeness measure, our current proxy is an efficient proxy among many possibilities. We have provided detailed discussions in § G to help interpret it. There could be other proxies – one interesting direction is to completely remove the dependence on the reward model and directly use model likelihoods to make the prompt selection.

We note that perplexity is not commonly used or not the most preferred measure for data quality in practical training of large language models (Team et al., 2023; Fang et al., 2024), and can be computationally heavy to measure. We have added experiments in § E.1 which we follow Liu et al. (2023b) to generatively measure the complexity and quality of prompt distributions. As in Table 30, there is a gradual improvement of prompt complexity and quality over iterations with **eva**. We hope this would address the reviewer's concerns.

| Prompt Set ($\downarrow$) / Metric ($\rightarrow$) | Complexity (1-5) | Quality (1-5) |
|---|---|---|
| UltraFeedback (seed) | 2.90 | 3.18 |
| UltraFeedback-**eva**-Iter-1 | 3.84 | 3.59 |
| UltraFeedback-**eva**-Iter-2 | 3.92 | 3.63 |
| UltraFeedback-**eva**-Iter-3 | **3.98** | **3.73** |

Table 30: **eva** improves prompt quality and complexity.

> **Q1 and Q2 (KL in the Solver Loop)**: Since equation 7, can't be directly solved, and is solved in an asymmetric fashion, then in the solver loop the KL should be over the response distribution and not joint right? How is the KL divergence w.r.t reference policy for the algorithm? Please provide detailed ablation.

**Answer:** (i) Yes, in the solver loop, the KL regularization is applied over the response distribution, not the joint distribution, as shown in Line 5 of Algo. 1. (ii) The KL divergence *w.r.t.* reference policy is determined by the plug-in solver (*e.g.*, DPO, SimPO, ...), which is orthogonal to our framework. We have added detailed explanation in § G to illustrate the whole process.

> **Q4 (Literature)**: There is a recent line of works on Stacklberg and Bilevel RLHF which deals with the entanglement in a leader-follower setting. Although not specific to updating prompt dist, but can be trivially applied. Provide a detailed comparison with the literature around that [1,2,3].

**TL;DR:** We thank the reviewer for this nice suggestion. Please see below for a detailed review on the relevant literature, highlighting the unique contribution of **eva**. We have added § I in the manuscript.

**Rebuttal:** Bi-level optimization refers to optimization problems where the cost function is defined *w.r.t.* the optimal solution to another optimization problem (Grosse, 2022). There is a recent line of works applying bi-level optimization to RLHF. While they all rely on a fixed dataset of prompts, **eva** propose to dynamically update the prompt set, as in § 1. We present a detailed comparison of **eva** with Ding et al. (2024); Shen et al. (2024); Makar-Limanov et al. (2024). We thank the anonymous reviewer for the kind references, and welcome suggestions for any other works we may have missed.

Ding et al. (2024) formulate iterative online RLHF as a bi-level optimization problem, where the upper-level represents the reward learning, and the lower-level represents the policy optimization. Leveraging reward re-parameterization tricks in Rafailov et al. (2023), Ding et al. (2024) reduces the problem to a single-level objective with regard to the policy. The differences of this work and our work lie in the prompt distribution and preference oracle: (i) **eva** features by **dynamic prompt set generation for Open-Ended RLHF**, whereas (Ding et al., 2024) remains using a static prompt set; (ii) we assume the existence of the preference oracle (as discussed in § 4), while Ding et al. (2024) consider online training of reward models and ablate on self-rewarding by the current LLM policy. Our usage of a pre-trained reward model follows from industrial practices (Team et al., 2023; 2024b), which is also commonly used by prior works in academia (Meng et al., 2024; Wu et al., 2024).

Makar-Limanov et al. (2024) provide an interesting exploration on formulating RLHF as a leader-follower game, where the language model (LM) policy is the leader and the reward model (RM) policy is the follower, and the solution is **Stackelberg equilibrium** (von Stackelberg, 1934; Rajeswaran et al., 2020), where the *leader does not likewise best respond to the follower's strategy*. Here, following the curriculum RL literature (Dennis et al., 2020; Parker-Holder et al., 2022), we seek the **Nash equilibrium** (Nash et al., 1950) between the creator for prompt generation and the solver for response generation. In the current setting of **eva**, the goal is to search for an optimal solver policy with a best supporting prompt distribution, *and* an optimal prompt distribution with a best supporting solver policy. Nevertheless, the LM-RM iterative optimization may be added on top of **eva**'s framework, and we look forward to future works exploring the leader-follower re-formulation of **eva**.

Shen et al. (2024) present a rigorous theoretical work (though it does not directly involve practical post-training of large language models). The authors propose to reduce the bi-level problem to a single-level problem with a penalty-based reformulation, and apply it in the setting of LM-RM optimization within a *fixed* environment, whereas **eva** focuces on dynamic prompt generation and practically train large language models with extensive empirical experiments conducted. We believe it would be interesting to adapt similar first-order optimization techniques to solve Open-Ended RLHF.

In summary, existing bi-level RLHF works focus on online optimization of both the RM and the LM (as the response policy), all with **fixed** prompt/state distribution. **eva** presents an orthogonal direction on **dynamic** prompt generation for Open-Ended RLHF, with an empirical algorithm which attains state-of-the-art performance with large language models on a variety of benchmarks. It is possible to incorporate the online RM training within **eva** – we have shown in § 4.2.3 that **eva** scales with quality of reward models, thus integrating online RM training may further enhance performance and mitigate potential distributional mismatch problems as we evolves for more prompts. This direction may have not been widely adopted in real-world training of language models, due to concerns about practicality (Team et al., 2023; 2024a;b; Adler et al., 2024). We look forward to future works exploring *efficient* variations unifying **eva** and existing bi-level RM-LM frameworks.

> **Final Remarks.** We thank the reviewer for the precious time and efforts on the **eva** method. We value all those opinions, and have made every effort to thoroughly address the concerns raised and revise our manuscript accordingly. Regarding the rejection, we hope the reviewer may kindly consider the points that we have summarized at the beginning of this rebuttal, on the **strong empirical alignment gain** brought by the **simple design** of **eva**, also on judging the merit of a work (*cf.,* (Castro, 2021)) *w.r.t.* the practicality and how the community may easily build on top of the principle and the method we proposed (*cf.,* (Hamming, 1986)), which we are confident are valuable to the broader alignment community. We look forward to any future discussion, and would be grateful if the reviewer may consider revising the score if the revision is satisfactory.

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
