## REBUTTAL TO REVIEWER AC2P

**Summary.** We sincerely thank the reviewer for the appreciation of **eva** and the constructive feedback. We have made every effort to throughly address the concerns. Specifically, we have:

- added experiments on implementing **different evolving methods** and discussed relevant strengths and weaknesses in § D.1;
- added **visualization on the learning curriculum** in § E;
- provided detailed discussion on **scaling up eva** with million-level data on larger-scale seed sets and/or inference-time scaling for synthesizing prompts.

> **Q1 (Choice of the Evolving Method)**: Could you explain more about the particular choice of evolution algorithm used in your implementation of eva and different potential strengths and weaknesses related to this choice?

**TL;DR:** We use EvolInstruct (Xu et al., 2023a) as it is among the most easy-to-implement methods. We added new experiments w/ other methods, including SelfInstruct (Wang et al., 2022), EvolQuality and EvolComplexity (Liu et al., 2023b), and show that **eva** remains to be effective in § D.1.

**Answer:** As an addition to Table 1, we have experimented with three different $\texttt{evolve}(\cdot)$ methods:

- **SelfInstruct** (Wang et al., 2022): Given seed prompts, variations are created based on criteria such as verb diversity and style blending (mixing interrogative and imperative styles). Unlike EvolInstruct (Xu et al., 2023a), which generates prompt variations sequentially, this approach generates independently. We follow the one-shot implementation in `self_instruct.py` of `distilabel==1.4.1` and modified the instruction on conciseness so that newly generated prompts have similar lengths compared to the seed prompts.
- **EvolQuality** and **EvolComplexity** (Liu et al., 2023b): The two methods use the same evolutionary approach (*i.e.*, sequential generation), but with slightly different meta-instructions for prompt generation, where EvolQuality asks to improve the quality (*i.e.*, helpfulness, relevance, etc) of the seed prompt and EvolComplexity asks to improve the complexity (*i.e.*, increased reasoning steps, etc) of the seed prompt. We follow the implementation in `evol_quality/utils.py` and `evol_complexity/utils.py` of `distilabel==1.4.1`.

| Model Family ($\rightarrow$) | GEMMA-2-9B-IT | |
|---|---|---|
| Benchmark ($\rightarrow$) | **Arena-Hard** | |
| Method ($\downarrow$) / Metric ($\rightarrow$) | **WR** (%) | **avg. len** |
| $\theta_0$: SFT | 41.3 | 544 |
| $\theta_{0\rightarrow1}$: DPO | 51.6 | 651 |
| $\theta_{1\rightarrow\bar{1}}$:   + **eva** ($\texttt{evolve}(\cdot) = \texttt{EvolInstruct}$) | 60.1 | 733 |
| $\theta_{1\rightarrow\bar{1}}$:   + **eva** ($\texttt{evolve}(\cdot) = \texttt{EvolQuality}$) | 58.7 | 721 |
| $\theta_{1\rightarrow\bar{1}}$:   + **eva** ($\texttt{evolve}(\cdot) = \texttt{EvolComplexity}$) | **60.6** | 749 |
| $\theta_{1\rightarrow\bar{1}}$:   + **eva** ($\texttt{evolve}(\cdot) = \texttt{