# OpenReview forum: "Evolving Alignment via Asymmetric Self-Play"
_ICLR.cc/2025/Conference — Submitted to ICLR 2025_

### Official Review · Reviewer_zXTk · 2024-11-02

**Soundness:** 3
**Presentation:** 4
**Contribution:** 3
**Rating:** 5
**Confidence:** 5

**Summary:**

This paper proposes EVA: Evolving Alignment via A symmetric Self-Play, a framework that that casts alignment as an asymmetric game between two players, a prompt creator and a policy learner. They selects prompts which induce responses with largest reward gaps, and generate similar prompts with those selected prompts. The proposed method improves the baseline by a margin.

**Strengths:**

This paper is well-written, easy-to-follow and informative; It contains thorough ablation study to show the effectiveness of the proposed method; The empirical evaluation shows much improvement.

**Weaknesses:**

**Main concerns**

1. The number of iterations in the main results is 2, with only one EVA step in each experiment, which is a little different from what the demonstration in Figure 3 shows. If the EVA step is performed multiple times, would the results be better or worse? What is performance like when you access all data in ultrafeedback?

2. The connection between the minimax regret objective and the algorithm is a somehow vague. The regret concerns the performance gap with the optimal policy. It's not reflected by the informativeness proxy.

**Minor issues**

Line 278: type wrost -> worst

**Questions:**

The informativeness proxy seems to be similar to the variance of the rewards because they all concern the diversity of the generated responses. However, in lines 393-395, the results shows using variance leads to poor performance. How to interpret this?

---

> ### Author Response · Authors · 2024-11-24
> **Summary on Rebuttal to Reviewer zXTk**
>
> **Summary.** We sincerely thank the reviewer for all the constructive feedback to help improve the `eva` method. We have prepared our [rebuttal PDF](https://openreview.net/attachment?id=TMYe4rUuTc&name=supplementary_material) at this [anonymous link](https://openreview.net/attachment?id=TMYe4rUuTc&name=supplementary_material) to provide a clearer presentation of the added figures and equations for your convenience.
>
> In response, we have provided:
> - **experiments on more iterations** in Section D.2.
> - **extended discussions on the regret objective and the proxy** in Section G.
> - **evidence on the distinction between advantage-based metrics and variance-based ones** in Section F.
>
> We believe the additional discussions and rebuttals have sufficiently addressed the weaknesses and questions raised by the reviewer. We sincerely appreciate the reviewer’s insights and look forward to any further discussions.

---

> > ### Comment · Reviewer_zXTk · 2024-11-27
> >
> > In Line 1399, why is this objective ideal? Optimizing the prompt distribution seems like cheating by selecting easier prompts for a higher reward.

---

> > > ### Author Response · Authors · 2024-11-28
> > > **On the open-ended objective**
> > >
> > > Thanks for the comment! We have revised our manuscript and responded in full at Q2 of the new [**Rebuttal PDF for Reviewer zXTK**](https://drive.google.com/file/d/1omRIdYTbNfd8gHByEEXG25UXNoxFiRyp/view?usp=sharing) ([anonymous link](https://drive.google.com/file/d/1omRIdYTbNfd8gHByEEXG25UXNoxFiRyp/view?usp=sharing)). For this question:
> > >
> > > ---
> > > **[The shorter answer]**:
> > > The intended design is to avoid cheating (**in principle**: adding KL regularization to guide the creator policy towards the open-ended reference; **in practice**: using the creator to generate new prompts to maximize the solver's regret). We also present further **empirical evidence** showing `eva` helps create more complex prompts.
> > >
> > > ---
> > >
> > > **[The longer answer]**:
> > > **Conceptually**, cheating occurs when the reference distribution is narrow or incorrectly defined. In our case, it is important that $p_{\text{ref}}(x)$ represents an underspecified, potentially intractable probability distribution over possible tasks (instantiated via prompts) in the wild. This serves as an open-ended reference that encompasses the full diversity and complexity of tasks agents may encounter (not the initial static prompt set). The joint regularization towards $ \pi_{\text{ref}}(x, y) $ captures the objective for agents to generalize alignment in $ p_{\text{ref}}(x) $ with broader open-ended prompts while remaining close to the SFT policy $ \pi_{\text{SFT}}(y | x) $. In summary, the conceptual definition of $ p_{\text{ref}}(x) $ and the regularization helps avoid the collapsing to distributions with easier prompts.
> > >
> > >
> > >
> > > **Practically**, we do not directly optimize this principle; rather, we design a creator-solver game to implicitly approach this by iteratively creating a sequence of informative prompt distributions for training. It is important that we use *regret* as the objective and approximate it by the estimate of the *optimal reward advantage*. This approach avoids selecting prompts that are too easy or too hard by design. See also Section 3.4 on auto-curricula and learning potential for prompt selection, and Appendix G for more connections between the objective and the algorithm.
> > >
> > > **Empirically**, we have run additional experiments evaluating the complexity and quality of `eva`'s prompt distribution in Appendix E. As shown below, there is a gradual improvement of prompt complexity and quality over iterations.
> > >
> > > | **Prompt Set ($\downarrow$) / Metric ($\rightarrow$)** | **Complexity** (1-5) | **Quality** (1-5) |
> > > |--------------------------------------------------------|-----------------------|-------------------|
> > > | UltraFeedback (seed)                                   | 2.90                 | 3.18             |
> > > | **UltraFeedback-eva-Iter-1**                      | 3.84                 | 3.59             |
> > > | **UltraFeedback-eva-Iter-2**                      | 3.92                 | 3.63             |
> > > | **UltraFeedback-eva-Iter-3**                      | **3.98**             | **3.73**         |
> > >
> > > We also observe the creator auto-prioritizes learning in problems like math and coding, which are initially hard for the solver. Thus the creator also practically does not select easier prompts in the `eva` game. Please check the relevant figures in our Appendix E ([**link**](https://openreview.net/pdf?id=TMYe4rUuTc)).
> > >
> > > In addition, the literature of curriculum RL, open-ended learning and so on are about designing the right metric for the agents to learn increasingly complex and general capabilities, and we summarize at Appendix H for the reviewer's reference.
> > >
> > > ---
> > >
> > >
> > > Please let us know if this help address your concerns, and we are happy to discuss more future works.
> > >
> > >
> > > Happy Thanksgiving,
> > > Authors of Paper #12746

---

> ### Author Response · Authors · 2024-12-02
> **Engaging in Further Discussion**
>
> Dear Reviewer zXTK,
>
> Today is the last day of the discussion period. Previoysly, we have uploaded a **[Rebuttal PDF for Reviewer zXTK](https://drive.google.com/file/d/1omRIdYTbNfd8gHByEEXG25UXNoxFiRyp/view?usp=sharing)**. For your reference, we re-organize the major content below:
>
> > **W1 (Running for More Iterations)**: The number of iterations in the main results is 2, with only one EVA step in each experiment, which is a little different from what the demonstration in Figure 3 shows. If the eva step is performed multiple times, would the results be better or worse? What is performance like when you access all data in UltraFeedback?
>
> **TL; DR**: We have added experimental results in Appendix D on running more iterations with more data, where `eva` remains to be effective.
>
> **Rebuttal**: As an addition to Section 4.2.4, we have experimented with the following settings:
> - 10K prompts per iteration with 3 iterations.
> - 20K prompts per iteration with 3 iterations (i.e., all UltraFeedback prompts are used).
> - 60K prompts per iteration with 2 iterations (i.e., all UltraFeedback prompts are used).
>
> Due to time constraints, we did not perform an extensive hyper-parameter search; however, we believe the results presented below sufficiently demonstrate the performance gains achieved by `eva`.
>
>
> | No. Prompts Per Iteration ($\rightarrow$)     |          10K        |
> |-----------------------------------------------|---------------------|
> | **Method ($\downarrow$) / Metric ($\rightarrow$)** | **WR (%)** |
> | $\theta _{0 \rightarrow 1}$: DPO          | 51.6   |
> | $\theta _{1 \rightarrow 2}$: DPO          | 59.8   |
> | $\theta _{2 \rightarrow 3}$: DPO          | 61.2   |
> | $\theta _{1 \rightarrow \tilde{1}}$: + **`eva`**          | **60.1**   |
> | $\theta _{\tilde{1} \rightarrow \tilde{2}}$: + **`eva`**  | **62.0**   |
> | $\theta _{\tilde{2} \rightarrow \tilde{3}}$: + **`eva`**  | **62.2**   |
>
>
> | No. Prompts Per Iteration ($\rightarrow$)     |          20K        |
> |-----------------------------------------------|---------------------|
> | **Method ($\downarrow$) / Metric ($\rightarrow$)** | **WR (%)** |
> | $\theta _{0}$: SFT                              | 41.3   |
> | $\theta _{0 \rightarrow 1}$: DPO               | 53.2   |
> | $\theta _{1 \rightarrow 2}$: DPO               | 47.0   |
> | $\theta _{2 \rightarrow 3}$: DPO               | 46.8   |
> | $\theta _{1 \rightarrow \tilde{1}}$: + **`eva`** | **59.5**   |
> | $\theta _{\tilde{1} \rightarrow \tilde{2}}$: + **`eva`** | **60.0**   |
> | $\theta _{\tilde{2} \rightarrow \tilde{3}}$: + **`eva`** | **61.4**   |
>
> | No. Prompts Per Iteration ($\rightarrow$)     |          60K        |
> |-----------------------------------------------|---------------------|
> | **Method ($\downarrow$) / Metric ($\rightarrow$)** | **WR (%)** |
> | $\theta _{0}$: SFT                              | 41.3   |
> | $\theta _{0 \rightarrow 1}$: DPO               | 58.9   |
> | $\theta _{1 \rightarrow \tilde{1}}$: + **`eva`** | **59.6**   |
> | $\theta _{\tilde{1} \rightarrow \tilde{1}'}$: + **`eva`** | **61.9**   |
>
> In general, **`eva` can bring robust gains with multiple iterations**.
>
> Please check our rebuttal PDF and Appendix A & D for training specifications and detailed analysis; we also present a bonus experiment with a rewriter in the loop which further improves `eva`'s performance.
>
>
> ****
>
> > **W2 (Connection in Minimax Regret and The Algorithm)**: The connection between the minimax regret objective and the algorithm is a somehow vague. The regret concerns the performance gap with the optimal policy. It’s not reflected by the informativeness proxy.
>
> **TL;DR**: We have added Appendix G to clearly illustrate the connection in details. In short:
> - In the current algorithm, the `solver` explicitly minimizes the regret by plug-in preference optimization algorithms (e.g., DPO), while the `creator` implicitly maximizes the regret by first finding high-regret prompts and generate variations as new prompt distributions for training.
> - The **informativeness proxy** is an **advantage-based estimate of the regret** via the stochastic policy; similar variants have been commonly used in prior literature like  [Jiang et al. (2021b)](https://arxiv.org/abs/2010.03934); [Parker-Holder et al. (2022)](https://accelagent.github.io/), as a standard approximation when we do not have the access to the optimal policy. Our choice of proxy is implementation-efficient and shows consistently strong empirical gains in vast experiments.
>
> Please take a look at our Rebuttal PDF for more background and detailed illustration.

---

> ### Author Response · Authors · 2024-12-02
> **Engaging in Further Discussion**
>
> > **Q1 (Advantage v.s. Variance)**: The informativeness proxy seems to be similar to the variance of the rewards because they all concern the diversity of the generated responses. However, in lines 393-395, the results show using variance leads to poor performance. How to interpret this?
>
> **TL;DR**: We have added Appendix F to help readers interpret this. In short, (**i**) **conceptually**, variance does not directly capture the **learning potential** in preference optimization, while advantage-based informativeness proxy is better aligned to the learning objective; (**ii**) **empirically**, we show that variance and advantage are only **weakly correlated**, thus they will likely result in different sampling.
>
> Please check Figure 8 and Figure 9 in the Rebuttal PDF for more empirical evidence.
>
> ****
>
> **Final Remarks.**  We thank Reviewer zxTK for the constructive feedback. We have made careful efforts to resolve all the weaknesses and questions raised. Please take a moment to check our rebuttal and let us know if any further clarification is needed -- we are always happy to discuss, and would be grateful if the reviewer may kindly consider again the rating for **`eva`**.
>
> With the new open-ended RLHF principle, the easy-to-implement method, and the strong empirical gains, we believe `eva` can offer valuable contribution to the alignment community.
>
> **[[Rebuttal PDF for Reviewer zXTK](https://drive.google.com/file/d/1omRIdYTbNfd8gHByEEXG25UXNoxFiRyp/view?usp=sharing)]**

---

### Official Review · Reviewer_i9kx · 2024-11-04

**Soundness:** 2
**Presentation:** 3
**Contribution:** 2
**Rating:** 3
**Confidence:** 4

**Summary:**

Majority of the recent alignment/RLHF pipelines have been designed on static prompt distribution leading to several issues of distribution shift on different prompt distribution. Hence to mitigate this issue, the current work formulates the problem as an asymmetric game where the creator generates novel and informative prompts whereas the solver solves it. This approach improves the performance of existing baselines on SOTA benchmarks.

**Strengths:**

The reliance of static prompt distribution has inherently limited the capabilities of various alignment designs eventually resulting in issues such as distribution shifts and even jailbreaks can be considered as a special case of the same. Hence, the problem statement is timely and crucial. The paper introduces a creator-solver framework evolving the prompt distribution which enhances the generalization ability of LLMs, allowing them to adapt to more diverse and challenging prompts, which better mirrors real-world applications. An additional advantage lies in the fact, the algorithm can be added on top of current baselines which is crucial rather than building new algorithm from scratch. The empirical results show improvements on existing benchmarks when applied on top of baseline algorithms.

**Weaknesses:**

1. How is Equation 10 tractable and how is it being solved? Any heuristic of sampling and approximating should result in sub-optimality which is not clear where its accounted.
2. The optimization is over \pi in equation 9 for solving the minimax regret. However, its not absolutely clear how the KL divergence plays a role here and how it is ensured that the response and prompt distributions are close to reference. Without that, the alignment problem is ill-defined. Please provide concrete justifications in theory and empirical results.
3. As described in Algorithm 1, informativeness is evaluated and a prompt subset is created based on current policy estimate and then the policy is updated based on the prompt subset. However, this causes an inter-dependence between the two which leads to nested structure, which is not clearly explained. Specifically, while computing the informativeness score for the prompts, it depends on \theta^*(x_t-1) i.e optimal parameter for the previous distribution. Provide clear explaination on the same.
4. While iterating, every new prompt distribution will require generating new response, how is the evaluation coming from which reward model?  Is the ground reward available, if not please explain how the preference is obtained and how does it affect suboptimality?
5. Overall, which expression/Theorem guides us in understanding the improvement of prior suboptimality is not clear? Can you please point out/highlight how the current method improves upon the prior suboptimality due to static prompt distribution?
6. Its extremely crucial to show the prompt distribution and demonstrate its perplexity to ensure its not generating some meaningless or irrelevant prompts, since its not very evident on the KL divergence in the prompt space and its relation with the informative measure. Please provide detailed explaination to clarify that.

**Questions:**

1. Since equation 7, can't be directly solved, and is solved in an asymmetric fashion, then in the solver loop the KL should be over the response distribution and not joint right?
2. How is the KL divergence w.r.t reference policy for the algorithm? Please provide detailed ablation.
3. Whats the reward model availability? Is the true reward model available?
4. There is a recent line of works on Stacklberg and Bilevel RLHF which deals with the entanglement in a leader-follower setting. Although not specific to updating prompt dist, but can be trivially applied. Provide a detailed comparison with the literature around that [1,2, 3].
5. Can you provide intuitions behind equation 7, on the KL divergence between the joint policy for both prompt and response? Is it even tractable to estimate or approximate this KL


[1]. Principled Penalty-based Methods for Bilevel Reinforcement Learning and RLHF
[2]. SAIL: Self-Improving Efficient Online Alignment of Large Language Models
[3]. STA-RLHF: Stackelberg Aligned Reinforcement Learning with Human Feedback

---

> ### Author Response · Authors · 2024-11-24
> **Summary on Rebuttal to Reviewer i9kx**
>
> **Summary.** We thank the reviewer for the thoughtful and detailed feedback. We have carefully revised the manuscript, and provided a new **[Rebuttal PDF for Reviewer i9kx](https://drive.google.com/file/d/1LGplgJy6PCu1RTED-Iiyna8o29EEmEP-/view?usp=sharing)** ([anonymous link](https://drive.google.com/file/d/1LGplgJy6PCu1RTED-Iiyna8o29EEmEP-/view?usp=sharing)) -- we kindly invite you to review it at your convenience..
>
> In response, we have:
>
> - provided a point-by-point rebuttal **fully addressing every suggested weakness/question**;
> - **improved definitions** for Algorithm 1, the regret and the informativeness proxy in Section 3;
> - **added a detailed review** on bi-level RLHF and open-ended learning in Appendix I and Appendix H;
> - **added detailed discussion** to illustrate the method, from the principle to the asymmetric game setting, then to the regret minimization by the solver and maximization by the creator in Appendix G;
> - **added new experiments** on empirical gains and prompt evaluation of `eva` in Appendix D and Appendix E.
>
> We thank the reviewer's precious time and efforts helping us improving this submission. We have made every effort to thoroughly address the concerns raised and revise our manuscript accordingly. We welcome any future discussion, and would be grateful if you may consider revising the score if you find the revision satisfactory.
>
> Happy Thanksgiving,
> Authors of Paper #12746
>
> *p.s.* We will re-organize the PDF content onto markdown format for OpenReview later (if that format works better!). In the mean time, please feel free to share any additional insights, and we are happy to discuss more future works.

---

> > ### Author Response · Authors · 2024-11-30
> > **Follow-Up on the Rebuttal**
> >
> > Dear Reviewer i9kx,
> >
> > We previously prepared a **[Rebuttal PDF for Reviewer i9kx](https://drive.google.com/file/d/1LGplgJy6PCu1RTED-Iiyna8o29EEmEP-/view?usp=sharing)** at this [anonymous link](drive.google.com/file/d/1LGplgJy6PCu1RTED-Iiyna8o29EEmEP-/view?usp=sharing). To help with the review process, we now re-organize the content below:
> >
> > > **Q5 (Intuition)**: Can you provide intuitions behind equation 7, on the KL divergence between the joint policy for both prompt and response? Is it even tractable to estimate or approximate on this KL?
> > > **W2 (Regret and KL)**: The optimization is over $\pi$ in Eq. 9 for solving the minimax regret. However, its not absolutely clear how the KL divergence plays a role here and how it is ensured that the response and prompt distributions are close to reference. Without that, the alignment problem is ill-defined. Please provide concrete justifications in theory and empirical results.
> > > **W1 (Proxy)**: How is Eq. 10 tractable and being solved? Any heuristic of sampling and approximating should result in sub-optimality which is not clear where its accounted.
> >
> > We added a comprehensive illustration in Appendix G in our paper. For short answers:
> >
> > **[Q5 - Intuition]**
> > The joint regularization towards $\pi_{\text{ref}}(\mathbf{x}, \mathbf{y})$ captures the open-ended objective for agents to (**i**) optimize in $p_{\text{ref}}(x)$ with broader open-ended tasks for more general alignment capabilities (note $p_{\text{ref}}(x)$ represents the underspecified open-ended reference as a conceptual guide, *not* the initial static set $\mathcal{D}$), while (**ii**) staying close to $\pi_{\text{SFT}}(y | x)$. To see the two probability matching problems more clear, we can reformulate the principle:
> > $$
> > \max _{\phi, \theta} \mathbb{E} _{\mathbf{x} \sim \pi _\phi(\cdot)}\left[\mathbb{E} _{\mathbf{y} \sim \pi _\theta(\cdot \mid \mathbf{x})}[r(\mathbf{x}, \mathbf{y})] - \beta _1 \mathbb{D} _{\mathrm{KL}}\left[\pi _\theta(\mathbf{y} \mid \mathbf{x}) \| \pi _{\mathrm{SFT}}(\mathbf{y} \mid \mathbf{x})\right]\right]-\beta_2 \mathbb{D} _{\mathrm{KL}}\left[\pi _\phi(\mathbf{x}) \| p _{\mathrm{ref}}(\mathbf{x})\right].
> > $$
> > It is tractable to solve $\mathbb{D} _{\mathrm{KL}}[\pi _\theta(\mathbf{y} \mid \mathbf{x}) \| \pi _{\mathrm{SFT}}(\mathbf{y} \mid \mathbf{x})]$ by existing plug-in preference optimization algorithms, yet we need to approximate $\mathbb{D} _{\mathrm{KL}}[\pi _\phi(\mathbf{x}) \| p _{\mathrm{ref}}(\mathbf{x})]$ -- one way to achieve this heuristically is by iteratively creating a *sequence* of prompt distributions directing its learning towards new, informative prompts to continue training. This is the key unconventional approach that `eva` takes.
> >
> > Please see Appendix G for detailed explanation, along with our review in open-ended learning, where the main theme is on automatically designing curriculum to expand across the task space to develop increasingly general capabilities.
> >
> > **[Q2 - KL and well-defined alignment]**
> > We have revised Eq. 9 (thanks for catching this!) so that it is more clear that the regret is the difference in expected reward of the current and KL-optimal policy.
> > $$
> > \text{Regret}\left(\pi _\phi, \pi _{{\theta}}\right)=\mathbb{E} _{\mathbf{x} \sim \pi _\phi(\cdot)}\left[\mathbb{E} _{\mathbf{y} \sim \pi _{\boldsymbol{\theta}}(\mathbf{y} \mid \mathbf{x})}[r(\mathbf{x}, \mathbf{y})]-\mathbb{E} _{\mathbf{y} \sim \pi _{\mathrm{kL}}^*(\mathbf{y} \mid \mathbf{x})}[r(\mathbf{x}, \mathbf{y})]\right]
> > $$
> > For the **solver**, by design, preference optimization would be equivalent to regret minimization, thus the alignment problem remains to be correctly defined. For the **creator**, the distribution matching to the open-ended reference is implicitly achieved by prompt curriculum construction, and we present empirical evidence in Appendix E to justify that prompts are evolving towards broader tasks with higher complexity.
> >
> > **[W1 - Proxy]**
> > We have revised Definition 2 for better readability. It is estimated by sampling multiple responses from the stochastic policy and calculating the reward advantage (or other proxies). Specifically, for the advantage-based estimate:
> > $$
> > |\hat{\text{Regret}}(\mathbf{x}, \pi _{\theta})| \gets  \text{info} _{\theta}(\mathbf{x}) :=   r(\mathbf{x}, \mathbf{y} _{+}) - r(\mathbf{x}, \mathbf{y} _{\text{baseline}}),
> > $$
> > where
> > $$
> > \mathbf{y} _{+} := {{\arg \max}} _{\mathbf{y} _i} \, r(\mathbf{x}, \mathbf{y}), \quad \mathbf{y} _{\text{baseline}} := {{\arg \min}} _{\mathbf{y} _i} \, r(\mathbf{x}, \mathbf{y}) \text{ or }  \arg _{\mathbf{y} _i} \, r(\mathbf{x}, \mathbf{y}),
> > $$
> > and $\{ \mathbf{y} _i \} _{i=1}$ is a set of responses sampled from $\pi _{\theta}(\cdot \mid \mathbf{x})$ and $r(\cdot, \cdot)$ is the reward oracle. We leave exploration of other informativeness proxy designs in `eva` to future work. This approximation will result in sub-optimality for creator’s regret maximization process. Please see our Appendix G for details.

---

> > > ### Author Response · Authors · 2024-11-30
> > > **Follow-Up on the Rebuttal**
> > >
> > > > **W3 (Understanding the Iterative Algorithm)**: As described in Algorithm 1, informativeness is evaluated and a prompt subset is created based on current policy estimate and then the policy is updated based on the prompt subset. However, this causes an inter-dependence between the two which leads to nested structure, which is not clearly explained. Specifically, while computing the informativeness score for the prompts, it depends on $\theta^\star(x_{t-1})$ , *i.e.*, optimal parameter for the previous distribution. Provide clear explaination on the same.
> > >
> > >
> > > **TL;DR**: (i) We revised Algo. 1 with updated subscripts to reflect the training process -- please take a look in our [main paper](https://openreview.net/pdf?id=TMYe4rUuTc). Given a current model checkpoint, we evaluate the prompt informativeness based on it, and evolve a new prompt set more informative to the current checkpoint, and use the new prompt set for continual training. (ii) We intend to use an iterative best-response framework to approximate equilibrium in expectation, balancing computational efficiency and practicality.
> > >
> > > **Rebuttal**: The iterative updates in `eva`, as described in Algo. 1, are based on a best-response-to-best-response framework. Specifically, the creator updates the prompt distribution based on the solver’s current policy, and the solver then optimizes its policy over the updated prompts, and the process repeats. This sequential structure approximates a Nash equilibrium in expectation over iterations, inspired by works such as Freund and Schapire (1999); Wu et al. (2024), which establish convergence to optimal policies on average through iterative optimization.
> > >
> > > We intentionally avoid simultaneous joint optimization as it would significantly increase computational and memory overhead, making it less practical for integration into current RLHF pipelines.
> > >
> > > ****
> > >
> > > > **W4 (Understanding Reward Models)**: While iterating, every new prompt distribution will require generating new response, how is the evaluation coming from which reward model? Is the ground reward available, if not please explain how the preference is obtained and how does it affect suboptimality?
> > > > Also **Q3 (RM Availability)**: What's the reward model availability? Is the true reward model available?
> > >
> > >
> > > **TL;DR**: We assume a preference oracle provided by an external, pre-trained reward model, which is practically  used in many real-world LLM training scenarios ([Team et al., 2023](https://arxiv.org/abs/2312.11805)).
> > >
> > > **Answer**: As discussed in the beginning of Section 4, we assume the availability of a pre-trained, fixed reward model. This approach is practically adopted in industry ([Team et al., 2023](https://arxiv.org/abs/2312.11805); [2024a](https://arxiv.org/abs/2403.05530);[b](https://arxiv.org/abs/2408.00118)) and is also commonly used in academia works ([Meng et al., 2024](https://arxiv.org/abs/2405.14734); [Wu et al., 2024](https://arxiv.org/abs/2405.00675)). The reason is more on efficiency concerns. For example, in practice, the reward model is often *an order of magnitude larger* than the policy ([Team et al., 2024b](https://arxiv.org/abs/2408.00118)); it would be impractical, or the gain may only be marginal, if we update the reward model on-the-fly.
> > >
> > >
> > > Nevertheless, it is possible to incorporate the online RM training within `eva` -- we have shown in Section 4.2.3 (ablation studies) that `eva` scales with quality of reward models, thus integrating online RM training may further enhance performance and address the potential distribution mismatch problem. We believe this is an interesting direction to pursue, and have listed it in Section 6 (future works) on adding more players including rewarders in the self-play loop.

---

> > > > ### Author Response · Authors · 2024-11-30
> > > > **Follow-Up on the Rebuttal**
> > > >
> > > > > **W5 (Improvement of Sub-Optimality)**: Overall, which expression/Theorem guides us in understanding the improvement of prior suboptimality is not clear? Can you please point out/highlight how the current method improves upon the prior suboptimality due to static prompt distribution?
> > > >
> > > >
> > > > **TL;DR**: The improvement of sub-optimality is guided by the minimax regret objective (Remark 1) through its iterative implementation. While this work does not explicitly derive suboptimality bounds, our approach has demonstrated strong empirical gains over the training by static distributions, as shown in Section 4 (main experiments), Appendix E (benchmark performance), and Appendix D.2 (alignment gains over iterations).
> > > >
> > > > **Answer**: In general, the improvement of prior suboptimality (due to static prompt distributions) is guided by the minimax game outlined in Remark 1. This expression forms the basic foundation for our iterative algorithm, where the creator updates prompts to maximize informativeness (proxy for regret), and the solver minimizes regret (through direct preference optimization). This iterative process ensures the solver and creator adapt to each other, implicitly forming a curriculum and addressing sub-optimality  in static prompts. We also added Appendix G to help illustrate the intuition behind.
> > > >
> > > > The empirical results in Section 4 (main experiments), Appendix E (benchmark performance), and Appendix D.2 (alignment gains over iterations) demonstrate that the dynamic prompt distribution improves solver performance and alignment metrics, thereby mitigating suboptimality. While the current package does not explicitly derive sub-optimality bounds (as would be typical in formal RL/bandit theory papers) and emphasizes practicality and usability as a methodology paper, we would love to learn if the reviewer has any suggestions for this as the future work.
> > > >
> > > > ****
> > > >
> > > > > **Q1 and Q2 (KL in the Solver Loop)**: Since equation 7, can't be directly solved, and is solved in an asymmetric fashion, then in the solver loop the KL should be over the response distribution and not joint right? How is the KL divergence w.r.t reference policy for the algorithm? Please provide detailed ablation.
> > > >
> > > > **Answer**: (**i**) Yes, in the solver loop, the KL regularization is applied over the response distribution, not the joint distribution, as shown in Algorithm 1 and explained in the rebuttal to W2. (**ii**) The KL divergence *w.r.t.* the reference policy is determined by the plug-in solver (*e.g.*, DPO), which is orthogonal to our framework. We have added detailed explanation in Appendix G to illustrate the whole process.

---

> > > > > ### Author Response · Authors · 2024-11-30
> > > > > **Follow-Up on the Rebuttal**
> > > > >
> > > > > > **W6 (Prompt Distribution)**: It is extremely crucial to show the prompt distribution and demonstrate its perplexity to ensure its not generating some meaningless or irrelevant prompts, since its not very evident on the KL divergence in the prompt space and its relation with the informative measure. Please provide detailed explanation to clarify that.
> > > > >
> > > > > **TL; DR**: Please see (**i**) new experimental results in Appendix E (prompt distribution visualization) and Appendix J (prompt examples), which verify that `eva` evolves meaningful and relevant prompts with improved complexity and quality; and (**ii**) added explanation in Appendix G on the conceptual KL regularization in the prompt space.
> > > > >
> > > > > **Answer**: We have revised the manuscript with additional visualization on potential curriculum learned in **Appendix E**. In general, we observe the creator prioritizes learning in math and coding for the generated prompt distribution, which brings gradual improvement on benchmark performance on relevant categories over iterations. In other words, `eva` effectively shifts focus towards harder yet learnable categories. Please take a look at our [paper](https://openreview.net/pdf?id=TMYe4rUuTc) for the relevant distributions and radar plots (Figure 12 and 13), and examples of generated prompts (Table 24).
> > > > >
> > > > > As noted in Appendix G, the solver maintains KL regularization during optimization, ensuring that the response distribution remain close to the reference policy;  in the this work, we do not explicitly add KL regularization in the prompt distribution since we do not directly conduct parameter update for the creator (which we empirically find to bring training instability); rather, we use *meta instructions* and *buffer sampling* to constrain the prompt generations (as described in Section 3.3), which is empirically very effective, and introduces only minimal changes to existing pipeline thus can be easily applied.
> > > > >
> > > > > Regarding the relation with the informativeness measure, our current proxy is an efficient proxy among many possibilities to approximate regret (where one way to implicitly achieve $\mathbb{D} _{\mathrm{KL}}\left[\pi _\phi(\mathbf{x}) \| p _{\mathrm{ref}}(\mathbf{x})\right]$ is by training on the prompt curriculum created by the creator with regret maximization as its strategy). We have provided detailed discussions in Appendix G to help interpret it.
> > > > >
> > > > > We note that perplexity is not commonly used nor the most preferred  measure for data quality in practical training of large language models ([Team et al., 2023](https://arxiv.org/abs/2312.11805); [Fang et al., 2024](https://arxiv.org/pdf/2410.23771)), and can be computationally heavy to measure. We have added experiments in Appendix E to generatively measure the complexity and quality of prompt distributions.  As in Table below, there is a gradual improvement of prompt complexity and quality over iterations with `eva`.  Combined with the empirical evidence mentioned earlier, we believe the concerns should have been adequately addressed.
> > > > >
> > > > > | **Prompt Set ($\downarrow$) / Metric ($\rightarrow$)** | **Complexity** (1-5) | **Quality** (1-5) |
> > > > > |--------------------------------------------------------|-----------------------|-------------------|
> > > > > | UltraFeedback (seed)                                   | 2.90                 | 3.18             |
> > > > > | **UltraFeedback-eva-Iter-1**                      | 3.84                 | 3.59             |
> > > > > | **UltraFeedback-eva-Iter-2**                      | 3.92                 | 3.63             |
> > > > > | **UltraFeedback-eva-Iter-3**                      | **3.98**             | **3.73**         |

---

> > > > > > ### Author Response · Authors · 2024-11-30
> > > > > > **Follow-Up on the Rebuttal**
> > > > > >
> > > > > > > **Q4 (Literature)**: There is a recent line of works on Stacklberg and Bilevel RLHF which deals with the entanglement in a leader-follower setting. Although not specific to updating prompt dist, but can be trivially applied. Provide a detailed comparison with the literature around that [1,2,3].
> > > > > >
> > > > > > **Answer**: We thank the reviewer for this nice suggestion. Please see below for a detailed literature review and comparison on the three references. We have added Appendix I in the paper for this.
> > > > > >
> > > > > >
> > > > > > Bi-level optimization refers to optimization problems where the cost function is defined w.r.t. the optimal solution to another optimization problem (Grosse, 2022). There is a recent line of works applying bi-level optimization to RLHF. While they all rely on a fixed dataset of prompts, eva propose to dynamically update the prompt set. We hereby present a detailed comparison of eva with Ding et al. (2024); Shen et al. (2024); Makar-Limanov et al. (2024). We thank the anonymous reviewer for the kind references, and welcome suggestions for any other works we may have missed.
> > > > > >
> > > > > > Ding et al. (2024) formulate iterative online RLHF as a bi-level optimization problem, where the upper-level represents the reward learning, and the lower-level represents the policy optimization. Leveraging reward re-parameterization tricks in Rafailov et al. (2023), Ding et al. (2024) reduces the problem to a single-level objective with regard to the policy. The differences of this work and our work lie in the prompt distribution and preference oracle: (i) eva features by dynamic prompt set generation for Open-Ended RLHF, whereas (Ding et al., 2024) remains using a static prompt set; (ii) we assume the existence of the preference oracle (as discussed in § 4), while Ding et al. (2024) consider online training of reward models and ablate on self-rewarding by the current LLM policy. Our usage of a pre-trained reward model follows from industrial practices (Team et al., 2023; 2024b), which is also commonly used by prior works in academia (Meng et al., 2024; Wu et al., 2024).
> > > > > >
> > > > > > Makar-Limanov et al. (2024) provide an interesting exploration on formulating RLHF as a leaderfollower game, where the language model (LM) policy is the leader and the reward model (RM) policy is the follower, and the solution is Stackelberg equilibrium (von Stackelberg, 1934; Rajeswaran et al., 2020), where the leader does not likewise best respond to the follower's strategy. Here, following the curriculum RL literature (Dennis et al., 2020; Parker-Holder et al., 2022), we seek the Nash equilibrium (Nash et al., 1950) between the creator for prompt generation and the solver for response generation. Nevertheless, the LM-RM iterative optimization may be added on top of eva's framework, and we look forward to future works exploring the leader-follower re-formulation of eva.
> > > > > >
> > > > > > Shen et al. (2024) present a rigorous theoretical work (though it does not directly involve practical post-training of large language models). The authors propose to reduce the bi-level problem to a single-level problem with a penalty-based reformulation, and apply it in the setting of LM-RM optimization within a fixed environment, whereas eva focuces on dynamic prompt generation and practically train large language models with extensive empirical experiments conducted. We believe it would be interesting to adapt similar first-order optimization techniques to solve Open-Ended RLHF.
> > > > > >
> > > > > >
> > > > > > In summary, existing bi-level RLHF works focus on online optimization of both the RM and the LM (as the response policy), all with fixed prompt/state distribution. eva presents an orthogonal direction on dynamic prompt generation for Open-Ended RLHF, with an empirical algorithm which attains state-of-the-art performance with large language models on a variety of benchmarks. It is possible to incorporate the online RM training within eva - we have shown in Section 4.2.3 that eva scales with quality of reward models, thus integrating online RM training may further enhance performance and mitigate potential distributional mismatch problems as we evolves for more prompts. This direction may have not been widely adopted in real-world training of language models, due to concerns about practicality (Team et al., 2023; 2024a;b; Adler et al., 2024). We look forward to future works exploring efficient variations unifying eva and existing bi-level RM-LM frameworks.

---

> > > > > > > ### Author Response · Authors · 2024-11-30
> > > > > > > **Follow-Up on the Rebuttal**
> > > > > > >
> > > > > > > **Final remarks.**  We sincerely thank the reviewer for the precious time and efforts on the `eva` method. We value all those opinions, and have made every effort to thoroughly address the concerns raised and revise our manuscript accordingly. Regarding the rejection, we hope the reviewer may kindly consider the points that we have summarized at the beginning of this rebuttal, on the **strong empirical alignment gain** brought by the **simple design** of eva, also on judging the merit of a work (cf., ([Castro, 2021](https://psc-g.github.io/posts/mentoring/reviewing/#conclusion))) *w.r.t.* the practicality and how the community may easily build on top of the principle and the method we proposed (cf., ([Hamming, 1986](https://d37ugbyn3rpeym.cloudfront.net/stripe-press/TAODSAE_zine_press.pdf))), which we are confident are valuable to the broader alignment community. We look forward to any future discussion, and would be grateful if the reviewer may consider revising the score if the revision is satisfactory.
> > > > > > >
> > > > > > > **Rebutal PDF**: [anonymous link](https://drive.google.com/file/d/1LGplgJy6PCu1RTED-Iiyna8o29EEmEP-/view?usp=sharing).

---

> > > > ### Public Comment · ~Lai_Wei7 · 2026-02-07
> > > >
> > > > Hello, authors. I think this work is interesting. But I also have a similar question. As you mention that you assume a preference oracle provided by an external pre-trained reward model, which is practically used in many real-world LLM training scenarios, does it mean that this is not a stict self-play setting? Because we need an external supervision from the external reward model. In my opinion, self-play maybe means that there is totally no external supervision. We just use the policy model to self play. Looking forward to your response. Thanks a lot!!

---

> > > > > ### Public Comment · ~Ziyu_Ye1 · 2026-02-12
> > > > > **Replying to Public Comment by Lai Wei**
> > > > >
> > > > > Hello Lai, thanks for your interest in this preliminary work. Yes, we use a fixed model as the reward oracle. Re: "strict" self-play: i am uncertain if it is meaningful to develop reinforcement learning agents without any external feedback; we thought a continual and/or active learning setting might be more useful to research at the current stage, where learning agents are incentivized to actively acquire external signals and adapt to the real environment.

---

> ### Author Response · Authors · 2024-12-02
>
> Dear Reviewer,
>
> Tomorrow is the deadline for our discussion period. We have carefully considered your valuable feedback and have made every effort to thoroughly address the concerns raised. It would be great if you could take some time to review our responses, and we would be more than delighted to discuss with you!
>
> Many thanks,
> The Authors of Paper #12746
> [**[Rebuttal PDF for Reviewer i9kx](https://drive.google.com/file/d/1LGplgJy6PCu1RTED-Iiyna8o29EEmEP-/view)**]

---

### Official Review · Reviewer_aC2p · 2024-11-05

**Soundness:** 3
**Presentation:** 3
**Contribution:** 2
**Rating:** 8
**Confidence:** 3

**Summary:**

This paper looks to bring RLHF training past using a fixed prompt distribution. The authors pose the learning problem as an asymmetric game between two players: a prompt creator and solver. The authors then introduce eva, which is a simple prompt evolution approach that can be used in conjunction with existing RLHF algorithms. eva improves the performance of DPO, SimPO, SPPO, and ORPO
on widely-used benchmarks within the community.

**Strengths:**

- It is very cool to see a curriculum learning approach leading to improvements in practical RLHF settings with LLMs.
- I thought the writing quality and presentation of the paper was generally pretty strong throughout with the discourse feeling well structured.
- I appreciated the way that the approach was motivated through discussion of minimax regret in section 3.2.
- I also found the connection with learning progress highlighted in section 3.4 quite interesting and feel that my concerns about the heuristic nature of the approach are largely addressed through this motivation.
- The results in Table 1 are very impressive, especially in comparison to human prompts.
- It is nice to see that the success of eva is very robust to the particular RLHF optimizer used.
- A nice set of benchmarks and ablations are considered for the experiments.
- I also thought the robustness with respect to more training iterations was a nice benefit to highlight.

**Weaknesses:**

- The novelty of the approach is not huge based on the prior literature, although I do believe a very practical contribution is made.
- The approach is motivated intuitively and includes heuristics i.e. the "Informativeness Proxy" rather than a mathematical derivation of the optimization rule from first principles.
- Evolving seems important for achieving performance based off table 4, but the particular algorithm chosen is not really described or contrasted with other approaches. This is discussed in the "Prompt synthesis for language models" section of the related work as an orthogonal contribution because other related approaches could be plugged into evolve(·), but it would be very interesting to understand more about how critical this particular choice is for the success of eva.
- There are some typos that the authors missed such as "DIFFERENT INTUITIVE WYAS" and "Limitations and future directionss".

**Questions:**

Q1: Could you explain more about the particular choice of evolution algorithm used in your implementation of eva and different potential strengths and weaknesses related to this choice?

Q2: Do you see empirical evidence of your intuition about learning progress discussed in section 3.4? It seems like some of these claims are directly testable.

Q3: Could you visualize the curriculum learned in your experiments with eva? It would be very nice to get an intuition for why performance improves and what the heuristic prioritizes over time.

Minor - Q4: When discussing future directions, the authors write  "(vii) further scaling up w/ million-level data". Can you clarify what this means? Seems like some important context is missing?

---

> ### Author Response · Authors · 2024-11-24
> **Summary on Rebuttal to Reviewer aC2p**
>
> **Summary.** We sincerely thank the reviewer for their appreciation of the `eva` method and their constructive feedback. To provide a clearer presentation of the added figures and equations, we have prepared our **[rebuttal PDF](https://drive.google.com/file/d/1yyIhda3vXLYr52fs7i72_loVZMz9NJ-U/view?usp=sharing)** at this [anonymous link](https://drive.google.com/file/d/1yyIhda3vXLYr52fs7i72_loVZMz9NJ-U/view?usp=sharing). (If the hyperlink in the rebuttal PDF does not work, please refer to our [main paper](https://openreview.net/pdf?id=TMYe4rUuTc), which also includes the rebuttal text following the Appendix.)
>
> In response, we have:
> - added experiments on implementing **different evolving methods** and discussed their relevant strengths and weaknesses in Section D.1;
> - included **visualizations on the learning curriculum** in Section E;
> - provided a detailed discussion on **scaling up** the `eva` method.
>
> We appreciate the reviewer’s insights which helps improve our paper, and warmly look forward to future discussions and opportunities to learn from you.

---

> ### Comment · Reviewer_aC2p · 2024-12-03
>
> Thank you for your thorough response to my questions. I think the clarity is much improved with respect to the choice of evolving method now. I also really appreciate the effort from the authors in visualizing the learned curriculum, which I think will be interesting to many readers.
>
> I think the authors over thought my question regarding "million-level data." I just literally have never heard of this "level" terminology before and did not know what was meant by this i.e. like a definition. Terms in the response like "ten-thousands level" were just as confusing in this regard. I guess it refers to the number of tokens based on context I am picking up in the author's response?

---

> > ### Author Response · Authors · 2024-12-03
> >
> > Thank you so much for your thoughtful feedback. We were referring to tens of thousands of prompts (e.g., UltraFeedback includes around 60K prompts) and more -- will fix the terminology in future revisions!
> >
> > Kind regards,
> > The Authors

---

### Author Response · Authors · 2024-11-26
**Update on Paper #12746 – Upcoming Revisions**

Dear reviewers –

Happy Thanksgiving! Thank you very much for your time and efforts helping us improving the `eva` method. We noticed that the discussion period has been extended to December 2, and we are currently finalizing our revisions to fully address your valuable feedback. We plan to share an updated manuscript and rebuttal tomorrow.

Our planned updates include:
- **Method**: expanded technical discussions to bridge the gap between the methodology and practical implementation.
- **Experiments**: additional results on (i) implementing `eva` with more iterations and data; (ii) visualization of the prompt curriculum with additional evaluation; (iii) further evidence supporting the validity of the proposed informativeness proxy, and more.
- **Literature Review**: in-depth discussions on bi-level RLHF and open-ended learning.

We look forward to upcoming discussions with you all.

Cheers,
Authors of Paper #12746

---

### Author Response · Authors · 2024-12-03
**Last Day of Discussion - Summary of Revisions**

We thank all the reviewers for your valuable suggestions. We have previously uploaded a **[revision paper](https://openreview.net/pdf?id=TMYe4rUuTc)** on OpenReview, and a **[rebuttal PDF](https://drive.google.com/file/d/1t16wLCa_UPQHUk7Qm-9-RU8z-eFajPeP/view?usp=sharing)** to everyone.

As a recap, the proposed `eva` is an **easy-to-use method** that allows **scalable preference fine-tuning beyond static human prompts**. It (**i**) lays a *new principle* for open-ended RLHF, (**ii**) presents a new and *easy-to-implement iterative algorithm* that can be flexibly integrated into any existing alignment algorithms, and (**iii**) shows *strong performance gains* over existing SOTA benchmarks.

We are encouraged that reviewers find the merit of `eva` with regard to its:
- **Impactful real-world problem**: *"timely and crucial"* (`i9kx`); *"better mirrors real-world applications"* (`i9kx`).
- **Intuitive and practical algorithm**: *"the algorithm can be added on top of current baselines which is crucial rather than building new algorithm from scratch"* (`i9kx`); *"evolving the prompt distribution which enhances the generalization ability of LLMs"* (`i9kx`); *"cool to see a curriculum learning approach leading to improvements"* (`aC2p`); *"connection with learning progress ... quite interesting"* (`aC2p`).
- **Impressive empirical results**: *"results in table 1 are very impressive"* (`aC2p`); *"improves the performance of existing baselines on SOTA benchmarks"* (`i9kx`); *"empirical evaluation shows much improvement"* (`zXTk`).
- **Thorough ablation studies**: *"a nice set of benchmarks and ablations"*, *"robustness with respect to more training iterations"* (`aC2p`);  *"thorough ablation study"* (`zXTk`);
- **Overall quality**: *"well-written, easy-to-follow, and informative"* (`zXTk`), *"pretty strong writing quality and presentation"* (`aC2p`).

In revision, we have thoroughly addressed each and every weakness and concern. These primarily centered around: (**i**) connections between the principle and the algorithm; (**ii**) effectiveness of the method under different settings. We summarize key revisions as follows:
- **Detailed explanations to the methods**:
    - Section 3: more clear definitions on the open-ended RLHF principle, the regret w.r.t. the KL-optimal policy, and the approximation by the informativeness proxy for the creator.
    - Appendix G: a very detailed illustration on the method, from the principle to the asymmetric game setting, then to the practical algorithm for regret minimization and maximization.
- **Further experiments on `eva`'s effectiveness**:
    - Appendix D.1: `eva` is robust under different evolving methods;
    - Appendix D.2: `eva` is effective over multiple iterations;
    - Appendix J: `eva` synthesizes meaningful prompts and responses;
    - Appendix E: `eva` automatically creates a prompt curriculum with increasing complexity and quality;
    - Appendix F: `eva`'s informativeness proxy is distinctive from baseline heuristics like variance;
    - Appendix H and J: detailed literature reviews on open-ended learning and minimax regret games (as the background) and bi-level RLHF (as orthogonal works).

As a concluding remark, we have designed `eva` for it to be easy-to-use by everyone. We hope it can serve as a scalable starting point for alignment practitioners to build open-ended, robust, and self-improving language models that align with human values.

We thank every reviewer once again for your time and efforts that help make `eva` better.

Kind regards,
The Authors of Paper #12746

***p.s.,*** Today is the last day for reviewers to interact with us. We believe all the concerns raised should have been properly addressed, and we encourage you to **please review our rebuttal** and let us know your thoughts if the revisions are satisfactory. Your feedback would mean a lot to us and to the community!

---

### Comment · Area_Chair_SThc · 2024-12-03
**Engage in discussion**

Dear Reviewer i9kx and Reviewer zXTK,

Could you take a look at the author response?

Best,
AC

---

### Meta-Review · Area_Chair_SThc · 2024-12-23

**Metareview:**

Current training paradigms only leverage static distribution of prompts and it suffers from efficiency issues and generalization issues. Motivated by it, the authors proposed a new framework eva, which is able to create new prompts. eva improves the performance of the current training method.

The paper is well-written and the authors conduct substantial experiments to illustrate the performance of the proposed algorithm. However, the novelty of this paper is limited. Moreover, the intuition behind the design of the algorithm is not entirely stated, such as the connection between the minimax regret and informativeness proxy, which constitutes the main reason for rejection.

**Additional Comments On Reviewer Discussion:**

* Reviewer zXTk raised concerns about the connections between the principle and the algorithm, the authors included further explanations in section 3 and Appendix G.
* Reviewer i9kx asked about the method's effectiveness under different settings, and the authors updated the relevant content in Appendix D, J, E, F, H.

---

### Decision · Program_Chairs · 2025-01-22

Reject